# Shadowheart SGD: Distributed Asynchronous SGD with Optimal Time Complexity Under Arbitrary Computation and Communication Heterogeneity

**Alexander Tyurin**
KAUST,* AIRI,† Skoltech‡

**Marta Pozzi**
KAUST,* University of Pavia§

**Ivan Ilin**
KAUST*

**Peter Richtárik**
KAUST*

## Abstract

We consider *nonconvex stochastic* optimization problems in the *asynchronous centralized distributed* setup where the communication times from workers to a server can not be ignored, and the computation and communication times are potentially different for all workers. Using an unbiassed compression technique, we develop a new method—Shadowheart SGD—that provably improves the time complexities of all previous centralized methods. Moreover, we show that the time complexity of Shadowheart SGD is optimal in the family of centralized methods with compressed communication. We also consider the bidirectional setup, where broadcasting from the server to the workers is non-negligible, and develop a corresponding method.

## 1 Introduction

We consider the nonconvex smooth optimization problem

$$\min_{x \in \mathbb{R}^d} \left\{ f(x) := \mathbb{E}_{\xi \sim \mathcal{D}_\xi} \left[ f(x; \xi) \right] \right\}, \tag{1}$$

where $f(\cdot; \cdot) : \mathbb{R}^d \times \mathbb{S}_\xi \to \mathbb{R}$, and $\mathcal{D}_\xi$ is a distribution on $\mathbb{S}_\xi \neq \emptyset$. Given $\varepsilon > 0$, we seek to find a possibility random point $\hat{x}$ such that $\mathbb{E}[\|\nabla f(\hat{x})\|^2] \leq \varepsilon$. Such a point $\hat{x}$ is called an $\varepsilon$–stationary point. We focus on solving the problem in the following setup:

(a) $n$ *workers/nodes* are able to compute *stochastic* gradients $\nabla f(x; \xi)$ of $f$, *in parallel* and *asynchronously*, and it takes (at most) $h_i$ seconds for worker $i$ to compute a single stochastic gradient;

(b) the workers are connected to a *server* which acts as a communication hub;

(c) the workers can communicate with the server *in parallel* and *asynchronously*; it takes (at most) $\tau_i$ seconds for worker $i$ to send a *compressed* message to the server; compression is performed via applying lossy communication compression to the communicated message (a vector from $\mathbb{R}^d$); see Def. 2.1;

(d) the server can broadcast compressed vectors to the workers in (at most) $\tau_{\text{serv}}$ seconds; compression is performed via applying a lossy communication compression operator to the communicated message (a vector from $\mathbb{R}^d$); see Def. A.1.

The main goal of this work is to find an *optimal* optimization strategy/method that would work uniformly well in all scenarios characterized by the values of the computation times $h_1, \ldots, h_n$ and

---

*King Abdullah University of Science and Technology, Thuwal, Saudi Arabia

†AIRI, Moscow, Russia

‡Skolkovo Institute of Science and Technology, Moscow, Russia

§University of Pavia, Pavia, Italy

38th Conference on Neural Information Processing Systems (NeurIPS 2024).

Table 1: **Time Complexities of Centralized Distributed Algorithms.** Assume that it takes at most $h_i$ seconds to worker $i$ to calculate a stochastic gradient and $\dot{\tau}_i$ seconds to send *one coordinate/float* to server. Abbreviations: $L$ = smoothness constant, $\varepsilon$ = error tolerance, $\Delta = f(x^0) - f^*$, $n$ = # of workers, $d$ = dimension of the problem. We take the Rand$K$ compressor with $K = 1$ (Def. D.1) (as an example) in QSGD and Shadowheart SGD. Due to Property 6.2, the choice $K = 1$ is optimal for Shadowheart SGD up to a constant factor.

| Method | Time Complexity | $\max\{h_n, \dot{\tau}_n\} \to \infty$, $\max\{h_i, \dot{\tau}_i\} < \infty\ \forall i < n$ (the last worker is slow) | Time Complexities in Some Regimes $h_i = h, \dot{\tau}_i = \dot{\tau}\ \forall i \in [n]$ (equal performance) | Numerical Comparison[b] $\sigma^2/\varepsilon =$ | | |
| --- | --- | --- | --- | --- | --- | --- |
| | | | | 1 | $10^3$ | $10^6$ |
| Minibatch SGD (see (2)) | $\max\limits_{i \in [n]} \max\{h_i, d\dot{\tau}_i\}\left(\frac{L\Delta}{\varepsilon} + \frac{\sigma^2 L\Delta}{n\varepsilon^2}\right)$ | $\infty$ (non-robust) | $\max\{h, d\dot{\tau}, \frac{d\dot{\tau}\sigma^2}{n\varepsilon}, \frac{h\sigma^2}{n\varepsilon}\}\frac{L\Delta}{\varepsilon}$ (worse, e.g., when $\dot{\tau}$, $d$ or $n$ large) | $\times 10^3$ | $\times 10^3$ | $\times 10^4$ |
| QSGD (see (5)) (Alistarh et al., 2017) (Khaled and Richtárik, 2020) | $\max\limits_{i \in [n]} \max\{h_i, \dot{\tau}_i\}\left(\left(\frac{d}{n}+1\right)\frac{L\Delta}{\varepsilon} + \frac{d\sigma^2 L\Delta}{n\varepsilon^2}\right)$ | $\infty$ (non-robust) | $\geq \frac{dh\sigma^2}{n\varepsilon}\frac{L\Delta}{\varepsilon}$ (worse, e.g., when $\varepsilon$ small) | $\times 3$ | $\times 10^2$ | $\times 10^4$ |
| Rennala SGD (Tyurin and Richtárik, 2023c), Asynchronous SGD (e.g., (Mishchenko et al., 2022)) | $\geq \min\limits_{j \in [n]} \max\left\{h_{\pi_j}, d\dot{\tau}_{\pi_j}, \frac{\sigma^2}{\varepsilon}\left(\sum\limits_{i=1}^{j}\frac{1}{h_{\pi_i}}\right)^{-1}\right\}\frac{L\Delta}{\varepsilon}$ [a] | $< \infty$ (robust) | $\geq \max\left\{h, d\dot{\tau}, \frac{h\sigma^2}{n\varepsilon}\right\}\frac{L\Delta}{\varepsilon}$ (worse, e.g., when $\dot{\tau}$, $d$ or $n$ large) | $\times 10^2$ | $\times 10$ | $\times 1.5$ |
| Shadowheart SGD (see (9) and Alg. 1) (Corollary 4.3) | $t^*(d-1, \sigma^2/\varepsilon, [h_i, \dot{\tau}_i]_1^n)\frac{L\Delta}{\varepsilon}$ [c] | $< \infty$ (robust) | $\max\left\{h, \dot{\tau}, \frac{d\dot{\tau}}{n}, \sqrt{\frac{d\dot{\tau}h\sigma^2}{n\varepsilon}}, \frac{h\sigma^2}{n\varepsilon}\right\}\frac{L\Delta}{\varepsilon}$ | $\times 1$ | $\times 1$ | $\times 1$ |
| Lower Bound (see Section 5 and Theorem O.5) | $t^*(d-1, \sigma^2/\varepsilon, [h_i, \dot{\tau}_i]_1^n)\frac{L\Delta}{\varepsilon}$ [d] | — | — | — | — | — |

The time complexity of Shadowheart SGD is not worse than the time complexity of the competing centralized methods, and is *strictly better in many regimes* (see Section 7.1).
We show that (10) is the *optimal time complexity* in the family of centralized methods with compression (see Section 5 and Theorem O.5).

[a] Upper bound time complexities are not derived for Rennala SGD and Asynchronous SGD. However, we can derive the lower bound using Theorem O.5 with $\omega = 0$. One should take $d\dot{\tau}_i$ instead of $\tau_i$ when apply Theorem O.5 because these methods send $d$ coordinates. $\bar{\pi}$ is a permutation that sorts $\max\{h_i, d\dot{\tau}_i\}$: $\max\{h_{\pi_1}, d\dot{\tau}_{\pi_1}\} \leq \cdots \leq \max\{h_{\pi_n}, d\dot{\tau}_{\pi_n}\}$
[b] We numerically compute time complexities for $d = 10^6$, $n = 10^3$, $h_i \sim U(0.1, 1)$, $\dot{\tau}_i \sim U(0.1, 1)$ (uniform i.i.d.), and three noise regimes $\sigma^2/\varepsilon \in \{1, 10^3, 10^6\}$. We report the factors by which the time complexities of the competing methods are worse compared to the time complexity of our method Shadowheart SGD. So, for example, Minibatch SGD, QSGD and Asynchronous SGD can be worse by the factors $\times 10^4$, $\times 10^4$, and $\times 10^2$, respectively.
[c] The mapping $t^*$ is defined in Def. 3.1.
[d] The lower bound constructed with the Rand$K$ compressor and the dimension $d = \Theta(L\Delta/\varepsilon)$.

communication times $\tau_1, \ldots, \tau_n$ and $\tau_{\text{serv}}$. Since we allow these times to be arbitrarily heterogeneous, designing a single algorithm that would be optimal in all these scenarios seems challenging.

Let us clarify what we mean by $\{\tau_i\}$ and take, for instance, the Rand$K$ compressor (see Def. D.1 and Sec. 2.2) (sends $K$ random entries). Then $\tau_i$ is # of seconds required to send $K$ coordinates of a vector. Now assume that the communication is proportional to the number of coordinates/bits that a worker sends, i.e., it takes $\dot{\tau}_i$ seconds to send a *one coordinate/bit*. Then, clearly, we have $\tau_i = K \times \dot{\tau}_i$. and, $\tau_i$ is a function of $K$. While the dependence is clear for Rand$K$, for all possible compressors, the dependence on the amount of sent information can be less nontrivial. We generally fix an arbitrary unbiased compressor and assume it takes $\tau_i$ seconds to send the compressed message.

From the viewpoint of federated learning (Konečný et al., 2016; Kairouz et al., 2021), our work is a theoretical study of device heterogeneity. Moreover, our formalism captures both *cross-silo* and *cross-device* settings as special cases. Due to our in-depth focus on device heterogeneity and the challenges that need to be overcome, we do not consider statistical heterogeneity, and leave an extension to this setup to future work.

We rely on assumptions which are standard in the literature on stochastic gradient methods: smoothness, lower-boundedness and bounded variance.

**Assumption 1.1.** $f$ is differentiable & $L$–smooth, i.e., $\|\nabla f(x) - \nabla f(y)\| \leq L\|x - y\|, \forall x, y \in \mathbb{R}^d$.

**Assumption 1.2.** There exist $f^* \in \mathbb{R}$ such that $f(x) \geq f^*$ for all $x \in \mathbb{R}^d$. We define $\Delta := f(x^0) - f^*$, where $x^0 \in \mathbb{R}^d$ is a starting point of all algorithms we consider.

**Assumption 1.3.** For all $x \in \mathbb{R}^d$, the stochastic gradients $\nabla f(x; \xi)$ are unbiased, and their variance is bounded by $\sigma^2 \geq 0$, i.e., $\mathbb{E}_\xi[\nabla f(x; \xi)] = \nabla f(x)$ and $\mathbb{E}_\xi[\|\nabla f(x; \xi) - \nabla f(x)\|^2] \leq \sigma^2$.

To simplify the exposition, in what follows we first focus on the regime in which the broadcast cost can be ignored. We describe a strategy for extending our algorithm to the more general regime in Sec. A.

## 2 Related Work

### 2.1 Communication time can be ignored

We now briefly review related work and important concepts. Consider the regime when the communication cost is negligible ($\tau_i = 0$ for all $i$), and the computation times $h_i$ are arbitrary

but fixed. It is well-known that under Assumptions 1.1, 1.2, and 1.3, the vanilla SGD method $x^{k+1} = x^k - \gamma \nabla f(x^k; \xi^k)$, where $x^0 \in \mathbb{R}^d$ is a starting point, and $\gamma > 0$ is the step size, solves (1) using *the optimal number of stochastic gradients* (Ghadimi and Lan, 2013; Arjevani et al., 2022). Since the # of iterations of SGD to get an $\varepsilon$–stationary point is $\mathrm{O}\left(L\Delta/\varepsilon + \sigma^2 L\Delta/\varepsilon^2\right)$, SGD run on a *single worker* whose computation time is $h_1$ seconds would have *time complexity* $\mathrm{O}\left(h_1 \times \left(L\Delta/\varepsilon + \sigma^2 L\Delta/\varepsilon^2\right)\right)$ seconds. The time complexity of Minibatch SGD with $n$ workers, i.e.,

$$x^{k+1} = x^k - \tfrac{\gamma}{n} \sum_{i=1}^{n} \nabla f(x^k; \xi_i^k), \tag{2}$$

can be shown (Gower et al., 2019) to be

$$\mathrm{O}\left(h_{\max} \times \left(\tfrac{L\Delta}{\varepsilon} + \tfrac{\sigma^2 L\Delta}{n\varepsilon^2}\right)\right), \tag{3}$$

where $h_{\max} := \max_{i \in [n]} h_i$, where $[n]$ denotes $\{1, \dots, n\}$. The dependence on $h_{\max}$ is due to Minibatch SGD employing *synchronous* parallelism which forces it to wait for the slowest worker. While the stochastic part of (3) can be $n$ times smaller than in the single worker case, (3) does not guarantee an improvement since $h_{\max}$ can be arbitrarily large. In real systems, computation times can be very heterogeneous and vary in time in chaotic ways (Dutta et al., 2018; Chen et al., 2016).

Recently, Cohen et al. (2021); Mishchenko et al. (2022) and Koloskova et al. (2022) showed that it is possible to improve upon (3) using the celebrated Asynchronous SGD method (Recht et al., 2011; Feyzmahdavian et al., 2016; Nguyen et al., 2018) and get the time complexity $\mathrm{O}((1/n \sum_{i=1}^{n} \frac{1}{h_i})^{-1} \times \left(L\Delta/\varepsilon + \sigma^2 L\Delta/n\varepsilon^2\right))$, which improves the dependence from $h_{\max}$ to the harmonic mean of the computation times. Subsequently, Tyurin and Richtárik (2023c) developed the Rennala SGD method whose time complexity is

$$\mathrm{O}\left(\min_{m \in [n]} \left(\tfrac{1}{m} \sum_{i=1}^{m} \tfrac{1}{h_{\pi_i}}\right)^{-1} \times \left(\tfrac{L\Delta}{\varepsilon} + \tfrac{\sigma^2 L\Delta}{m\varepsilon^2}\right)\right), \tag{4}$$

where $\pi$ is a permutation for which $h_{\pi_1} \leq \cdots \leq h_{\pi_n}$. They also showed that the time complexity (4) is *optimal* by providing a matching lower bound.

## 2.2 Communication time is a factor

In many practical scenarios, communication times can be the main bottleneck, and can not be ignored, e.g., in distributed/federated training of machine learning models (Ramesh et al., 2021; Kairouz et al., 2021; Wang et al., 2023). There are two main techniques for reducing the communication bottleneck: local training steps (McMahan et al., 2017) and compressed communication (Seide et al., 2014; Alistarh et al., 2017). In our work, we investigate the latter technique. In particular, efficient methods with compressed communication such as DIANA (Mishchenko et al., 2019), Accelerated DIANA (Li et al., 2020), MARINA (Gorbunov et al., 2021) and DASHA (Tyurin and Richtárik, 2023b) employ *unbiased compressors*, defined next. Assume that $\mathbb{S}_\nu$ is a nonempty arbitrary set of samples, and $\mathcal{D}_\nu$ is a distribution on $\mathbb{S}_\nu$.

**Definition 2.1.** A mapping $\mathcal{C} : \mathbb{R}^d \times \mathbb{S}_\nu \to \mathbb{R}^d$ is an *unbiased compressor* if there exists $\omega \geq 0$ such that $\mathbb{E}_\nu[\mathcal{C}(x;\nu)] = x$, $\mathbb{E}_\nu[\|\mathcal{C}(x;\nu) - x\|^2] \leq \omega \|x\|^2$ for all $x$. Let $\mathbb{U}(\omega)$ denote the family of such compressors[5].

**Assumption 2.2.** Samples from $\mathcal{D}_\xi$ and $\mathcal{D}_\nu$ are mutually independent.

The canonical example of an unbiased compressor is the Rand$K$ compressor (see Def. D.1) that scales $K$ random entries of the input vector $x$ by $d/K$ and zeros out the rest. Many more examples of unbiased compressors are considered in the literature (Beznosikov et al., 2020; Xu et al., 2021a; Horváth et al., 2022). One of the most straightforward methods which use compression is QSGD[6] (Alistarh et al., 2017):

$$x^{k+1} = x^k - \tfrac{\gamma}{n} \sum_{i=1}^{n} \mathcal{C}_i \left(\nabla f(x^k; \xi_i^k)\right), \tag{5}$$

---

[5]For convenience, following the previous literature, we use the shortcuts $\mathcal{C}(x;\nu) \equiv \mathcal{C}(x)$ and $\mathcal{C}(x;\nu_{ij}) \equiv \mathcal{C}_{ij}(x)$ assuming that $\nu$ and $\nu_{ij}$ are mutually independent.

[6]It is also called the distributed compressed stochastic gradient descent method (DCGD/DCSGD) (Khaled and Richtárik, 2020).

where each worker calculates one stochastic gradient, compresses it using $\mathcal{C}_i \in \mathbb{U}(\omega)$ drawn independently, and sends it to the server. The server aggregates the compressed vectors and performs step (5). With a proper stepsize choice $\gamma$, QSGD converges after $O\left((\omega/n + 1) \times L\Delta/\varepsilon + (\omega + 1) \times \sigma^2 L\Delta/n\varepsilon^2\right)$ iterations[7] (Khaled and Richtárik, 2020). Let's assume it takes $\tau_i$ seconds for worker $i$ to send one compressed vector to the server. Since the workers act in parallel, the time complexity of QSGD is

$$\max_{i \in [n]} (h_i + \tau_i) \times \left((\tfrac{\omega}{n} + 1)\tfrac{L\Delta}{\varepsilon} + (\omega + 1)\tfrac{\sigma^2 L\Delta}{n\varepsilon^2}\right). \tag{6}$$

We can go through a similar exercise with any other method that uses compressed communication (e.g., (Tyurin and Richtárik, 2023a; Gauthier et al., 2023; Jia et al., 2023)). Nevertheless, as far as we know, *the optimal time complexities for asynchronous centralized distributed optimization with communication compression are not known.*

## 3 Summary of Contributions

In the regime in which the communication time can be ignored (see Sec. 2.1), Tyurin and Richtárik (2023c) showed that (4) is the optimal time complexity. In this work we endeavor to take the next step: we wish to understand the fundamental limits of the regime in which communication time is a factor. Our main contributions are:

♠ We develop a new method—Shadowheart SGD (Algorithm 1)—that guarantees to find an $\varepsilon$–stationary point of problem (1) with time complexity $T_*$ given in (10). While the general expression we give for $T_*$ is hard to parse since it involves the equilibrium time $t_*(\cdot)$ whose definition is implicit (see Def. 3.1), we show (see Sec. 7) that $T_*$ is not worse than the time complexity of known centralized[8] methods, and also who that it can be *strictly* better in many regimes, even by *many degrees of magnitude* (see Sec. 7.1 and Table 1).

♣ In Sec. 5 we show that (10) is the *optimal time complexity* in the family of centralized methods with compression. This is the first such result in the literature.

♦ We also developed Adaptive Shadowheart SGD (Sec. 4.2 and M), which does not require the knowledge of the computation and communication times and can work with arbitrary changing times. Moreover, we designed Bidirectional Shadowheart SGD (Sec. A), which works in the regime when broadcast cost not negligible as well.

♥ Our theoretical study of Shadowheart SGD is supported by judiciously designed synthetic experiments and machine-learning experiments with logistic regression; see Sec. Q.

## 4 Development of Shadowheart SGD

Our method bears some resemblance to Rennala SGD (Tyurin and Richtárik, 2023c) and QSGD (Alistarh et al., 2017), and involves some additional algorithmic elements which play a key role. First, we adopted the main suggestion of Tyurin and Richtárik (2023c)[Sec.7] behind the design of Rennala SGD that an optimal method should calculate stochastic gradients at the last iterate. Second, QSGD served as an inspiration for how to perform gradient compression. In particular, Shadowheart SGD has the form $x^{k+1} = x^k - \gamma g^k$, where

$$g^k = \sum_{i=1}^{n} w_i \sum_{j=1}^{m_i} \mathcal{C}_{ij}\left(\sum_{l=1}^{b_i} \nabla f(x^k; \xi_{il}^k)\right) / \sum_{i=1}^{n} w_i m_i b_i. \tag{9}$$

In Shadowheart SGD, worker $i$ calculates $b_i$ stochastic gradients, adds them up to form $\sum_{l=1}^{b_i} \nabla f(x^k; \xi_{il}^k)$, and compresses the result $m_i$ times using independently drawn compressors. The compressed messages are sent to the server. The first non-trivial step in the design of our method is the presence of weights $w_i$: the server aggregates the $\sum_{i=1}^{n} m_i$ compressed messages across all workers by performing a conic combination with coefficient $\frac{w_i}{\sum_{i=1}^{n} w_i m_i b_i}$ for messages coming from worker $i$. One can easily show that (9) is equivalent to Alg. 1. Note that we recover QSGD (see (5)) as a special (suboptimal) case with $w_i = b_i = m_i = 1$ for all $i \in [n]$.

---

[7]For $\omega = 0$, the rate reduces to the rate of Minibatch SGD.

[8]We say that a method is *centralized* if the workers calculate stochastic gradients only at points calculated by the server.

**Algorithm 1** Shadowheart SGD

1: **Input:** starting point $x^0 \in \mathbb{R}^d$, stepsize $\gamma > 0$, ratio $\sigma^2/\varepsilon$, computation times $h_i > 0$, and communication times $\tau_i > 0$ for $i \in [n]$
2: Find equilibrium time $t^*$ using Def. 3.1
3: Set $b_i = \left\lfloor \frac{t^*}{h_i} \right\rfloor$ and $m_i = \left\lfloor \frac{t^*}{\tau_i} \right\rfloor$ for all $i \in [n]$
4: Find $S_A = \{i \in [n] : b_i \wedge m_i > 0\}$
5: **for** $k = 0, 1, \ldots, K-1$ **do**
6:     Run Alg. 2 in active workers $S_A$
7:     Broadcast $x^k, b_i, m_i$ to active workers $S_A$
8:     Initialize $g^k = 0$
9:     **for** $i \in S_A$ **in parallel do**
10:         $w_i \overset{(a)}{=} \left( b_i \omega + \omega \frac{\sigma^2}{\varepsilon} + m_i \frac{\sigma^2}{\varepsilon} \right)^{-1}$
11:         **for** $j = 1, \ldots, m_i$ **do**
12:             Receive $\mathcal{C}_{ij}\left(g_i^k\right)$ from worker $i$
13:             $g^k = g^k + w_i \mathcal{C}_{ij}\left(g_i^k\right)$
14:         **end for**
15:     **end for**
16:     $g^k = g^k / \left( \sum_{i=1}^n w_i m_i b_i \right)$
17:     $x^{k+1} = x^k - \gamma g^k$
18: **end for**

---

**Algorithm 2** Strategy of Worker $i$

1: Receive $x^k, b_i, m_i$ from server, init $g_i^k = 0$
2: **for** $l = 1, \ldots, b_i$ **do**
3:     Calculate $\nabla f(x^k; \xi_{il}^k)$,   $\xi_{il}^k \sim \mathcal{D}_\xi$
4:     $g_i^k = g_i^k + \nabla f(x^k; \xi_{il}^k)$
5: **end for**
6: **for** $j = 1, \ldots, m_i$ **do**
7:     Send $\mathcal{C}_{ij}\left(g_i^k\right) \equiv \mathcal{C}\left(g_i^k; \nu_{ij}^k\right)$ to server, $\nu_{ij}^k \sim \mathcal{D}_\nu, \mathcal{C}_{ij} \in \mathbb{U}(\omega)$
8: **end for**

---

**Definition 3.1** (Equilibrium Time).
A mapping $t^* : \underbrace{\mathbb{R}_{\geq 0}}_{\omega} \times \underbrace{\mathbb{R}_{\geq 0}}_{\sigma^2/\varepsilon} \times \underbrace{(\mathbb{R}_{\geq 0} \times \mathbb{R}_{\geq 0})}_{(h_1, \tau_1)}$

$\times \cdots \times \underbrace{(\mathbb{R}_{\geq 0} \times \mathbb{R}_{\geq 0})}_{(h_n, \tau_n)} \to \mathbb{R}_{\geq 0}$

with inputs $\omega, \sigma^2/\varepsilon, h_1, \tau_1, \ldots, h_n, \tau_n$ is called the *equilibrium time* if it is defined as follows. Find a permutation[a] $\pi$ that sorts the pairs $(h_i, \tau_i)$ as $\max\{h_{\pi_1}, \tau_{\pi_1}\} \leq \cdots \leq \max\{h_{\pi_n}, \tau_{\pi_n}\}$ and find the solution $s^*(j) \in [0, \infty]$ in $s$ of[b]

$$\left( \sum_{i=1}^j \frac{1}{2\tau_{\pi_i}\omega + \frac{4\tau_{\pi_i} h_{\pi_i} \sigma^2 \omega}{s \times \varepsilon} + \frac{2h_{\pi_i} \sigma^2}{\varepsilon}} \right)^{-1} = s \tag{7}$$

for all $j \in [n]$. Then the mapping returns the value

$$t^*(\omega, \sigma^2/\varepsilon, h_1, \tau_1, \ldots, h_n, \tau_n)$$
$$\equiv \min_{j \in [n]} \max\{\max\{h_{\pi_j}, \tau_{\pi_j}\}, s^*(j)\} \in [0, \infty]. \tag{8}$$

We shall use the short notation $t^*(\omega, \sigma^2/\varepsilon, [h_i, \tau_i]_1^n)$.

---

[a]It is possible that a permutation is not unique. The result of the mapping does not depend on the choice of the permutation. See the proof of Property 4.1.

[b]For convenience, we use *the projectively extended real line* and define $1/0 = \infty$.

$(a)$ : If $\omega = 0$ and $\frac{\sigma^2}{\varepsilon} = 0$, then $w_i = 1$

---

The weights $\{w_i\}$ are chosen so as to minimize the variance in the proof of Lemma H.1. However, we still need to find the right values for $b_i$ and $m_i$. Since the computation and communication times of worker $i$ are $h_i$ and $\tau_i$, respectively, the following strategy makes intuitive sense: the server sets some time budget $t$ for all workers, and each worker then calculates $\lfloor t/h_i \rfloor$ stochastic gradients and sends $\lfloor t/\tau_i \rfloor$ compressed vectors to the server. But what is the right way to choose $t$? If $t$ is too small, then, intuitively, some workers may not have time to calculate "enough" gradients, or may even not have time to send any messages to the server. On the other hand, if $t$ is too large, then the workers will eventually send information of diminishing utility which will not be worth the extra time this takes.

We find out that, and this one of the key insights of our work, that there exists an optimal time budget $t^*$ which depends on the quantities $\omega, \sigma^2/\varepsilon, h_1, \tau_1, \ldots, h_n, \tau_n$, for which we coin the name *equilibrium time*; see Def. 3.1. Admittedly, the definition of the equilibrium time is implicit; we do not know if it is possible to give a more explicit formula in general. To provide for some peace of mind, we prove the following property:

**Property 4.1.** *If all inputs of the equilibrium time are non-negative, then the equilibrium time is well defined.*

More importantly, in Sec. 5 we provide a *lower bound* that involves the same mapping. Thus, the equilibrium time is not an "artifact" of our method, but is of a fundamental nature. We use the equilibrium time $t^*$ in Shadowheart SGD when we choose $b_i$ and $m_i$.

Our first main result provides *iteration complexity:*

**Theorem 4.2.** *Lett Assumptions 1.1, 1.2, 1.3, 2.2 hold. Let us take $\gamma = 1/2L$ in* Shadowheart SGD *(Alg. 1). Then as long as $K \geq 16L\Delta/\varepsilon$, we have the guarantee $\frac{1}{K}\sum_{k=0}^{K-1} \mathbb{E}\left[\left\|\nabla f(x^k)\right\|^2\right] \leq \varepsilon$.*

This result guarantees that Shadowheart SGD will converge after $\mathcal{O}\left(L\Delta/\varepsilon\right)$ iterations. Our second main result provides a much more relevant complexity measure: *time complexity.*

**Corollary 4.3.** Shadowheart SGD *(Alg. 1) converges after at most $T_*$ seconds, where*

$$T_* := \frac{32L\Delta}{\varepsilon} \times t^*(\omega, \sigma^2/\varepsilon, h_1, \tau_1, \ldots, h_n, \tau_n). \tag{10}$$

Surprisingly, we show in Sec. 5 that our time complexity guarantee (10) is *optimal* for the family of centralized methods with compressed communication. Moreover, in Sec. 7 and 7.1, we show that (10) is no worse and can be significantly better than the time complexities of previous centralized methods (see also Table 1 for a summary).

## 4.1 Tighter result with per-iteration times $h_i^k$ and $\tau_i^k$

A slight modification of Alg. 1 leads to Alg. 4, which can work with iteration-dependent computation and communication times $h_i^k$ and $\tau_i^k$. Our main result in this setup is Theorem H.3; here we present its corollary.

**Theorem 4.4.** *Alg. 4 converges after*

$$\sum_{k=0}^{\left\lceil\frac{16L\Delta}{\varepsilon}\right\rceil} 2t^*(\omega, \sigma^2/\varepsilon, h_1^k, \tau_1^k, \ldots, h_n^k, \tau_n^k) \tag{11}$$

*seconds, where $h_i^k > 0$ and $\tau_i^k > 0$ are computation and communication times for worker $i$ in iteration $k$.*

For presentation simplicity sake, in the main part we continue to work with static $\{h_i\}$ and $\{\tau_i\}$.

## 4.2 On the problem of estimating the times in Algorithms 1 and 4

One of the main features of asynchronous methods (e.g., Rennala SGD, Asynchronous SGD) is their adaptivity to and independence from processing times. In Sec. M, we design Adaptive Shadowheart SGD (Alg. 7) with this feature. Unlike Alg. 1, it does not require the knowledge of $\{h_i\}$ and $\{\tau_i\}$ (or $\{h_i^k\}$ and $\{\tau_i^k\}$ in the case of Alg. 4), and does not calculate the equilibrium time $t^*$. However, as a byproduct of this flexibility, this method has a slightly worse time complexity guarantee. In order to present our result, we need to define an auxiliary sequence.

**Definition 4.5.** Assume that the workers have computation and communication times less or equal to $\{h_i\}$ and $\{\tau_i\}$. Assume that $\bar{h}_{ij}$ is the actual time required to calculate the $j^{\text{th}}$ stochastic gradient by worker $i$, $h_{\min} > 0$ is the smallest possible computation time. Then

$$r_i := \sup_{k \geq 0} \frac{\sup_{1 \leq j \leq l_{\max}} \bar{h}_{i,(k+j)}}{\inf_{1 \leq j \leq l_{\max}} \bar{h}_{i,(k+j)}}, \quad l_{\max} := \left\lceil\frac{t_{\max}}{h_{\min}}\right\rceil,$$

$t_{\max} := 128 \times t^*(\omega, \sigma^2/\varepsilon, [\max\{h_i, \tau_i\}, \max\{h_i, \tau_i\}]_1^n)$.

That is, $r_i \in [1, \infty]$ is the largest ratio between the fastest and the slowest computation of stochastic gradients in local time windows. $r_i$ defines a degree of fluctuations in computation times. Note that $r_i$ describes *local* fluctuations; it is true that $r_i \leq \sup_{j\geq 1} \bar{h}_{i,j}/\inf_{j\geq 1} \bar{h}_{i,j}$ for all $i \in [n]$ and $r_i$ can be arbitrarily smaller.

A corollary of our main result in this part (Theorem M.1) is presented next.

**Corollary 4.6.** *If the computation and communication times are positive, the time complexity of Alg. 7 is $\frac{L\Delta}{\varepsilon} \times t^*(\omega, \sigma^2/\varepsilon, [\max\{h_i, \tau_i\}, \min\{\tau_i r_i, \max\{h_i, \tau_i\}\}]_1^n)$ up to a constant factor, where $r_i$ is defined in Def. 4.5.*

Unlike Alg. 1 and Alg. 4, Alg. 7 is more "greedy"; it calculates stochastic gradients and sends compressed vectors in parallel, and it does not know the times $h_i$ and $\tau_i$ (or $h_i^k$ and $\tau_i^k$). That is why this method gets a suboptimal complexity and depends on $r_i$. Nevertheless, if we assume that i) the computation times do not fluctuate significantly, i.e., $r_i = \Theta(1)$, and ii) $\tau_i \leq h_i$ for all $i \in [n]$, then this complexity reduces to the optimal complexity $L\Delta/\varepsilon \times t^*(\omega, \sigma^2/\varepsilon, [h_i, \tau_i]_1^n)$.

# 5   Lower Bound

---
**Protocol 3** Simplified Representation of Protocol 9

---
1: Init $S = \emptyset$ on the server (all available information)
2: **while** True **do**
3:     Server calculates a new point $\bar{x}$ using $S$ and broadcasts $\bar{x}$ and $S$ to any worker
       (broadcasting does not take time)
4: **end while**
   $i^{\text{th}}$ **Worker (in parallel):**
5: **while** True **do**
6:     Receives $\bar{x}$ and $S$, calculates as many stochastic gradients as it want at the point $\bar{x}$ (each
       calculation takes $h_i$ seconds), aggregates all available information, and sends compressed
       vectors (each dispatch takes $\tau_i$ seconds), which will be added to the set $S$
7: **end while**

---

In Sec. 4, we stated that Shadowheart SGD converges after $T_*$ seconds; with $T_*$ given in (10). Our next step is to understand if it might be possible to improve this complexity. In Sec. O, we formalize our setup and show in Theorem O.5 that up to a constant factor, the result (10) is *optimal*. Here we present a simplified illustration of our approach.

Protocol 3 can describe all centralized methods (the server updates the iterates, and the workers calculate stochastic gradients at these points), including Minibatch SGD, Asynchronous SGD, Rennala SGD, and Shadowheart SGD. In Theorem O.5, we show that up to a constant factor, no method described by Protocol 3 can converge faster than (10) seconds. In order to use our lower bound, the workers must calculate stochastic gradients at a point that was calculated by the server.

Let us briefly explain the proof's idea. The general approach is the same as in (Nesterov, 2003; Arjevani et al., 2022; Huang et al., 2022): we take the "difficult" function (Sec. P.1), which has large gradients while the last coordinate equals to zero. Every algorithm starts with the point $x^0 = 0$, and the only way to discover the next coordinate is to calculate a stochastic gradient. Oracles associated with the workers return the next non-zero coordinate with the probability $p_\sigma \approx \varepsilon/\sigma^2$. Even if the stochastic oracle returns a non-zero coordinate for some worker, the corresponding communication oracle on this worker also has to return a non-zero coordinate, which happens with probability $p_\omega \approx 1/\omega+1$. We fix the $\text{Rand}K$ compressor in the lower bound theorem with $K \approx L\Delta/\varepsilon(\omega+1)$, and the number of coordinates $\approx L\Delta/\varepsilon$; thus, indeed, $p_\omega \approx 1/\omega+1$. Since all $n$ workers work in parallel, they can discover and send to the server the next non-zero coordinate not earlier than after $\min_{m \in [n]} \{h_m \eta_m + \tau_m \mu_m\}$ seconds, where $\eta_m$ and $\mu_m$ are i.i.d. *geometric* random variables with $p_\sigma$ and $p_\omega$. With a high probability, we show that this quantity is $\Omega(t^*(1/p_\omega, 1/p_\sigma, h_1, \tau_1, \ldots, h_n, \tau_n))$. The number of coordinates is $\approx L\Delta/\varepsilon$. Therefore, the lower bound is (10) seconds up to a constant factor.

# 6   Equilibrium Time

Since the time complexity (10) of Shadowheart SGD is optimal, we believe that the equilibrium time is a fundamental mapping that should be investigated more deeply.

## 6.1   Calculation strategy

The calculation of $t^*$ requires us to sort $\max\{h_i, \tau_i\}$. Next, it is sufficient to solve $n$ equations from (7). In Property 4.1, we prove that (7) has one unique solution that can be easily found, for instance, using the bisection method. Then, is it left to find the minimum in (8).

## 6.2   Intuition behind the equilibrium time $t^*$

Assuming we found an optimal $j^*$ in (8), we have $t^* = \max\{\max\{h_{\pi_{j^*}}, \tau_{\pi_{j^*}}\}, s^*(j^*)\}$. The first observation is that $t^*$ does not depend on the workers that correspond to $\max\{h_{\pi_{j^*+1}}, \tau_{\pi_{j^*+1}}\}, \ldots, \max\{h_{\pi_n}, \tau_{\pi_n}\}$. Since these values are greater or equal to $\max\{h_{\pi_{j^*}}, \tau_{\pi_{j^*}}\}$, the mapping "decides" to ignore them because they are too slow. The following

derivations are not rigorous and are merely supposed to offer some intuition. We define $\alpha_i := \tau_{\pi_i}\omega$ and $\beta_i := h_{\pi_i}\sigma^2/\varepsilon$. Next, using (7), we have

$$1 = \sum_{i=1}^{j^*} \frac{s^*(j^*)}{2\alpha_i + \frac{4\alpha_i\beta_i}{s^*(j^*)} + 2\beta_i} \approx s^*(j^*) \sum_{i\in\mathcal{M}} \frac{1}{\max\{\alpha_i,\beta_i\}} + s^*(j^*)^2 \sum_{i\notin\mathcal{M}} \frac{1}{\alpha_i\beta_i}, \qquad (12)$$

where $\mathcal{M} := \{i \in [j^*] \,:\, \max\{\alpha_i,\beta_i\} \geq \alpha_i\beta_i/s^*(j^*)\}$. Solving this, one can get that

$$s^*(j^*) \approx \left( \sum_{i\in\mathcal{M}} \frac{1}{\max\{\alpha_i,\beta_i\}} + \sqrt{\sum_{i\notin\mathcal{M}} \frac{1}{\alpha_i\beta_i}} \right)^{-1}. \qquad (13)$$

Thus, $s^*(j^*)$ divides the active workers into two groups $\mathcal{M}$ and $[j^*] \setminus \mathcal{M}$. Both groups contribute to (13) with a *harmonic mean*-like and a *quadratic harmonic mean*-like dependences, correspondingly. The transition between two groups is decided by the rule $\max\{\alpha_i,\beta_i\} \geq \alpha_i\beta_i/s^*(j^*) \Leftrightarrow s^*(j^*) \geq \min\{\alpha_i,\beta_i\}$. Intuitively, the last inequality means that if $\tau_i$ or $h_i$ is small (a worker can quickly compute a gradient or send a compressed vector), it belongs to $\mathcal{M}$. Otherwise, if a worker's computation and communication performance are balanced, it belongs to $[j^*] \setminus \mathcal{M}$.

## 6.3 Properties of the equilibrium time $t^*$

We now provide some properties and particular cases to understand $t^*$ better. One can find the proofs and more properties in Sec. E. The first result says that $t^*$ is monotonic.

**Property 6.1.** *If* $\bar{\omega} \geq \omega \geq 0, \bar{\sigma}^2/\bar{\varepsilon} \geq \sigma^2/\varepsilon \geq 0, \bar{h}_1 \geq h_1 \geq 0, \bar{\tau}_1 \geq \tau_1 \geq 0, \ldots, \bar{h}_n \geq h_n \geq 0$, *and* $\bar{\tau}_n \geq \tau_n \geq 0$, *then* $t^*(\bar{\omega}, \bar{\sigma}^2/\bar{\varepsilon}, [\bar{h}_i, \bar{\tau}_i]_1^n) \geq t^*(\omega, \sigma^2/\varepsilon, [h_i, \tau_i]_1^n)$.

Consider the Rand$K$ compressor. If it takes $\dot{\tau}_i$ sec to send *one coordinate* by worker $i$, then, up to a constant factor, Property 6.2 ensures that an optimal choice of $K$ is 1.

**Property 6.2.** *For all* $K \in [1, d], \sigma^2/\varepsilon, h_1, \dot{\tau}_1, \ldots, h_n, \dot{\tau}_n \geq 0$, *we have* $24 \times t^*\left(d/K - 1, \sigma^2/\varepsilon, h_1, K\dot{\tau}_1, \ldots, h_n, K\dot{\tau}_n\right) \geq t^*\left(d - 1, \sigma^2/\varepsilon, h_1, \dot{\tau}_1, \ldots, h_n, \dot{\tau}_n\right)$.

## 6.4 Examples

We now list several examples, starting with simple corner/extreme cases. One can find the derivations in Sec. F. For brevity, we will sometimes write $t^*$ instead of $t^*(\omega, \sigma^2/\varepsilon, h_1, \tau_1, \ldots, h_n, \tau_n)$.

*Example* 6.3. **[Infinitely Fast Worker]** If exists $j \in [n]$ such that $\tau_j = 0$ and $h_j = 0$, then $t^* = 0$.

*Example* 6.4. **[Infinitely Slow Workers]** If $\tau_i = \infty$ and $h_i = \infty$ for all $i \in [n]$, then $t^* = \infty$.

*Example* 6.5. **[Equal Performance]** If $\tau_i = \tau$ and $h_i = h$ for all $i \in [n]$, then[9]

$$t^* \leq 6 \max\left\{ h, \tau, \frac{\tau\omega}{n}, \frac{h\sigma^2}{n\varepsilon}, \sqrt{\frac{\tau h\sigma^2\omega}{n\varepsilon}} \right\}. \qquad (14)$$

In the next example, we consider the setting from Sec. 2.1. Example 6.6 and Corollary 4.3 restore the optimal rate (4) of Rennala SGD.

*Example* 6.6. **[Infinitely Fast Communication]** If $\tau_i = 0$ for all $i \in [n]$, then

$$t^* \leq 2 \min_{m\in[n]} \max\left\{ h_{\pi_m}, \frac{\sigma^2}{\varepsilon}\left(\sum_{i=1}^m \frac{1}{h_{\pi_i}}\right)^{-1} \right\} = \Theta\left( \min_{m\in[n]} \left(\frac{1}{m}\sum_{i=1}^m \frac{1}{h_{\pi_i}}\right)^{-1}\left(1 + \frac{\sigma^2}{m\varepsilon}\right) \right), \qquad (15)$$

where $\pi$ is a permutation that sorts $\{h_i\}_{i=1}^n$.

The following two examples show that $t^*$ is robust to slow workers or workers that do not participate.

*Example* 6.7. **[Ignoring Slow Workers]** If $h_i$ and $\tau_i$ are fixed and finite for all $i \leq p$, and $\max\{h_i, \tau_i\} = m \in \mathbb{R}$ for all $i > p$, then, for $m$ large enough, we have $t^*(\omega, \sigma^2/\varepsilon, [h_i, \tau_i]_1^n) = t^*(\omega, \sigma^2/\varepsilon, [h_i, \tau_i]_1^p)$.

*Example* 6.8. **[Partial Participation]** If $\max\{h_i, \tau_i\} = \infty$ for all $i > p \geq 1$, then $t^*(\omega, \sigma^2/\varepsilon, [h_i, \tau_i]_1^n) = t^*(\omega, \sigma^2/\varepsilon, [h_i, \tau_i]_1^p)$.

---

[9]From the proof, it is clear that the result is tight up to a constant factor.

# 7 Comparison with Baselines

In the previous sections, we did not invoke any assumptions about the compressors except for Def. 2.1. Inspired by Property 6.2, to make the comparisons with the baselines easier, we consider the Rand$K$ compressor with $K = 1$. Using Theorem D.2, we have $\omega = d - 1$. We also assume that worker $i$ takes $\dot{\tau}_i$ seconds to send *one coordinate* to the server; thus $\tau_i = \dot{\tau}_i$, since we use Rand1. Also, it takes $d\dot{\tau}_i$ to send a non-compressed vector for all $i \in [n]$.

**Minibatch SGD and QSGD.** It is well known (Lan, 2020) that the number of iterations of Minibatch SGD required to find an $\varepsilon$–solution is $O\left(L\Delta/\varepsilon + \sigma^2 L\Delta/\varepsilon^2\right)$. In Minibatch SGD, each worker calculates one stochastic gradient and sends a non-compressed vector. Since the server waits for the slowest worker, the time complexity of such method (up to a constant factor) is

$$T_{\mathrm{MB}} := \max_{i \in [n]} (h_i + d\dot{\tau}_i) \left(\frac{L\Delta}{\varepsilon} + \frac{\sigma^2 L\Delta}{n\varepsilon^2}\right). \tag{16}$$

In Sec. J, we compare (16) with (10) and show

*Comparison* 7.1. $T_* = O(T_{\mathrm{MB}})$.

However, there are many regimes when $T_* \ll T_{\mathrm{MB}}$. For instance, if $\max\{h_i, \dot{\tau}_i\} = \infty$ for some worker (Example 6.8), then $T_{\mathrm{MB}} = \infty$ and $T_* < \infty$. Also, under the conditions of Example 6.7, if $m \to \infty$, we get $T_{\mathrm{MB}} \to \infty$ whereas $T_*$ is bounded. The same reasoning applies to QSGD because its time complexity (6) depends on $\max_{i \in [n]} (h_i + \dot{\tau}_i)$. Due to Theorem O.5, up to a constant factor, $T_*$ is less or equal to (6); see also Table 1.

**Rennala SGD and Asynchronous SGD.** When the communication time is negligible, Tyurin and Richtárik (2023c) proved that the optimal time complexity is attained by Rennala SGD. When $\dot{\tau}_i \to 0$ for all $i \in [n]$, we show in Example 6.6 that (10) is the same as the time complexity of Rennala SGD obtained by Tyurin and Richtárik (2023c). Assume $\dot{\tau}_i > 0$ for all $i \in [n]$. We can apply the result from Theorem O.5 to Rennala SGD, thus the time complexity of Rennala SGD is not better than

$$T_{\mathrm{R}} := \frac{L\Delta}{\varepsilon} \times t^*(0, \sigma^2/\varepsilon, h_1, d\dot{\tau}_1, \ldots, h_n, d\dot{\tau}_n). \tag{17}$$

Note that Asynchronous SGD also has the same lower bound. In Sec. J, we compare (17) with (10) and show

*Comparison* 7.2. $T_* = O(T_{\mathrm{R}})$.

## 7.1 Shadowheart SGD **is strictly better in many regimes.**

Due to the non-explicit nature, it is not transparent that the time complexity of Shadowheart SGD is universally strictly better than in all baselines in many practical regimes. Let us prove it. Take

$$h_i = h \text{ for all } i < n, h_n = \infty, \text{ and } \dot{\tau}_i = \dot{\tau} \text{ for all } i \in [n]. \tag{18}$$

Due to (16) and (17), the time complexities of Minibatch SGD and QSGD are $\infty$, and the time complexities of Asynchronous SGD and Rennala SGD are not smaller than

$$\Omega\left(\max\left\{h, d\dot{\tau}, \frac{h\sigma^2}{(n-1)\varepsilon}\right\} \frac{L\Delta}{\varepsilon}\right), \text{ which } \overset{n\to\infty}{\to} \Omega\left(\max\{h, d\dot{\tau}\} \frac{L\Delta}{\varepsilon}\right).$$

From (14), the time complexity of Shadowheart SGD is at most

$$O\left(\max\left\{h, \dot{\tau}, \frac{d\dot{\tau}}{n-1}, \frac{h\sigma^2}{(n-1)\varepsilon}, \sqrt{\frac{\dot{\tau}h\sigma^2 d}{(n-1)\varepsilon}}\right\} \frac{L\Delta}{\varepsilon}\right), \text{ which } \overset{n\to\infty}{\to} O\left(\max\{h, \dot{\tau}\} \frac{L\Delta}{\varepsilon}\right).$$

Thus, Shadowheart SGD can be $d$ times faster than all previous methods if the number of workers $n$ is large. Using the same reasoning, Shadowheart SGD can be $n - 1$ times faster if the dimension $d$ or $\dot{\tau}$ is large. Due to the continuity of the complexities, such huge differences hold even if we take $n < \infty$, and start considering more heterogenous times $h_i$ and $\tau_i$ by perturbating (18).

## 7.2 The fastest worker works locally

Another important baseline is the vanilla SGD method, which works on the fastest worker, does not communicate with the server, and performs local steps (non-centralized method). For simplicity, assume that $\sigma^2/\varepsilon \geq 1$. Then, the time complexity of such an algorithm (Lan, 2020) is $T_{\mathrm{SGD}} :=$

$\min_{i \in [n]} h_i \times \sigma^2 L \Delta / \varepsilon^2$. Clearly, comparing $T_{\text{SGD}}$ and (10), if $\dot{\tau}_i$ are large enough, then $T_{\text{SGD}}$ can be smaller than $T_*$. However, this does not contradict our lower bounds because this method does not satisfy the conditions of Theorem O.5: it does not communicate with the server. In other words, if the communication channel is too slow, it does not make sense to communicate. One may now ask: "Under which conditions is it beneficial to communicate?" Comparing $T_{\text{SGD}}$ and (10), one can see that (10) is better when $t^*(d-1, \sigma^2/\varepsilon, h_1, \dot{\tau}_1, \ldots, h_n, \dot{\tau}_n) \leq \min_{i \in [n]} h_i \times \sigma^2/\varepsilon$. It is sufficient to substitute the initial parameters to this inequality and decide which method to use. For instance, in the view of Example 6.5, one should compare $\max\{h, \dot{\tau}, \dot{\tau}(d-1)/n, h\sigma^2/n\varepsilon, \sqrt{\dot{\tau}h\sigma^2(d-1)/n\varepsilon}\}$ vs. $h\sigma^2/\varepsilon$. In the regime when $n$ is large enough or $\varepsilon$ is small enough, we have $t^* < h\sigma^2/\varepsilon$, and Alg. 7 has better convergence guarantees. On the other hand, if $\dot{\tau}$ is large enough, then it is possible that $t^* > h\sigma^2/\varepsilon$.

## Acknowledgments and Disclosure of Funding

The research reported in this publication was supported by funding from King Abdullah University of Science and Technology (KAUST): i) KAUST Baseline Research Scheme, ii) Center of Excellence for Generative AI, under award number 5940, iii) SDAIA-KAUST Center of Excellence in Artificial Intelligence and Data Science. The work of A.T. was partially supported by the Analytical center under the RF Government (subsidy agreement 000000D730321P5Q0002, Grant No. 70-2021-00145 02.11.2021).

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

# Contents

# A  Bidirectional Compression

In this section, we discuss a simple way to use the Shadowheart SGD techniques in the setup when broadcasting is expensive (Line 7 in Alg. 1); i.e., when $\tau_{\text{serv}} \gg 0$. We will employ the following family of compressors.

**Definition A.1.** A mapping $\mathcal{C} : \mathbb{R}^d \times \mathbb{S}_\nu \to \mathbb{R}^d$ is a *biased compressor* if there exists $\alpha \in (0, 1]$ such that

$$\mathbb{E}_\nu \left[ \|\mathcal{C}(x; \nu) - x\|^2 \right] \leq (1 - \alpha) \|x\|^2, \ \forall x \in \mathbb{R}^d. \tag{19}$$

We shall use the shortcut $\mathcal{C}(x; \nu) \equiv \mathcal{C}(x)$, and denote the family of such biased compressors as $\mathbb{B}(\alpha)$.

The family $\mathbb{B}(\alpha)$ is more general than $\mathbb{U}(\omega)$ in the sense that if $\mathcal{C} \in \mathbb{U}(\omega)$, then $(\omega + 1)^{-1}\mathcal{C} \in \mathbb{B}((\omega + 1)^{-1})$. It includes the TopK and RankK compressors (Vogels et al., 2019; Beznosikov et al., 2020), among many others.

Let $\mathcal{C}_{\text{serv}} \in \mathbb{B}(\alpha)$ be the compressor used by the server. We use the primal error-feedback mechanism EF21-P (Gruntkowska et al., 2023) which requires us to add the following changes to Alg. 1 and Alg. 2. We add the steps

$$p^{k+1} = \mathcal{C}_{\text{serv}}(x^{k+1} - w^k), \quad w^{k+1} = w^k + p^{k+1}$$

to Alg. 1 and broadcast $p^{k+1}$ instead of $x^k$. This change leads to Bidirectional Shadowheart SGD (Alg. 5). In Alg. 2, the workers should receive $p^{k+1}$, calculate $w^{k+1}$, and use $w^k$ instead of $x^k$ in the calculations of stochastic gradients. We provide the pseudo-codes of these algorithms in Sec. K.

Our main results are:

**Theorem A.2.** *Let Assumptions 1.1, 1.2, 1.3, 2.2 hold. Choose $\gamma = \frac{\alpha}{16L}$. Then as long as $K \geq \frac{768L\Delta}{\alpha\varepsilon}$,* Bidirectional Shadowheart SGD *(Alg. 5) guarantees to find an $\varepsilon$–stationary point.*

**Corollary A.3.** *If the broadcast time of $\mathcal{C}_{\text{serv}}$ is not greater than $\tau_{\text{serv}}$, then* Bidirectional Shadowheart SGD *(Alg. 5) converges after at most*

$$T_{*,\text{serv}} := \frac{768L\Delta}{\alpha\varepsilon} \times \left( \tau_{\text{serv}} + 2t^*(\omega, \sigma^2/\varepsilon, [h_i, \tau_i]_1^n) \right) \tag{20}$$

*seconds.*

*Remark* A.4. If the broadcast cost can't be ignored, the time complexity of Alg. 1 changes from (10) to

$$T_* := \frac{16L\Delta}{\varepsilon} \times (\tau_{\text{serv}}^{\text{full}} + 2t^*(\omega, \sigma^2/\varepsilon, [h_i, \tau_i]_1^n)), \tag{21}$$

where $\tau_{\text{serv}}^{\text{full}}$ is the time required to broadcast a *full/ non-compressed* vector.

We should compare (21) obtained by the unidirectional algorithm and (20) obtained by the bidirectional algorithm. Consider that $\mathcal{C}_{\text{serv}} = \text{Top}K$ with $K \leq d$. We can see (20) that depends on $\tau_{\text{serv}}$, that is much less than $\tau_{\text{serv}}^{\text{full}}$ because $K \ll d$. At the same time, (20) is $1/\alpha$ times larger than (21). This is a standard price for the fact that we use a biased compressor (e.g. (Richtárik et al., 2021; Gruntkowska et al., 2023)). However, $\alpha$ is very close 1 in practice (Beznosikov et al., 2020; Vogels et al., 2019; Xu et al., 2021b). It turns out that we can always choose $K$ in TopK (we take this compressor as an example) in such a way that Alg. 5 is never worse than Alg. 1.

*Comparison* A.5. Assume that it takes $\dot\tau_{\text{serv}}$ seconds to send *one coordinate* from the server to the workers. If we take $K \geq \min\left\{d, t^*(\omega, \sigma^2/\varepsilon, [h_i, \tau_i]_1^n)/\dot\tau_{\text{serv}}\right\}$ in TopK, then $T_{*,\text{serv}} = \text{O}(T_*)$.

If $\tau_{\text{serv}}^{\text{full}} = d\dot\tau_{\text{serv}}$ is the bottleneck in (21) with Alg. 1, i.e., $d\dot\tau_{\text{serv}} \gg t^*$, then one can take $K = t^*/\dot\tau_{\text{serv}} \ll d$ in TopK with Alg. 5 and improve the time complexity.

# B Frequently Used Notation

We thought a table of frequently used notation could be useful. Here it is:

| Notation | Meaning |
|---|---|
| $\varepsilon$ | error tolerance |
| $f$ | Function $f : \mathbb{R}^d \to \mathbb{R}$ whose $\varepsilon$-stationary point we want to find (see (1)) |
| $L$ | Smoothness parameter of $f$ (see Assumption 1.1) |
| $f^*$ | Lower bound on $f$ (see (1.2)) |
| $\sigma^2$ | Stochastic gradients $\nabla f(x; \xi)$ have variance bounded by $\sigma^2$ (see Assumption 1.3) |
| $x^0$ | Starting point of all algorithms; a vector in $\mathbb{R}^d$ |
| $\gamma$ | Positive stepsize used by all algorithms |
| $\Delta$ | $\Delta := f(x^0) - f^*$ |
| $n$ | number of workers |
| $h_i$ | Maximal time it takes for worker $i$ to compute one stochastic gradient of $\nabla f(\cdot; \xi)$ |
| $b_i$ | Minibatch size associated with worker $i$ (worker $i$ compresses minibatch gradients) |
| $m_i$ | Number of compressed messages sent to the server by worker $i$ in a single iteration |
| $\mathbb{U}(\omega)$ | Set of unbiased compressors with variance parameter $\omega \geq 0$ (see Definition 2.1) |
| $\mathcal{C}_{ij}$ | Compressors used by worker $i$; $\mathcal{C}_{ij} \in \mathbb{U}(\omega)$, $j \in \{1, \ldots, m_i\}$ |
| $\tau_i$ | Maximal time it takes for worker $i$ to communicate vector $\mathcal{C}_{ij}(\cdot)$, where $\mathcal{C}_{ij} \in \mathbb{U}(\omega)$, to the server |
| $\dot{\tau}_i$ | Time it takes to send to worker $i$ one float to the server (equal to $\tau_i$ of the Rand1 compressor is used) |
| $\mathbb{B}(\alpha)$ | Set of biased compressors with contraction parameter $0 < \alpha \leq 1$ (see Definition A.1) |
| $\mathcal{C}_{\mathrm{serv}}$ | Compressor used by the server; $\mathcal{C}_{\mathrm{serv}} \in \mathbb{B}(\alpha)$ |
| $\tau_{\mathrm{serv}}^{\mathrm{full}}$ | Maximal time it takes for server to broadcast a non-compressed vector from $\mathbb{R}^d$ to the workers |
| $\tau_{\mathrm{serv}}$ | Maximal time it takes for server to broadcast a vector $\mathcal{C}_{\mathrm{serv}}(\cdot)$, where $\mathcal{C}_{\mathrm{serv}} \in \mathbb{B}(\alpha)$, to the workers |
| $\dot{\tau}_{\mathrm{serv}}$ | Time it takes for server to broadcast one float to the workers |
| $t^*$ | Equilibrium time; a function of $\omega, \sigma^2/\varepsilon, h_1, \tau_1, \ldots, h_n, \tau_n$ (see Definition 3.1) |
| $T_*$ | Time complexity of Shadowheart SGD (see Corollary 4.3) |
| $T_{\mathrm{MB}}$ | Time complexity of Minibatch SGD (see (16)) |
| $T_{\mathrm{R}}$ | Time complexity of Rennala SGD (see (17)) |
| $g = \mathrm{O}(f)$ | Exist $C > 0$ such that $g(z) \leq C \times f(z)$ for all $z \in \mathcal{Z}$ |
| $g = \Omega(f)$ | Exist $C > 0$ such that $g(z) \geq C \times f(z)$ for all $z \in \mathcal{Z}$ |
| $g = \Theta(f)$ | $g = \mathrm{O}(f)$ and $g = \Omega(f)$ |
| $\{a, \ldots, b\}$ | Set $\{i \in \mathbb{Z} \mid a \leq i \leq b\}$ |
| $[n]$ | $\{1, \ldots, n\}$ |

# C Basic Facts

Here we collect some basic facts which are used repeatedly in the proofs.

**Variance decomposition.** Let $x \in \mathbb{R}^d$ be a random vector with finite mean and finite variance. Then for any deterministic vector $c \in \mathbb{R}^d$, we have the identity

$$\mathbb{E}\left[\|x - \mathbb{E}[x]\|^2\right] = \mathbb{E}\left[\|x - c\|^2\right] - \|\mathbb{E}[x] - c\|^2. \tag{22}$$

**Lemma C.1.** *Consider a sequence $q_1, \ldots, q_n \in [0, 1]$, then*

$$1 - \sum_{m=1}^{n} q_m \leq \prod_{m=1}^{n} (1 - q_m).$$

*Proof.* We prove by induction. For $n = 1$, is it true: $1 - \sum_{m=1}^{1} q_m = \prod_{m=1}^{1} (1 - q_m)$. Assume that that it is true for $n - 1$. Then

$$1 - \sum_{m=1}^{n-1} q_m \leq \prod_{m=1}^{n-1} (1 - q_m).$$

Multiply both parts by $1 - q_n \in [0, 1]$ to obtain

$$\prod_{m=1}^{n} (1 - q_m) \geq (1 - q_n) \left(1 - \sum_{m=1}^{n-1} q_m\right) = 1 - \sum_{m=1}^{n-1} q_m - q_n + q_n \left(\sum_{m=1}^{n-1} q_m\right) \geq 1 - \sum_{m=1}^{n} q_m$$

since $q_m \in [0, 1]$ for all $m \in [n]$. $\qquad\qquad\square$

# D  Rand$K$ Compressor

**Definition D.1.** Assume that $S$ is a random subset from $[d]$, $|S| = K$, $K \in [d]$. A stochastic mapping $\mathcal{C} : \mathbb{R}^d \times \mathbb{S}_\nu \to \mathbb{R}^d$ is Rand$K$ if

$$\mathcal{C}(x; S) = \frac{d}{K} \sum_{j \in S} x_j e_j,$$

where $\{e_i\}_{i=1}^d$ is the standard unit basis.

**Theorem D.2.** *If $\mathcal{C}$ is RandK, then $\mathcal{C} \in \mathbb{U}\left(\frac{d}{k} - 1\right)$ .*

One can find the proof in (Beznosikov et al., 2020).

# E  Proofs of the Properties of the Equilibrium Time

**Property 4.1.** *If all inputs of the equilibrium time are non-negative, then the equilibrium time is well defined.*

*Proof.*
*(Part 1: $s^*(j)$ is well-defined)*
First, we show that $s^*(j)$ is well-defined for all $j \in [n]$. We fix $j \in [n]$ and consider the equation from Def. 3.1:

$$\underbrace{\left(\sum_{i=1}^{j} \frac{1}{2\tau_{\pi_i}\omega + \frac{4\tau_{\pi_i} h_{\pi_i}\sigma^2\omega}{s \times \varepsilon} + \frac{2h_{\pi_i}\sigma^2}{\varepsilon}}\right)^{-1}}_{\phi(s)} = \underbrace{s}_{\psi(s)} \tag{23}$$

w.r.t $s$. The function $\phi(s)$ is a **non-increasing** function for all $s \geq 0$, and the function $\psi(s)$ is an **increasing** function for all $s \geq 0$. Let us consider two cases.

1) Exists $p \leq j$ such that $\tau_{\pi_p}\omega = 0$ and $\frac{h_{\pi_p}\sigma^2}{\varepsilon} = 0$, then

$$\phi(s) = \left(\sum_{i \neq p} \frac{1}{2\tau_{\pi_i}\omega + \frac{4\tau_{\pi_i} h_{\pi_i}\sigma^2\omega}{s \times \varepsilon} + \frac{2h_{\pi_i}\sigma^2}{\varepsilon}} + \frac{1}{0}\right)^{-1} = (\infty)^{-1} = 0.$$

for all $s \geq 0$, then the only solution to the equation is $s = 0$.
2) Otherwise, we have

$$\phi(s) = \left(\sum_{i=1}^{j} \frac{1}{2\tau_{\pi_i}\omega + \frac{4\tau_{\pi_i} h_{\pi_i}\sigma^2\omega}{s \times \varepsilon} + \frac{2h_{\pi_i}\sigma^2}{\varepsilon}}\right)^{-1} \geq \left(\sum_{i=1}^{j} \frac{1}{2\tau_{\pi_i}\omega + \frac{2h_{\pi_i}\sigma^2}{\varepsilon}}\right)^{-1} > 0$$

for all $s \geq 0$. Then $\phi(0) > 0$ (can be equal to $\infty$). Using $\psi(0) = 0$ and the monotonicity of the functions, one can show the unique solution (greater zero) exists.

If a permutation $\pi$ is unique, then the formula $\min_{j \in [n]} \max\{\max\{h_{\pi_j}, \tau_{\pi_j}\}, s^*(j)\}$ is well-defined and we can finish the proof.

*(Part 2: non-unique permutation)*
We assume that there exists $i \in [n]$ such that $\max\{h_i, \tau_i\} < \infty$. Otherwise, $\min_{j \in [n]} \max\{\max\{h_{\pi_j}, \tau_{\pi_j}\}, s^*(j)\} = \infty$ for any permutation. Next, note that $s^*(j+1) \leq s^*(j)$ for all $j < n$ because

$$\left(\sum_{i=1}^{j+1} \frac{1}{2\tau_{\pi_i}\omega + \frac{4\tau_{\pi_i} h_{\pi_i}\sigma^2\omega}{s \times \varepsilon} + \frac{2h_{\pi_i}\sigma^2}{\varepsilon}}\right)^{-1} \leq \left(\sum_{i=1}^{j} \frac{1}{2\tau_{\pi_i}\omega + \frac{4\tau_{\pi_i} h_{\pi_i}\sigma^2\omega}{s \times \varepsilon} + \frac{2h_{\pi_i}\sigma^2}{\varepsilon}}\right)^{-1}$$

for all $s \geq 0$. We will use this property later.

Consider that there are two non-equal permutations $\pi$ and $\bar{\pi}$ that sort the pairs $(h_i, \tau_i)$ by $\max\{h_i, \tau_i\}$, and there are two corresponding solutions $s^*(j)$ and $\bar{s}^*(j)$. Each permutation divides the pairs $(h_i, \tau_i)$ into the same equivalence classes:

$$\underbrace{\max\{h_{\pi_1}, \tau_{\pi_1}\} = \cdots = \max\{h_{\pi_{j_1}}, \tau_{\pi_{j_1}}\}}_{C_1} < \underbrace{\max\{h_{\pi_{j_1+1}}, \tau_{\pi_{j_1+1}}\} = \cdots = \max\{h_{\pi_{j_2}}, \tau_{\pi_{j_2}}\}}_{C_2} < \ldots,$$

$$\underbrace{\max\{h_{\bar{\pi}_1}, \tau_{\bar{\pi}_1}\} = \cdots = \max\{h_{\bar{\pi}_{j_1}}, \tau_{\bar{\pi}_{j_1}}\}}_{C_1} < \underbrace{\max\{h_{\bar{\pi}_{j_1+1}}, \tau_{\bar{\pi}_{j_1+1}}\} = \cdots = \max\{h_{\bar{\pi}_{j_2}}, \tau_{\bar{\pi}_{j_2}}\}}_{C_2} < \ldots.$$

The order within each class can be different, but the elements are the same. Next, since $s^*(j+1) \leq s^*(j)$ for all $j < n$, we can conclude that the minimum in

$$\min_{j \in [n]} \max\{\max\{h_{\pi_j}, \tau_{\pi_j}\}, s^*(j)\}$$

is attained for $j^*$ such that $\max\{h_{\pi_{j^*}}, \tau_{\pi_{j^*}}\} < \max\{h_{\pi_{j^*+1}}, \tau_{\pi_{j^*+1}}\}$ $(\max\{h_{\pi_{n+1}}, \tau_{\pi_{n+1}}\} \equiv \infty)$. Since $\max\{h_{\pi_{j^*}}, \tau_{\pi_{j^*}}\} < \max\{h_{\pi_{j^*+1}}, \tau_{\pi_{j^*+1}}\}$, we have $\max\{h_{\pi_{j^*}}, \tau_{\pi_{j^*}}\} = \max\{h_{\bar{\pi}_{j^*}}, \tau_{\bar{\pi}_{j^*}}\}$. Therefore, we obtain

$$\min_{j \in [n]} \max\{\max\{h_{\pi_j}, \tau_{\pi_j}\}, s^*(j)\} = \max\{\max\{h_{\pi_{j^*}}, \tau_{\pi_{j^*}}\}, s^*(j^*)\} = \max\{\max\{h_{\bar{\pi}_{j^*}}, \tau_{\bar{\pi}_{j^*}}\}, s^*(j^*)\}.$$

Also, for all $j \in [n]$ such that $\max\{h_{\pi_j}, \tau_{\pi_j}\} < \max\{h_{\pi_{j+1}}, \tau_{\pi_{j+1}}\}$, we have

$$\left( \sum_{i=1}^{j} \frac{1}{2\tau_{\pi_i}\omega + \frac{4\tau_{\pi_i}h_{\pi_i}\sigma^2\omega}{s \times \varepsilon} + \frac{2h_{\pi_i}\sigma^2}{\varepsilon}} \right)^{-1} = \left( \sum_{i=1}^{j} \frac{1}{2\tau_{\bar{\pi}_i}\omega + \frac{4\tau_{\bar{\pi}_i}h_{\bar{\pi}_i}\sigma^2\omega}{s \times \varepsilon} + \frac{2h_{\bar{\pi}_i}\sigma^2}{\varepsilon}} \right)^{-1}$$

Therefore, we get $s^*(j^*) = \bar{s}^*(j^*)$ and

$$\min_{j \in [n]} \max\{\max\{h_{\pi_j}, \tau_{\pi_j}\}, s^*(j)\} = \max\{\max\{h_{\bar{\pi}_{j^*}}, \tau_{\bar{\pi}_{j^*}}\}, \bar{s}^*(j^*)\} \geq \min_{j \in [n]} \max\{\max\{h_{\bar{\pi}_j}, \tau_{\bar{\pi}_j}\}, \bar{s}^*(j)\}.$$

Using the same reasoning, we can show that

$$\min_{j \in [n]} \max\{\max\{h_{\bar{\pi}_j}, \tau_{\bar{\pi}_j}\}, \bar{s}^*(j)\} \geq \min_{j \in [n]} \max\{\max\{h_{\pi_j}, \tau_{\pi_j}\}, s^*(j)\}.$$

It means that $\min_{j \in [n]} \max\{\max\{h_{\bar{\pi}_j}, \tau_{\bar{\pi}_j}\}, \bar{s}^*(j)\} = \min_{j \in [n]} \max\{\max\{h_{\pi_j}, \tau_{\pi_j}\}, s^*(j)\}$, thus the final result of the mapping does not depend on a chosen permutation. □

**Property 6.1.** *If* $\bar{\omega} \geq \omega \geq 0, \bar{\sigma}^2/\bar{\varepsilon} \geq \sigma^2/\varepsilon \geq 0, \bar{h}_1 \geq h_1 \geq 0, \bar{\tau}_1 \geq \tau_1 \geq 0, \ldots, \bar{h}_n \geq h_n \geq 0,$ *and* $\bar{\tau}_n \geq \tau_n \geq 0,$ *then* $t^*(\bar{\omega}, \bar{\sigma}^2/\bar{\varepsilon}, [\bar{h}_i, \bar{\tau}_i]_1^n) \geq t^*(\omega, \sigma^2/\varepsilon, [h_i, \tau_i]_1^n).$

*Proof.* Assume that $\pi$ is a permutation that sorts the pairs $(h_i, \tau_i)$ by $\max\{h_i, \tau_i\}$, and $\bar{\pi}$ is a permutation that sorts the pairs $(\bar{h}_i, \bar{\tau}_i)$ by $\max\{\bar{h}_i, \bar{\tau}_i\}$, then

$$\left( \sum_{i=1}^{j} \frac{1}{2\tau_{\pi_i}\omega + \frac{4\tau_{\pi_i}h_{\pi_i}\sigma^2\omega}{s\varepsilon} + \frac{2h_{\pi_i}\sigma^2}{\varepsilon}} \right)^{-1} \leq \left( \sum_{i=1}^{j} \frac{1}{2\bar{\tau}_{\bar{\pi}_i}\bar{\omega} + \frac{4\bar{\tau}_{\bar{\pi}_i}\bar{h}_{\bar{\pi}_i}\bar{\sigma}^2\bar{\omega}}{s\bar{\varepsilon}} + \frac{2\bar{h}_{\bar{\pi}_i}\bar{\sigma}^2}{\bar{\varepsilon}}} \right)^{-1}$$

for all $j \in [n]$. It means $\bar{s}^*(j) \geq s^*(j)$, where $s^*(j)$ and $\bar{s}^*(j)$ are the solutions of the equation (7) with the pairs $(h_i, \tau_i)$ and $(\bar{h}_i, \bar{\tau}_i)$ and corresponding permutations $\pi$ and $\bar{\pi}$. Also, we have $\max\{h_{\pi_j}, \tau_{\pi_j}\} \leq \max\{\bar{h}_{\bar{\pi}_j}, \bar{\tau}_{\bar{\pi}_j}\}$ for all $j \in [n]$. Therefore, we have

$$t^*(\omega, \sigma^2/\varepsilon, h_1, \tau_1, \ldots, h_n, \tau_n) = \min_{j \in [n]} \max\{\max\{h_{\pi_j}, \tau_{\pi_j}\}, s^*(j)\} \leq \min_{j \in [n]} \max\{\max\{\bar{h}_{\bar{\pi}_j}, \bar{\tau}_{\bar{\pi}_j}\}, \bar{s}^*(j)\}$$

$$= t^*(\bar{\omega}, \bar{\sigma}^2/\bar{\varepsilon}, \bar{h}_1, \bar{\tau}_1, \ldots, \bar{h}_n, \bar{\tau}_n).$$

□

**Property E.1.** *For all* $c \in (0, 1]$ *and* $\omega, \sigma^2/\varepsilon, h_1, \tau_1, \ldots, h_n, \tau_n \geq 0,$ *we have*

$$t^*(c \times \omega, c \times \sigma^2/\varepsilon, h_1, \tau_1, \ldots, h_n, \tau_n) \geq c \times t^*(\omega, \sigma^2/\varepsilon, h_1, \tau_1, \ldots, h_n, \tau_n).$$

*Proof.* Using the definition of the equilibrium time, we have

$$t^*(\omega, \sigma^2/\varepsilon, h_1, \tau_1, \ldots, h_n, \tau_n) = \min_{j \in [n]} \max\{\max\{h_{\pi_j}, \tau_{\pi_j}\}, s^*(j)\},$$

where $s^*(j)$ is the solution of

$$\left(\sum_{i=1}^{j} \frac{1}{2\tau_{\pi_i}\omega + \frac{4\tau_{\pi_i}h_{\pi_i}\sigma^2\omega}{s\varepsilon} + \frac{2h_{\pi_i}\sigma^2}{\varepsilon}}\right)^{-1} = s, \tag{24}$$

and

$$t^*(c \times \omega, c \times \sigma^2/\varepsilon, h_1, \tau_1, \ldots, h_n, \tau_n) = \min_{j \in [n]} \max\{\max\{h_{\pi_j}, \tau_{\pi_j}\}, s_c^*(j)\},$$

where $s_c^*(j)$ is the solution of

$$\left(\sum_{i=1}^{j} \frac{1}{2c\tau_{\pi_i}\omega + \frac{4c^2\tau_{\pi_i}h_{\pi_i}\sigma^2\omega}{s\varepsilon} + \frac{2ch_{\pi_i}\sigma^2}{\varepsilon}}\right)^{-1} = s.$$

Using simple algebra, we obtain

$$\left(\sum_{i=1}^{j} \frac{1}{2c\tau_{\pi_i}\omega + \frac{4c^2\tau_{\pi_i}h_{\pi_i}\sigma^2\omega}{s\varepsilon} + \frac{2ch_{\pi_i}\sigma^2}{\varepsilon}}\right)^{-1} = c\left(\sum_{i=1}^{j} \frac{1}{2\tau_{\pi_i}\omega + \frac{4\tau_{\pi_i}h_{\pi_i}\sigma^2\omega}{\frac{s}{c}\varepsilon} + \frac{2h_{\pi_i}\sigma^2}{\varepsilon}}\right)^{-1}.$$

Thus, $s_c^*(j)$ is the solution of

$$\left(\sum_{i=1}^{j} \frac{1}{2\tau_{\pi_i}\omega + \frac{4\tau_{\pi_i}h_{\pi_i}\sigma^2\omega}{\frac{s}{c}\varepsilon} + \frac{2h_{\pi_i}\sigma^2}{\varepsilon}}\right)^{-1} = \frac{s}{c} \tag{25}$$

Comparing (24) and (25), one can see that $s_c^*(j) = c \times s^*(j)$ for all $j \in [n]$. Using this and $c \in (0, 1]$, we get

$$t^*(c \times \omega, c \times \sigma^2/\varepsilon, h_1, \tau_1, \ldots, h_n, \tau_n) = \min_{j \in [n]} \max\{\max\{h_{\pi_j}, \tau_{\pi_j}\}, c \times s^*(j)\}$$

$$\geq c \times \min_{j \in [n]} \max\{\max\{h_{\pi_j}, \tau_{\pi_j}\}, s^*(j)\} = c \times t^*(\omega, \sigma^2/\varepsilon, h_1, \tau_1, \ldots, h_n, \tau_n).$$

$\square$

**Property E.2.** *For all $c \geq 1$ and $\omega, \sigma^2/\varepsilon, h_1, \tau_1, \ldots, h_n, \tau_n \geq 0$, we have*

$$t^*(c \times \omega, c \times \sigma^2/\varepsilon, h_1, \tau_1, \ldots, h_n, \tau_n) \leq c \times t^*(\omega, \sigma^2/\varepsilon, h_1, \tau_1, \ldots, h_n, \tau_n).$$

*Proof.* The proof of this property repeats the proof of Property E.1 up to the last inequality. Using $c \geq 1$, we get

$$t^*(c \times \omega, c \times \sigma^2/\varepsilon, h_1, \tau_1, \ldots, h_n, \tau_n) = \min_{j \in [n]} \max\{\max\{h_{\pi_j}, \tau_{\pi_j}\}, c \times s^*(j)\}$$

$$\leq c \times \min_{j \in [n]} \max\{\max\{h_{\pi_j}, \tau_{\pi_j}\}, s^*(j)\} = c \times t^*(\omega, \sigma^2/\varepsilon, h_1, \tau_1, \ldots, h_n, \tau_n).$$

$\square$

**Property E.3.** *For all $c \in (0, 1]$ and $\omega, \sigma^2/\varepsilon, h_1, \tau_1, \ldots, h_n, \tau_n \geq 0$, we have*
$t^*(c \times \omega, \sigma^2/\varepsilon, [h_i, \tau_i]_1^n) \geq c \times t^*(\omega, \sigma^2/\varepsilon, [h_i, \tau_i]_1^n)$ *and* $t^*(\omega, c \times \sigma^2/\varepsilon, [h_i, \tau_i]_1^n) \geq c \times t^*(\omega, \sigma^2/\varepsilon, [h_i, \tau_i]_1^n)$.
*For all $c \geq 1$ and $\omega, \sigma^2/\varepsilon, h_1, \tau_1, \ldots, h_n, \tau_n \geq 0$, we have $t^*(c \times \omega, \sigma^2/\varepsilon, [h_i, \tau_i]_1^n) \leq c \times t^*(\omega, \sigma^2/\varepsilon, [h_i, \tau_i]_1^n)$ and $t^*(\omega, c \times \sigma^2/\varepsilon, [h_i, \tau_i]_1^n) \leq c \times t^*(\omega, \sigma^2/\varepsilon, [h_i, \tau_i]_1^n)$*

*Remark* E.4. We can obtain stronger inequalities. See Properties E.1 and E.2.

*Proof.* For all $c \in (0, 1]$, using Properties 6.1 and E.1, we have

$$t^*(c \times \omega, \sigma^2/\varepsilon, h_1, \tau_1, \ldots, h_n, \tau_n) \geq t^*(c \times \omega, c \times \sigma^2/\varepsilon, h_1, \tau_1, \ldots, h_n, \tau_n) \geq c \times t^*(\omega, \sigma^2/\varepsilon, h_1, \tau_1, \ldots, h_n, \tau_n)$$

and

$$t^*(\omega, c \times \sigma^2/\varepsilon, h_1, \tau_1, \ldots, h_n, \tau_n) \geq t^*(c \times \omega, c \times \sigma^2/\varepsilon, h_1, \tau_1, \ldots, h_n, \tau_n) \geq c \times t^*(\omega, \sigma^2/\varepsilon, h_1, \tau_1, \ldots, h_n, \tau_n).$$

For all $c \geq 1$, using Properties 6.1 and E.2, we have

$$t^*(c \times \omega, \sigma^2/\varepsilon, h_1, \tau_1, \ldots, h_n, \tau_n) \leq t^*(c \times \omega, c \times \sigma^2/\varepsilon, h_1, \tau_1, \ldots, h_n, \tau_n) \leq c \times t^*(\omega, \sigma^2/\varepsilon, h_1, \tau_1, \ldots, h_n, \tau_n)$$

and

$$t^*(\omega, c \times \sigma^2/\varepsilon, h_1, \tau_1, \ldots, h_n, \tau_n) \leq t^*(c \times \omega, c \times \sigma^2/\varepsilon, h_1, \tau_1, \ldots, h_n, \tau_n) \leq c \times t^*(\omega, \sigma^2/\varepsilon, h_1, \tau_1, \ldots, h_n, \tau_n).$$

$\square$

**Property E.5.** *For all $c \geq 0$ and $\omega, \sigma^2/\varepsilon, h_1, \tau_1, \ldots, h_n, \tau_n \geq 0$, we have*

$$t^*(\omega, \sigma^2/\varepsilon, c \times h_1, c \times \tau_1, \ldots, c \times h_n, c \times \tau_n) = c \times t^*(\omega, \sigma^2/\varepsilon, h_1, \tau_1, \ldots, h_n, \tau_n).$$

*Proof.* For $c = 0$, it it clear. Assume that $c > 0$. Using the definition of the equilibrium time, we have

$$t^*(\omega, \sigma^2/\varepsilon, h_1, \tau_1, \ldots, h_n, \tau_n) = \min_{j \in [n]} \max\{\max\{h_{\pi_j}, \tau_{\pi_j}\}, s^*(j)\},$$

where $s^*(j)$ is the solution of

$$\left( \sum_{i=1}^{j} \frac{1}{2\tau_{\pi_i}\omega + \frac{4\tau_{\pi_i}h_{\pi_i}\sigma^2\omega}{s\varepsilon} + \frac{2h_{\pi_i}\sigma^2}{\varepsilon}} \right)^{-1} = s, \tag{26}$$

and

$$t^*(\omega, \sigma^2/\varepsilon, c \times h_1, c \times \tau_1, \ldots, c \times h_n, c \times \tau_n) = \min_{j \in [n]} \max\{\max\{c \times h_{\pi_j}, c \times \tau_{\pi_j}\}, s_c^*(j)\},$$

where $s_c^*(j)$ is the solution of

$$\left( \sum_{i=1}^{j} \frac{1}{2c\tau_{\pi_i}\omega + \frac{4c^2\tau_{\pi_i}h_{\pi_i}\sigma^2\omega}{s\varepsilon} + \frac{2ch_{\pi_i}\sigma^2}{\varepsilon}} \right)^{-1} = s.$$

For both cases, we can take the same permutation $\pi$. Using simple algebra, we obtain

$$\left( \sum_{i=1}^{j} \frac{1}{2c\tau_{\pi_i}\omega + \frac{4c^2\tau_{\pi_i}h_{\pi_i}\sigma^2\omega}{s\varepsilon} + \frac{2ch_{\pi_i}\sigma^2}{\varepsilon}} \right)^{-1} = c \left( \sum_{i=1}^{j} \frac{1}{2\tau_{\pi_i}\omega + \frac{4\tau_{\pi_i}h_{\pi_i}\sigma^2\omega}{\frac{s}{c}\varepsilon} + \frac{2h_{\pi_i}\sigma^2}{\varepsilon}} \right)^{-1}.$$

Thus, $s_c^*(j)$ is the solution of

$$\left( \sum_{i=1}^{j} \frac{1}{2\tau_{\pi_i}\omega + \frac{4\tau_{\pi_i}h_{\pi_i}\sigma^2\omega}{\frac{s}{c}\varepsilon} + \frac{2h_{\pi_i}\sigma^2}{\varepsilon}} \right)^{-1} = \frac{s}{c} \tag{27}$$

Comparing (26) and (27), one can see that $s_c^*(j) = c \times s^*(j)$ for all $j \in [n]$. Using this, we get

$$t^*(\omega, \sigma^2/\varepsilon, c \times h_1, c \times \tau_1, \ldots, c \times h_n, c \times \tau_n) = \min_{j \in [n]} \max\{\max\{c \times h_{\pi_j}, c \times \tau_{\pi_j}\}, c \times s^*(j)\}$$

$$= c \times \min_{j \in [n]} \max\{\max\{h_{\pi_j}, \tau_{\pi_j}\}, s^*(j)\} = c \times t^*(\omega, \sigma^2/\varepsilon, h_1, \tau_1, \ldots, h_n, \tau_n).$$

$\square$

**Property E.6.** *We fix a nonempty subset $S = \{k_1, \ldots, k_m\}$ from the set $[n]$ with a size $m \geq 1$. For all $\omega, \sigma^2/\varepsilon, h_1, \tau_1, \ldots, h_n, \tau_n \geq 0$, we have*

$$t^*(\omega, \sigma^2/\varepsilon, h_1, \tau_1, \ldots, h_n, \tau_n) \leq t^*(\omega, \sigma^2/\varepsilon, h_{k_1}, \tau_{k_1}, \ldots, h_{k_m}, \tau_{k_m}).$$

*Proof.* Using Property 6.1 with $\bar{\tau}_i = \infty$ and $\bar{h}_i = \infty$ for all $i \notin S$ and $\bar{\tau}_i = \tau_i$ and $\bar{h}_i = h_i$ for all $i \in S$, we have

$$t^*(\omega, \sigma^2/\varepsilon, h_1, \tau_1, \ldots, h_n, \tau_n) \leq t^*(\omega, \sigma^2/\varepsilon, \bar{h}_1, \bar{\tau}_1, \ldots, \bar{h}_n, \bar{\tau}_n).$$

Next, using Def. 3.1, we obtain

$$t^*(\omega, \sigma^2/\varepsilon, \bar{h}_1, \bar{\tau}_1, \ldots, \bar{h}_n, \bar{\tau}_n) = \min_{j \in [n]} \max\{\max\{\bar{h}_{\pi_j}, \bar{\tau}_{\pi_j}\}, s^*(j)\},$$

where $s^*(j)$ is the solution of

$$\left( \sum_{i=1}^{j} \frac{1}{2\bar{\tau}_{\pi_i}\omega + \frac{4\bar{\tau}_{\pi_i}\bar{h}_{\pi_i}\sigma^2\omega}{s \times \varepsilon} + \frac{2\bar{h}_{\pi_i}\sigma^2}{\varepsilon}} \right)^{-1} = s \tag{28}$$

w.r.t $s$ for all $j \in [n]$, and $\pi$ is a permutation that sorts $\max\{\bar{h}_i, \bar{\tau}_i\}$ in such a way that the set $\{\pi_1, \ldots, \pi_m\}$ equals to the set $\{k_1, \ldots, k_m\}$ (the order of elements can be different). Such permutation exists because $\max\{\bar{h}_i, \bar{\tau}_i\} = \infty$ for all $i \notin S$. Using $\max\{\bar{h}_{\pi_i}, \bar{\tau}_{\pi_i}\} = \infty$ for all $i > m$, we have

$$t^*(\omega, \sigma^2/\varepsilon, \bar{h}_1, \bar{\tau}_1, \ldots, \bar{h}_n, \bar{\tau}_n) = \min_{j \in [m]} \max\{\max\{\bar{h}_{\pi_j}, \bar{\tau}_{\pi_j}\}, s^*(j)\}. \tag{29}$$

By the construction of $\pi$, (28) and (29) depend only on the elements from $S$. Thus, we have

$$t^*(\omega, \sigma^2/\varepsilon, h_1, \tau_1, \ldots, h_n, \tau_n) \leq t^*(\omega, \sigma^2/\varepsilon, \bar{h}_1, \bar{\tau}_1, \ldots, \bar{h}_n, \bar{\tau}_n) = t^*(\omega, \sigma^2/\varepsilon, \bar{h}_{k_1}, \bar{\tau}_{k_1}, \ldots, \bar{h}_{k_m}, \bar{\tau}_{k_m})$$
$$= t^*(\omega, \sigma^2/\varepsilon, h_{k_1}, \tau_{k_1}, \ldots, h_{k_m}, \tau_{k_m}).$$

$\square$

**Property E.7.** *For all $\sigma^2/\varepsilon, h_1, \dot{\tau}_1, \ldots, h_n, \dot{\tau}_n \geq 0$, we have*

$$12t^*\left(0, \sigma^2/\varepsilon, h_1, d\dot{\tau}_1, \ldots, h_n, d\dot{\tau}_n\right)$$
$$\geq t^*\left(d - 1, \sigma^2/\varepsilon, h_1, \dot{\tau}_1, \ldots, h_n, \dot{\tau}_n\right).$$

*Proof.* For $d = 1$, it is clear. Assume that $d > 1$. Using the definition of $t^*$, we have

$$t^*\left(0, \sigma^2/\varepsilon, h_1, d\dot{\tau}_1, \ldots, h_n, d\dot{\tau}_n\right) \geq \min_{j \in [n]} \max \left\{ \max\{h_{\bar{\pi}_j}, d\dot{\tau}_{\bar{\pi}_j}\}, \frac{\sigma^2}{\varepsilon}\left(\sum_{i=1}^{j} \frac{1}{h_{\bar{\pi}_i}}\right)^{-1} \right\} \tag{30}$$

where $\bar{\pi}$ is a permutation that sorts $\max\{h_i, d\dot{\tau}_i\}$. Assume that $j^*$ is the minimal index that minimizes (30). Then

$$t^*\left(0, \sigma^2/\varepsilon, h_1, d\dot{\tau}_1, \ldots, h_n, d\dot{\tau}_n\right) \geq \max \left\{ \max\{h_{\bar{\pi}_{j^*}}, d\dot{\tau}_{\bar{\pi}_{j^*}}\}, \frac{\sigma^2}{\varepsilon}\left(\sum_{i=1}^{j^*} \frac{1}{h_{\bar{\pi}_i}}\right)^{-1} \right\}. \tag{31}$$

Let us define

$$I_* := t^*(d - 1, \sigma^2/\varepsilon, h_1, \dot{\tau}_1, \ldots, h_n, \dot{\tau}_n).$$

Using Property E.6, we have

$$I_* \leq t^*(d - 1, \sigma^2/\varepsilon, h_{\bar{\pi}_1}, \dot{\tau}_{\bar{\pi}_1}, \ldots, h_{\bar{\pi}_{j^*}}, \dot{\tau}_{\bar{\pi}_{j^*}}).$$

Using Def. 3.1 of $t^*$, we get

$$I_* \leq \max\{ \max_{j \in [j^*]} \max\{h_{\bar{\pi}_j}, \dot{\tau}_{\bar{\pi}_j}\}, s^* \}, \tag{32}$$

where $s^*$ is the solution of

$$\left( \sum_{i=1}^{j^*} \frac{1}{2\dot{\tau}_{\bar{\pi}_i}(d-1) + \frac{4\dot{\tau}_{\bar{\pi}_i}h_{\bar{\pi}_i}\sigma^2(d-1)}{s \times \varepsilon} + \frac{2h_{\bar{\pi}_i}\sigma^2}{\varepsilon}} \right)^{-1} = s. \tag{33}$$

Let us take $s' = 12\max\left\{(d-1)\max_{j\in[j^*]}\dot{\tau}_{\bar\pi_j}, \frac{\sigma^2}{\varepsilon}\left(\sum_{i=1}^{j^*}\frac{1}{h_{\bar\pi_i}}\right)^{-1}\right\}$. Since $s' \geq (d-1)\max_{j\in[j^*]}\dot{\tau}_{\bar\pi_j}$, we have

$$
\left(\sum_{i=1}^{j^*}\frac{1}{2\dot{\tau}_{\bar\pi_i}(d-1)+\frac{4\dot{\tau}_{\bar\pi_i}h_{\bar\pi_i}\sigma^2(d-1)}{s'\times\varepsilon}+\frac{2h_{\bar\pi_i}\sigma^2}{\varepsilon}}\right)^{-1} \leq \left(\sum_{i=1}^{j^*}\frac{1}{2\dot{\tau}_{\bar\pi_i}(d-1)+\frac{4h_{\bar\pi_i}\sigma^2}{\varepsilon}+\frac{2h_{\bar\pi_i}\sigma^2}{\varepsilon}}\right)^{-1}
$$

$$
\leq \left(\sum_{i=1}^{j^*}\frac{1}{2\dot{\tau}_{\bar\pi_i}(d-1)+\frac{6h_{\bar\pi_i}\sigma^2}{\varepsilon}}\right)^{-1}
$$

$$
\leq 12\left(\sum_{i=1}^{j^*}\min\left\{\frac{1}{\dot{\tau}_{\bar\pi_i}(d-1)},\frac{1}{\frac{h_{\bar\pi_i}\sigma^2}{\varepsilon}}\right\}\right)^{-1}.
$$

If there exists $p\in[j^*]$ such that $\frac{1}{\dot{\tau}_{\bar\pi_p}(d-1)} < \frac{1}{\frac{h_{\bar\pi_p}\sigma^2}{\varepsilon}}$, then

$$
\sum_{i=1}^{j^*}\min\left\{\frac{1}{\dot{\tau}_{\bar\pi_i}(d-1)},\frac{1}{\frac{h_{\bar\pi_i}\sigma^2}{\varepsilon}}\right\} \geq \frac{1}{\dot{\tau}_{\bar\pi_p}(d-1)}
$$

and

$$
\left(\sum_{i=1}^{j^*}\frac{1}{2\dot{\tau}_{\bar\pi_i}(d-1)+\frac{4\dot{\tau}_{\bar\pi_i}h_{\bar\pi_i}\sigma^2(d-1)}{s'\times\varepsilon}+\frac{2h_{\bar\pi_i}\sigma^2}{\varepsilon}}\right)^{-1} \leq 12(d-1)\dot{\tau}_{\bar\pi_p} \leq 12(d-1)\max_{j\in[j^*]}\dot{\tau}_{\bar\pi_j}.
$$

Otherwise, we have

$$
\sum_{i=1}^{j^*}\min\left\{\frac{1}{\dot{\tau}_{\bar\pi_i}(d-1)},\frac{1}{\frac{h_{\bar\pi_i}\sigma^2}{\varepsilon}}\right\} = \sum_{i=1}^{j^*}\frac{1}{\frac{h_{\bar\pi_i}\sigma^2}{\varepsilon}}
$$

and

$$
\left(\sum_{i=1}^{j^*}\frac{1}{2\dot{\tau}_{\bar\pi_i}(d-1)+\frac{4\dot{\tau}_{\bar\pi_i}h_{\bar\pi_i}\sigma^2(d-1)}{s'\times\varepsilon}+\frac{2h_{\bar\pi_i}\sigma^2}{\varepsilon}}\right)^{-1} \leq 12\left(\sum_{i=1}^{j^*}\frac{1}{\frac{h_{\bar\pi_i}\sigma^2}{\varepsilon}}\right)^{-1} = 12\frac{\sigma^2}{\varepsilon}\left(\sum_{i=1}^{j^*}\frac{1}{h_{\bar\pi_i}}\right)^{-1}.
$$

Considering both cases, we have

$$
\left(\sum_{i=1}^{j^*}\frac{1}{2\dot{\tau}_{\bar\pi_i}(d-1)+\frac{4\dot{\tau}_{\bar\pi_i}h_{\bar\pi_i}\sigma^2(d-1)}{s'\times\varepsilon}+\frac{2h_{\bar\pi_i}\sigma^2}{\varepsilon}}\right)^{-1} \leq 12\max\left\{(d-1)\max_{j\in[j^*]}\dot{\tau}_{\bar\pi_j},\frac{\sigma^2}{\varepsilon}\left(\sum_{i=1}^{j^*}\frac{1}{h_{\bar\pi_i}}\right)^{-1}\right\} = s'.
$$

It means that $s^* \leq s'$ because $s^*$ is the solution of (33). Using (32), we get

$$
I_* \leq 12\max\left\{\max_{j\in[j^*]}\max\{h_{\bar\pi_j},\dot{\tau}_{\bar\pi_j}\},\max\left\{(d-1)\max_{j\in[j^*]}\dot{\tau}_{\bar\pi_j},\frac{\sigma^2}{\varepsilon}\left(\sum_{i=1}^{j^*}\frac{1}{h_{\bar\pi_i}}\right)^{-1}\right\}\right\}.
$$

Using $d \geq 1$ and $d\dot{\tau}_{\bar\pi_j} \leq \max\{h_{\bar\pi_j},d\dot{\tau}_{\bar\pi_j}\}$, we get

$$
I_* \leq 12\max\left\{\max_{j\in[j^*]}\max\{h_{\bar\pi_j},d\dot{\tau}_{\bar\pi_j}\},\max\left\{\max_{j\in[j^*]}\max\{h_{\bar\pi_j},d\dot{\tau}_{\bar\pi_j}\},\frac{\sigma^2}{\varepsilon}\left(\sum_{i=1}^{j^*}\frac{1}{h_{\bar\pi_i}}\right)^{-1}\right\}\right\}
$$

$$
\leq 12\max\left\{\max_{j\in[j^*]}\max\{h_{\bar\pi_j},d\dot{\tau}_{\bar\pi_j}\},\frac{\sigma^2}{\varepsilon}\left(\sum_{i=1}^{j^*}\frac{1}{h_{\bar\pi_i}}\right)^{-1}\right\}.
$$

Due to $\max_{j\in[j^*]}\max\{h_{\bar\pi_j}, d\dot\tau_{\bar\pi_j}\} = \max\{h_{\bar\pi_{j^*}}, d\dot\tau_{\bar\pi_{j^*}}\}$ and (31), we obtain

$$I_* = t^*(d-1, \sigma^2/\varepsilon, h_1, \dot\tau_1, \ldots, h_n, \dot\tau_n) \leq 12\max\left\{\max\{h_{\bar\pi_{j^*}}, d\dot\tau_{\bar\pi_{j^*}}\}, \frac{\sigma^2}{\varepsilon}\left(\sum_{i=1}^{j^*}\frac{1}{h_{\bar\pi_i}}\right)^{-1}\right\}$$

$$\leq 12t^*\left(0, \sigma^2/\varepsilon, h_1, d\dot\tau_1, \ldots, h_n, d\dot\tau_n\right).$$

$\square$

**Property 6.2.** *For all* $K \in [1, d], \sigma^2/\varepsilon, h_1, \dot\tau_1, \ldots, h_n, \dot\tau_n \geq 0,$ *we have* $24 \times t^*\left(d/K - 1, \sigma^2/\varepsilon, h_1, K\dot\tau_1, \ldots, h_n, K\dot\tau_n\right) \geq t^*\left(d-1, \sigma^2/\varepsilon, h_1, \dot\tau_1, \ldots, h_n, \dot\tau_n\right).$

*Proof.*
(Part 1: $K \leq \frac{d+1}{2}$)
For all $K \leq \frac{d+1}{2}$, we have

$$t^*\left(\frac{d}{K} - 1, \sigma^2/\varepsilon, h_1, K\dot\tau_1, \ldots, h_n, K\dot\tau_n\right) = \min_{j\in[n]}\max\{\max\{h_{\pi_j}, K\dot\tau_{\pi_j}\}, s^*(j)\}, \qquad (34)$$

where $s^*(j)$ is the solution of

$$\left(\sum_{i=1}^{j}\frac{1}{2K\dot\tau_{\pi_i}\left(\frac{d}{K} - 1\right) + \frac{4K\dot\tau_{\pi_i}h_{\pi_i}\sigma^2\left(\frac{d}{K} - 1\right)}{s\times\varepsilon} + \frac{2h_{\pi_i}\sigma^2}{\varepsilon}}\right)^{-1} = s,$$

and $\pi$ is a permutation that sorts $\max\{h_j, K\dot\tau_j\}$. Also, assume that $j^*$ is a minimizer in (34). For all $j \in [n]$, we get

$$s^*(j) = \left(\sum_{i=1}^{j}\frac{1}{2K\dot\tau_{\pi_i}\left(\frac{d}{K} - 1\right) + \frac{4K\dot\tau_{\pi_i}h_{\pi_i}\sigma^2\left(\frac{d}{K} - 1\right)}{s^*(j)\times\varepsilon} + \frac{2h_{\pi_i}\sigma^2}{\varepsilon}}\right)^{-1}$$

$$= \left(\sum_{i=1}^{j}\frac{1}{2\dot\tau_{\pi_i}(d - K) + \frac{4\dot\tau_{\pi_i}h_{\pi_i}\sigma^2(d-K)}{s^*(j)\times\varepsilon} + \frac{2h_{\pi_i}\sigma^2}{\varepsilon}}\right)^{-1}.$$

Since $K \leq \frac{d+1}{2}$, we have

$$s^*(j) \geq \frac{1}{2}\left(\sum_{i=1}^{j}\frac{1}{2\dot\tau_{\pi_i}(d - 1) + \frac{4\dot\tau_{\pi_i}h_{\pi_i}\sigma^2(d-1)}{s^*(j)\times\varepsilon} + \frac{2h_{\pi_i}\sigma^2}{\varepsilon}}\right)^{-1}$$

and

$$2 \times s^*(j) \geq \left(\sum_{i=1}^{j}\frac{1}{2\dot\tau_{\pi_i}(d - 1) + \frac{4\dot\tau_{\pi_i}h_{\pi_i}\sigma^2(d-1)}{2\times s^*(j)\times\varepsilon} + \frac{2h_{\pi_i}\sigma^2}{\varepsilon}}\right)^{-1}. \qquad (35)$$

At the same time, using Property E.6, we have

$$t^*\left(d-1, \sigma^2/\varepsilon, h_1, \dot\tau_1, \ldots, h_n, \dot\tau_n\right) \leq t^*\left(d-1, \sigma^2/\varepsilon, h_{\pi_1}, \dot\tau_{\pi_1}, \ldots, h_{\pi_{j^*}}, \dot\tau_{\pi_{j^*}}\right) \leq \max\{\max_{j\in[j^*]}\max\{h_{\bar\pi_j}, \dot\tau_{\bar\pi_j}\}, s'(j^*)\}, \qquad (36)$$

where $s'(j^*)$ is the solution of

$$\left(\sum_{i=1}^{j^*}\frac{1}{2\dot\tau_{\pi_i}(d - 1) + \frac{4\dot\tau_{\pi_i}h_{\pi_i}\sigma^2(d-1)}{s\times\varepsilon} + \frac{2h_{\pi_i}\sigma^2}{\varepsilon}}\right)^{-1} = s.$$

From (35), we can conclude that $2 \times s^*(j^*) \geq s'(j^*)$. Using this and (36), we obtain

$$t^*\left(d-1, \sigma^2/\varepsilon, h_1, \dot\tau_1, \ldots, h_n, \dot\tau_n\right) \leq \max\{\max_{j\in[j^*]}\max\{h_{\bar\pi_j}, \dot\tau_{\bar\pi_j}\}, 2s^*(j^*)\}$$

$$\leq 2\max\{\max_{j\in[j^*]}\max\{h_{\bar{\pi}_j}, \dot{\tau}_{\bar{\pi}_j}\}, s^*(j^*)\}$$

$$\leq 2\max\{\max_{j\in[j^*]}\max\{h_{\bar{\pi}_j}, K\dot{\tau}_{\bar{\pi}_j}\}, s^*(j^*)\}.$$

Note that $\max_{j\in[j^*]}\max\{h_{\bar{\pi}_j}, K\dot{\tau}_{\bar{\pi}_j}\} = \max\{h_{\bar{\pi}_{j^*}}, K\dot{\tau}_{\bar{\pi}_{j^*}}\}$, thus

$$t^*\left(d-1, \sigma^2/\varepsilon, h_1, \dot{\tau}_1, \ldots, h_n, \dot{\tau}_n\right) \leq 2\max\{\max\{h_{\bar{\pi}_{j^*}}, K\dot{\tau}_{\bar{\pi}_{j^*}}\}, s^*(j^*)\} = 2t^*\left(\frac{d}{K}-1, \sigma^2/\varepsilon, h_1, K\dot{\tau}_1, \ldots, h_n, K\dot{\tau}_n\right).$$

(Part 2: $K > \frac{d+1}{2}$)
For all $K > \frac{d+1}{2}$, using Property 6.1, we get

$$t^*\left(\frac{d}{K}-1, \sigma^2/\varepsilon, h_1, K\dot{\tau}_1, \ldots, h_n, K\dot{\tau}_n\right) \geq t^*\left(0, \sigma^2/\varepsilon, \frac{1}{2}h_1, \frac{d}{2}\dot{\tau}_1, \ldots, \frac{1}{2}h_n, \frac{d}{2}\dot{\tau}_n\right).$$

Next, using Property E.5, we have

$$t^*\left(\frac{d}{K}-1, \sigma^2/\varepsilon, h_1, K\dot{\tau}_1, \ldots, h_n, K\dot{\tau}_n\right) \geq \frac{1}{2}t^*\left(0, \sigma^2/\varepsilon, h_1, d\dot{\tau}_1, \ldots, h_n, d\dot{\tau}_n\right).$$

It is left to use Property E.7 to get

$$t^*\left(\frac{d}{K}-1, \sigma^2/\varepsilon, h_1, K\dot{\tau}_1, \ldots, h_n, K\dot{\tau}_n\right) \geq \frac{1}{24}t^*\left(d-1, \sigma^2/\varepsilon, h_1, \dot{\tau}_1, \ldots, h_n, \dot{\tau}_n\right).$$

$\square$

# F  Derivations of the Examples for the Equilibrium Time

*Example* 6.3. **[Infinitely Fast Worker]** If exists $j \in [n]$ such that $\tau_j = 0$ and $h_j = 0$, then $t^* = 0$.

*Proof.* Let us take a permutation $\pi$ where $\pi_1 = j$. Such a permutation exists because $\max\{h_j, \tau_j\} = 0$. By the definition of $t^*$, we have

$$\begin{aligned}
t^*(\omega, \sigma^2/\varepsilon, h_1, \tau_1, \ldots, h_n, \tau_n) &= \min_{j\in[n]}\max\{\max\{h_{\pi_j}, \tau_{\pi_j}\}, s^*(j)\} \\
&\leq \max\{\max\{h_{\pi_1}, \tau_{\pi_1}\}, s^*(1)\} \\
&= \max\{0, s^*(1)\},
\end{aligned}$$

(37)

where $s^*(1)$ is the solution of

$$\left(\sum_{i=1}^{1}\frac{1}{2\tau_{\pi_i}\omega + \frac{4\tau_{\pi_i}h_{\pi_i}\sigma^2\omega}{s\varepsilon} + \frac{2h_{\pi_i}\sigma^2}{\varepsilon}}\right)^{-1} = s.$$

Since

$$\left(\sum_{i=1}^{1}\frac{1}{2\tau_{\pi_i}\omega + \frac{4\tau_{\pi_i}h_{\pi_i}\sigma^2\omega}{s\varepsilon} + \frac{2h_{\pi_i}\sigma^2}{\varepsilon}}\right)^{-1} = \left(\sum_{i=1}^{1}\frac{1}{0}\right)^{-1} = (\infty)^{-1} = 0,$$

we obtain $s^*(1) = 0$. We substitute it to (37) to get $t^*(\omega, \sigma^2/\varepsilon, h_1, \tau_1, \ldots, h_n, \tau_n) = 0$. $\square$

*Example* 6.4. **[Infinitely Slow Workers]** If $\tau_i = \infty$ and $h_i = \infty$ for all $i \in [n]$, then $t^* = \infty$.

*Proof.* By the definition of $t^*$, we have

$$t^*(\omega, \sigma^2/\varepsilon, h_1, \tau_1, \ldots, h_n, \tau_n) = \min_{j\in[n]}\max\{\max\{h_{\pi_j}, \tau_{\pi_j}\}, s^*(j)\} = \min_{j\in[n]}\max\{\infty, s^*(j)\} = \infty.$$

$\square$

*Example* 6.5. [**Equal Performance**] If $\tau_i = \tau$ and $h_i = h$ for all $i \in [n]$, then[10]

$$t^* \leq 6 \max \left\{ h, \tau, \tfrac{\tau\omega}{n}, \tfrac{h\sigma^2}{n\varepsilon}, \sqrt{\tfrac{\tau h \sigma^2 \omega}{n\varepsilon}} \right\}. \tag{14}$$

*Proof.* By the definition, we have

$$t^*(\omega, \sigma^2/\varepsilon, h_1, \tau_1, \ldots, h_n, \tau_n) = \min_{j \in [n]} \max\{\max\{h, \tau\}, s^*(j)\} = \max\{\max\{h, \tau\}, \min_{j \in [n]} s^*(j)\} = \max\{\max\{h, \tau\}, s^*\} \tag{38}$$

where $s^*$ is the solution of

$$\left( \sum_{i=1}^{n} \frac{1}{2\tau\omega + \frac{4\tau h \sigma^2 \omega}{s\varepsilon} + \frac{2h\sigma^2}{\varepsilon}} \right)^{-1} = s.$$

Since

$$\left( \sum_{i=1}^{n} \frac{1}{2\tau\omega + \frac{4\tau h \sigma^2 \omega}{s\varepsilon} + \frac{2h\sigma^2}{\varepsilon}} \right)^{-1} = \left( \frac{n}{2\tau\omega + \frac{4\tau h \sigma^2 \omega}{s\varepsilon} + \frac{2h\sigma^2}{\varepsilon}} \right)^{-1} = \frac{2\tau\omega}{n} + \frac{4\tau h \sigma^2 \omega}{sn\varepsilon} + \frac{2h\sigma^2}{n\varepsilon},$$

we have to solve and find the non-negative solution of the quadratic equation

$$s^2 - s \left( \frac{2\tau\omega}{n} + \frac{2h\sigma^2}{n\varepsilon} \right) - \frac{4\tau h \sigma^2 \omega}{n\varepsilon} = 0.$$

The solution is

$$s^* = \left( \frac{\tau\omega}{n} + \frac{h\sigma^2}{n\varepsilon} \right) + \sqrt{\left( \frac{\tau\omega}{n} + \frac{h\sigma^2}{n\varepsilon} \right)^2 + \frac{4\tau h \sigma^2 \omega}{n\varepsilon}} \leq 2 \left( \frac{\tau\omega}{n} + \frac{h\sigma^2}{n\varepsilon} + \sqrt{\frac{\tau h \sigma^2 \omega}{n\varepsilon}} \right).$$

Therefore, we have

$$t^*(\omega, \sigma^2/\varepsilon, h_1, \tau_1, \ldots, h_n, \tau_n) \leq \max \left\{ \max\{h, \tau\}, 2 \left( \frac{\tau\omega}{n} + \frac{h\sigma^2}{n\varepsilon} + \sqrt{\frac{\tau h \sigma^2 \omega}{n\varepsilon}} \right) \right\}$$

$$\leq 6 \max \left\{ h, \tau, \frac{\tau\omega}{n}, \frac{h\sigma^2}{n\varepsilon}, \sqrt{\frac{\tau h \sigma^2 \omega}{n\varepsilon}} \right\}.$$

$\square$

*Example* 6.6. [**Infinitely Fast Communication**] If $\tau_i = 0$ for all $i \in [n]$, then

$$t^* \leq 2 \min_{m \in [n]} \max \left\{ h_{\pi_m}, \frac{\sigma^2}{\varepsilon} \left( \sum_{i=1}^{m} \frac{1}{h_{\pi_i}} \right)^{-1} \right\} = \Theta \left( \min_{m \in [n]} \left( \frac{1}{m} \sum_{i=1}^{m} \frac{1}{h_{\pi_i}} \right)^{-1} \left( 1 + \frac{\sigma^2}{m\varepsilon} \right) \right), \tag{15}$$

where $\pi$ is a permutation that sorts $\{h_i\}_{i=1}^{n}$.

*Proof.* By the definition, we have $t^*(\omega, \sigma^2/\varepsilon, h_1, \tau_1, \ldots, h_n, \tau_n) = \min_{j \in [n]} \max\{h_{\pi_j}, s^*(j)\}$, where $s^*(j)$ is the solution of

$$\left( \sum_{i=1}^{j} \frac{1}{\frac{2h_{\pi_i}\sigma^2}{\varepsilon}} \right)^{-1} = s.$$

Therefore, we have

$$t^*(\omega, \sigma^2/\varepsilon, h_1, \tau_1, \ldots, h_n, \tau_n) \leq 2 \min_{j \in [n]} \max \left\{ h_{\pi_j}, \frac{\sigma^2}{\varepsilon} \left( \sum_{i=1}^{j} \frac{1}{h_{\pi_i}} \right)^{-1} \right\} = \Theta \left( \min_{j \in [n]} \left( h_{\pi_j} + \frac{\sigma^2}{\varepsilon} \left( \sum_{i=1}^{j} \frac{1}{h_{\pi_i}} \right)^{-1} \right) \right).$$

Using Lemma F.1, we obtain

$$t^*(\omega, \sigma^2/\varepsilon, h_1, \tau_1, \ldots, h_n, \tau_n) \leq 2 \min_{j \in [n]} \max \left\{ h_{\pi_j}, \frac{\sigma^2}{\varepsilon} \left( \sum_{i=1}^{j} \frac{1}{h_{\pi_i}} \right)^{-1} \right\} = \Theta \left( \min_{j \in [n]} \left( j + \frac{\sigma^2}{\varepsilon} \right) \left( \sum_{i=1}^{j} \frac{1}{h_{\pi_i}} \right)^{-1} \right).$$

$\square$

---

[10]From the proof, it is clear that the result is tight up to a constant factor.

**Lemma F.1.** *Let us consider the two functions*

$$g(j) := h_j + a \left( \sum_{i=1}^{j} \frac{1}{h_i} \right)^{-1}, \qquad p(j) := (j + a) \left( \sum_{i=1}^{j} \frac{1}{h_i} \right)^{-1}$$

*for all $j \in [n]$, where $h_i \geq 0$ for all $i \in [n]$, $a \geq 0$, and $h_1 \leq \cdots \leq h_n$. Then*

$$\frac{1}{2} \min_{j \in [n]} g(j) \leq \min_{j \in [n]} p(j) \leq \min_{j \in [n]} g(j)$$

*Proof.* If $h_1 = 0$, then $p(1) = h(1) = 0$ and $\min_{i \in [n]} p(i) = \min_{i \in [n]} h(i) = 0$. Assume that $h_1 > 0$. Using the fact that a harmonic mean is less or equal to the maximum, we have

$$\min_{j \in [n]} p(j) = \min_{j \in [n]} (j + a) \left( \sum_{i=1}^{j} \frac{1}{h_i} \right)^{-1} \leq \min_{j \in [n]} \left( h_j + a \left( \sum_{i=1}^{j} \frac{1}{h_i} \right)^{-1} \right) = \min_{j \in [n]} g(j).$$

Thus, we proved the upper bound. Next, assume that $j^*$ is the smallest minimizer of $p(j)$. If $j^* = 1$, then

$$\min_{j \in [n]} p(j) = p(1) = (1 + a) h_1 = h_1 + a h_1 = g(1) \geq \min_{j \in [n]} g(j).$$

Otherwise, if $j^* > 1$, then $p(j^*) \leq p(j^* - 1)$. Using simple algebra, we obtain

$$(j^* + a) \left( \sum_{i=1}^{j^*} \frac{1}{h_i} \right)^{-1} \leq (j^* - 1 + a) \left( \sum_{i=1}^{j^*-1} \frac{1}{h_i} \right)^{-1}$$

$$\Leftrightarrow (j^* + a) \left( \sum_{i=1}^{j^*-1} \frac{1}{h_i} \right) \leq (j^* - 1 + a) \left( \sum_{i=1}^{j^*} \frac{1}{h_i} \right)$$

$$\Leftrightarrow \left( \sum_{i=1}^{j^*} \frac{1}{h_i} \right) \leq (j^* + a) \left( \frac{1}{h_{j^*}} \right)$$

$$\Leftrightarrow h_{j^*} \leq (j^* + a) \left( \sum_{i=1}^{j^*} \frac{1}{h_i} \right)^{-1}.$$

Using the last inequality, we get

$$\min_{j \in [n]} p(j) = (j^* + a) \left( \sum_{i=1}^{j^*} \frac{1}{h_i} \right)^{-1} = \frac{1}{2} (j^* + a) \left( \sum_{i=1}^{j^*} \frac{1}{h_i} \right)^{-1} + \frac{1}{2} (j^* + a) \left( \sum_{i=1}^{j^*} \frac{1}{h_i} \right)^{-1}$$

$$\geq \frac{1}{2} h_{j^*} + \frac{1}{2} a \left( \sum_{i=1}^{j^*} \frac{1}{h_i} \right)^{-1} = \frac{1}{2} g(j^*) \geq \frac{1}{2} \min_{j \in [n]} g(j).$$

$\square$

*Example* 6.7. [**Ignoring Slow Workers**] If $h_i$ and $\tau_i$ are fixed and finite for all $i \leq p$, and $\max\{h_i, \tau_i\} = m \in \mathbb{R}$ for all $i > p$, then, for $m$ large enough, we have $t^*(\omega, \sigma^2/\varepsilon, [h_i, \tau_i]_1^n) = t^*(\omega, \sigma^2/\varepsilon, [h_i, \tau_i]_1^p)$.

*Proof.* By the definition, we have

$$t^*(\omega, \sigma^2/\varepsilon, h_1, \tau_1, \ldots, h_n, \tau_n) = \min_{j \in [n]} \max\{\max\{h_{\pi_j}, \tau_{\pi_j}\}, s^*(j)\}.$$

For $m > \max_{i\in[p]} \max\{h_i,\tau_i\}$, we have $\max\{h_i,\tau_i\} < m$ for all $i \le p$ and the set $\{1,\ldots,p\}$ equals to $\{\pi_1,\ldots,\pi_p\}$. Thus, we get

$$t^*(\omega,\sigma^2/\varepsilon,h_1,\tau_1,\ldots,h_n,\tau_n) = \min\left\{\min_{j\in[p]}\max\{\max\{h_{\pi_j},\tau_{\pi_j}\},s^*(j)\}, \min_{j\in\{p+1,\ldots,n\}}\max\{m,s^*(j)\}\right\}$$

$$= \min\left\{t^*(\omega,\sigma^2/\varepsilon,h_1,\tau_1,\ldots,h_p,\tau_p), \min_{j\in\{p+1,\ldots,n\}}\max\{m,s^*(j)\}\right\}$$

By taking $m > t^*(\omega,\sigma^2/\varepsilon,h_1,\tau_1,\ldots,h_p,\tau_p)$, we obtain

$$\min_{j\in\{p+1,\ldots,n\}}\max\{m,s^*(j)\} \ge m > t^*(\omega,\sigma^2/\varepsilon,h_1,\tau_1,\ldots,h_p,\tau_p).$$

Therefore, we have

$$t^*(\omega,\sigma^2/\varepsilon,h_1,\tau_1,\ldots,h_n,\tau_n) = t^*(\omega,\sigma^2/\varepsilon,h_1,\tau_1,\ldots,h_p,\tau_p).$$

$\square$

*Example* 6.8. **[Partial Participation]** If $\max\{h_i,\tau_i\} = \infty$ for all $i > p \ge 1$, then $t^*(\omega,\sigma^2/\varepsilon,[h_i,\tau_i]_1^n) = t^*(\omega,\sigma^2/\varepsilon,[h_i,\tau_i]_1^p)$.

*Proof.* By the definition, we have

$$t^*(\omega,\sigma^2/\varepsilon,h_1,\tau_1,\ldots,h_n,\tau_n) = \min_{j\in[n]}\max\{\max\{h_{\pi_j},\tau_{\pi_j}\},s^*(j)\},$$

where $\pi$ is a permutation such that the set $\{\pi_{p+1},\ldots,\pi_n\}$ equals to the set $\{p+1,\ldots,n\}$. Such a permutation exists because $\max\{h_i,\tau_i\} = \infty$ for all $i > p$. Using this, we have

$$t^*(\omega,\sigma^2/\varepsilon,h_1,\tau_1,\ldots,h_n,\tau_n) = \min\left\{\min_{j\in[p]}\max\{\max\{h_{\pi_j},\tau_{\pi_j}\},s^*(j)\}, \min_{j\in\{p+1,\ldots,n\}}\max\{\max\{h_{\pi_j},\tau_{\pi_j}\},s^*(j)\}\right\}$$

$$= \min\left\{\min_{j\in[p]}\max\{\max\{h_{\pi_j},\tau_{\pi_j}\},s^*(j)\},\infty\right\}$$

$$= \min_{j\in[p]}\max\{\max\{h_{\pi_j},\tau_{\pi_j}\},s^*(j)\}$$

$$= t^*(\omega,\sigma^2/\varepsilon,h_1,\tau_1,\ldots,h_p,\tau_p).$$

$\square$

# G   Generic Lemma For Unbiased Gradient Estimators

We prove the following generic lemma that estimates the variance of the general family of unbiased gradient estimators.

**Lemma G.1.** *Consider that Assumptions 1.3 and 2.2 hold. Let us consider the gradient estimator*

$$g^k = \frac{1}{\sum_{i=1}^n\sum_{j=1}^{m_i}w_{ij}b_{ij}}\sum_{i=1}^n\sum_{j=1}^{m_i}w_{ij}\mathcal{C}_{ij}\left(\sum_{l=1}^{b_{ij}}\nabla f(x^k;\xi_{il}^k)\right),$$

*where $m_i \ge 0$ for all $i \in [n]$, $b_{ij} \ge 0$ for all $i \in [n]$ and $j \in [m_i]$ are ordered batch sizes ($b_{i1} \le \cdots \le b_{i,m_i}$ for all $i \in [n]$), $w_{ij} \ge 0$ are weights for all $i \in [n]$ and $j \in [m_i]$, and $\sum_{i=1}^n\sum_{j=1}^{m_i}w_{ij}b_{ij} > 0, \mathcal{C}_{ij} \in \mathbb{U}(\omega_{ij})$ are mutually independent compressors from Def. 2.1 for all $i \in [n]$ and $j \in [m_i]$, and $x^k \in \mathbb{R}^d$ is an arbitrary point. Then $\mathbb{E}\left[g^k\right] = \nabla f(x^k)$ and*

$$\mathbb{E}\left[\|g^k-\nabla f(x^k)\|^2\right] \le \frac{1}{\left(\sum_{i=1}^n\sum_{j=1}^{m_i}w_{ij}b_{ij}\right)^2}\sum_{i=1}^n\sum_{j=1}^{m_i}w_{ij}^2b_{ij}^2\omega_{ij}\|\nabla f(x^k)\|^2$$

$$+ \frac{1}{\left(\sum_{i=1}^n\sum_{j=1}^{m_i}w_{ij}b_{ij}\right)^2}\sum_{i=1}^n\left(\sum_{j=1}^{m_i}w_{ij}^2b_{ij}\omega_{ij}\sigma^2 + \sum_{j=1}^{m_i}\sum_{p=1}^{m_i}\min\{b_{ij},b_{ip}\}w_{ij}w_{ip}\sigma^2\right).$$

(39)

*Proof.* First, we show the gradient estimator is unbiased:

$$\mathbb{E}\left[g^k\right] = \mathbb{E}\left[\frac{1}{\sum_{i=1}^n \sum_{j=1}^{m_i} w_{ij} b_{ij}} \sum_{i=1}^n \sum_{j=1}^{m_i} w_{ij} \mathcal{C}_{ij}\left(\sum_{l=1}^{b_{ij}} \nabla f(x^k; \xi_{il}^k)\right)\right]$$

$$= \frac{1}{\sum_{i=1}^n \sum_{j=1}^{m_i} w_{ij} b_{ij}} \sum_{i=1}^n \sum_{j=1}^{m_i} w_{ij} \mathbb{E}\left[\mathcal{C}_{ij}\left(\sum_{l=1}^{b_{ij}} \nabla f(x^k; \xi_{il}^k)\right)\right].$$

Using Def. 2.1 and Assumption 1.3, we have

$$\mathbb{E}\left[g^k\right] = \frac{1}{\sum_{i=1}^n \sum_{j=1}^{m_i} w_{ij} b_{ij}} \sum_{i=1}^n \sum_{j=1}^{m_i} w_{ij} b_{ij} \nabla f(x^k) = \nabla f(x^k).$$

Next, we estimate the variance

$$\mathbb{E}\left[\left\|g^k - \nabla f(x^k)\right\|^2\right]$$

$$= \mathbb{E}\left[\left\|\frac{1}{\sum_{i=1}^n \sum_{j=1}^{m_i} w_{ij} b_{ij}} \sum_{i=1}^n \sum_{j=1}^{m_i} w_{ij} \mathcal{C}_{ij}\left(\sum_{l=1}^{b_{ij}} \nabla f(x^k; \xi_{il}^k)\right) - \nabla f(x^k)\right\|\right]$$

$$= \frac{1}{\left(\sum_{i=1}^n \sum_{j=1}^{m_i} w_{ij} b_{ij}\right)^2} \mathbb{E}\left[\left\|\sum_{i=1}^n \sum_{j=1}^{m_i} w_{ij} \mathcal{C}_{ij}\left(\sum_{l=1}^{b_{ij}} \nabla f(x^k; \xi_{il}^k)\right) - \sum_{i=1}^n \sum_{j=1}^{m_i} w_{ij} b_{ij} \nabla f(x^k)\right\|^2\right].$$

Using the independence and (22), we have

$$\mathbb{E}\left[\left\|g^k - \nabla f(x^k)\right\|^2\right]$$

$$= \frac{1}{\left(\sum_{i=1}^n \sum_{j=1}^{m_i} w_{ij} b_{ij}\right)^2} \sum_{i=1}^n \mathbb{E}\left[\left\|\sum_{j=1}^{m_i} w_{ij} \mathcal{C}_{ij}\left(\sum_{l=1}^{b_{ij}} \nabla f(x^k; \xi_{il}^k)\right) - \sum_{j=1}^{m_i} w_{ij} b_{ij} \nabla f(x^k)\right\|^2\right]$$

$$= \underbrace{\frac{1}{\left(\sum_{i=1}^n \sum_{j=1}^{m_i} w_{ij} b_{ij}\right)^2} \sum_{i=1}^n \mathbb{E}\left[\left\|\sum_{j=1}^{m_i} w_{ij} \mathcal{C}_{ij}\left(\sum_{l=1}^{b_{ij}} \nabla f(x^k; \xi_{il}^k)\right) - \sum_{j=1}^{m_i} w_{ij} \sum_{l=1}^{b_{ij}} \nabla f(x^k; \xi_{il}^k)\right\|^2\right]}_{I_1}$$

$$+ \underbrace{\frac{1}{\left(\sum_{i=1}^n \sum_{j=1}^{m_i} w_{ij} b_{ij}\right)^2} \sum_{i=1}^n \mathbb{E}\left[\left\|\sum_{j=1}^{m_i} w_{ij} \sum_{l=1}^{b_{ij}} \nabla f(x^k; \xi_{il}^k) - \sum_{j=1}^{m_i} w_{ij} b_{ij} \nabla f(x^k)\right\|^2\right]}_{I_2}.$$

$$(40)$$

Using the independence of the compressors and Def. 2.1, we get

$$I_1 = \frac{1}{\left(\sum_{i=1}^n \sum_{j=1}^{m_i} w_{ij} b_{ij}\right)^2} \sum_{i=1}^n \sum_{j=1}^{m_i} w_{ij}^2 \mathbb{E}\left[\left\|\mathcal{C}_{ij}\left(\sum_{l=1}^{b_{ij}} \nabla f(x^k; \xi_{il}^k)\right) - \sum_{l=1}^{b_{ij}} \nabla f(x^k; \xi_{il}^k)\right\|^2\right]$$

$$\leq \frac{1}{\left(\sum_{i=1}^n \sum_{j=1}^{m_i} w_{ij} b_{ij}\right)^2} \sum_{i=1}^n \sum_{j=1}^{m_i} w_{ij}^2 \omega_{ij} \mathbb{E}\left[\left\|\sum_{l=1}^{b_{ij}} \nabla f(x^k; \xi_{il}^k)\right\|^2\right].$$

In the view of (22), the independence of the stochastic gradients, and Assumption 1.3, we obtain

$$I_1 \leq \frac{1}{\left(\sum_{i=1}^n \sum_{j=1}^{m_i} w_{ij} b_{ij}\right)^2} \sum_{i=1}^n \sum_{j=1}^{m_i} w_{ij}^2 \omega_{ij} \mathbb{E}\left[\left\|\sum_{l=1}^{b_{ij}} \nabla f(x^k; \xi_{il}^k) - \sum_{l=1}^{b_{ij}} \nabla f(x^k)\right\|^2\right]$$

$$+ \frac{1}{\left(\sum_{i=1}^{n}\sum_{j=1}^{m_i} w_{ij}b_{ij}\right)^2} \sum_{i=1}^{n}\sum_{j=1}^{m_i} w_{ij}^2 \omega_{ij} b_{ij}^2 \left\|\nabla f(x^k)\right\|^2$$

$$= \frac{1}{\left(\sum_{i=1}^{n}\sum_{j=1}^{m_i} w_{ij}b_{ij}\right)^2} \sum_{i=1}^{n}\sum_{j=1}^{m_i} w_{ij}^2 \omega_{ij} \sum_{l=1}^{b_{ij}} \mathbb{E}\left[\left\|\nabla f(x^k; \xi_{il}^k) - \nabla f(x^k)\right\|^2\right]$$

$$+ \frac{1}{\left(\sum_{i=1}^{n}\sum_{j=1}^{m_i} w_{ij}b_{ij}\right)^2} \sum_{i=1}^{n}\sum_{j=1}^{m_i} w_{ij}^2 \omega_{ij} b_{ij}^2 \left\|\nabla f(x^k)\right\|^2$$

$$\leq \frac{1}{\left(\sum_{i=1}^{n}\sum_{j=1}^{m_i} w_{ij}b_{ij}\right)^2} \sum_{i=1}^{n}\sum_{j=1}^{m_i} w_{ij}^2 b_{ij}\omega_{ij}\sigma^2 + \frac{1}{\left(\sum_{i=1}^{n}\sum_{j=1}^{m_i} w_{ij}b_{ij}\right)^2} \sum_{i=1}^{n}\sum_{j=1}^{m_i} w_{ij}^2 b_{ij}^2 \omega_{ij} \left\|\nabla f(x^k)\right\|^2.$$

We now consider

$$I_2 = \frac{1}{\left(\sum_{i=1}^{n}\sum_{j=1}^{m_i} w_{ij}b_{ij}\right)^2} \sum_{i=1}^{n} \mathbb{E}\left[\left\|\sum_{j=1}^{m_i} w_{ij} \sum_{l=1}^{b_{ij}} \nabla f(x^k; \xi_{il}^k) - \sum_{j=1}^{m_i} w_{ij}b_{ij}\nabla f(x^k)\right\|^2\right].$$

Let us consider the set $S_{il} := \{j \in [m_i] \,|\, l \leq b_{ij}\}$ for all $i, l \in \mathbb{N}$. Then we can rewrite the norm in the following way

$$I_2 = \frac{1}{\left(\sum_{i=1}^{n}\sum_{j=1}^{m_i} w_{ij}b_{ij}\right)^2} \sum_{i=1}^{n} \mathbb{E}\left[\left\|\sum_{l=1}^{b_{i,m_i}} \left(\sum_{j \in S_{il}} w_{ij}\right) \left(\nabla f(x^k; \xi_{il}^k) - \nabla f(x^k)\right)\right\|^2\right].$$

The stochastic vectors are independent, thus

$$I_2 = \frac{1}{\left(\sum_{i=1}^{n}\sum_{j=1}^{m_i} w_{ij}b_{ij}\right)^2} \sum_{i=1}^{n}\sum_{l=1}^{b_{i,m_i}} \left(\sum_{j \in S_{il}} w_{ij}\right)^2 \mathbb{E}\left[\left\|\nabla f(x^k; \xi_{il}^k) - \nabla f(x^k)\right\|^2\right]$$

$$\leq \frac{1}{\left(\sum_{i=1}^{n}\sum_{j=1}^{m_i} w_{ij}b_{ij}\right)^2} \sum_{i=1}^{n}\sum_{l=1}^{b_{i,m_i}} \left(\sum_{j \in S_{il}} w_{ij}\right)^2 \sigma^2.$$

Note that

$$\sum_{l=1}^{b_{i,m_i}} \left(\sum_{j \in S_{il}} w_{ij}\right)^2 = \sum_{l=1}^{b_{i,m_i}} \sum_{j \in S_{il}} \sum_{p \in S_{il}} w_{ij}w_{ip}.$$

The number of appearances of the term $w_{ij}w_{ip}$ in the sum equals to $\min\{b_{ij}, b_{ip}\}$. Thus

$$I_2 \leq \frac{1}{\left(\sum_{i=1}^{n}\sum_{j=1}^{m_i} w_{ij}b_{ij}\right)^2} \sum_{i=1}^{n}\sum_{j=1}^{m_i}\sum_{p=1}^{m_i} \min\{b_{ij}, b_{ip}\} w_{ij}w_{ip}\sigma^2.$$

We now substitute the bounds on $I_1$ and $I_2$ to (40), and get (39). $\qquad\square$

**Algorithm 4** Shadowheart SGD        (Alg. 1 is equivalent to Alg. 4 when $h_i^k = h^k$ and $\tau_i^k = \tau^k$)

1: **Input:** starting point $x^0$, stepsize $\gamma$, the ratio $\sigma^2/\varepsilon$
2: **for** $k = 0, 1, \ldots, K - 1$ **do**
3:     Find the maximum computation speeds $h_i^k > 0$
    and compressors' communication speeds $\tau_i^k > 0$ of the workers in the current iteration
4:     Find the equilibrium time $t^*$ using Def. 3.1 with $h_i^k$ and $\tau_i^k$
5:     Set $b_i = \left\lfloor \frac{t^*}{h_i^k} \right\rfloor$ and $m_i = \left\lfloor \frac{t^*}{\tau_i^k} \right\rfloor$ for all $i \in [n]$  ($t^*$, $b_i$ and $m_i$ are local and can be different in
    every iteration)
6:     Find active workers $S_A = \{i \in [n] : b_i \wedge m_i > 0\}$
7:     Run Alg. 2 in all active workers $S_A$
8:     Broadcast $x^k, b_i$, and $m_i$ to all active workers $S_A$
9:     Init $g^k = 0$
10:    **for** $i \in S_A$ **in parallel do**
11:       $w_i \overset{(a)}{=} \left( b_i \omega + \omega \frac{\sigma^2}{\varepsilon} + m_i \frac{\sigma^2}{\varepsilon} \right)^{-1}$
12:       **for** $j = 1, \ldots, m_i$ **do**
13:          Receive $\mathcal{C}_{ij}\left(g_i^k\right)$ from the $i^{\text{th}}$ worker
14:          $g^k = g^k + w_i \mathcal{C}_{ij}\left(g_i^k\right)$
15:       **end for**
16:    **end for**
17:    $g^k = g^k / \left( \sum_{i=1}^{n} w_i m_i b_i \right)$
18:    $x^{k+1} = x^k - \gamma g^k$
19: **end for**
$(a)$ : If $\omega = 0$ and $\frac{\sigma^2}{\varepsilon} = 0$, then $w_i = 1$

## H   Proofs for Algorithms 1 and 4

In the appendix, we work with Alg. 4 instead of Alg. 1. Alg. 4 is more general and estimates all the parameters based on local per-iteration times $h_i^k$ and $\tau_i^k$ instead of $h_i$ and $\tau_i$. All results for Alg. 1 can be easily obtained by taking $h_i^k = h_i$ and $\tau_i^k = \tau_i$.

**Lemma H.1.** *Consider that Assumptions 1.3 and 2.2 hold. Then the gradient estimator* (9) *with the weights $w_i$ from Alg. 4 is unbiased and*

$$\mathbb{E}\left[\left\|g^k - \nabla f(x^k)\right\|^2\right] \leq \left( \sum_{i\,:\,b_i \wedge m_i > 0} \frac{b_i m_i}{b_i \omega + \omega \frac{\sigma^2}{\varepsilon} + m_i \frac{\sigma^2}{\varepsilon}} \right)^{-1} \left( \left\|\nabla f(x^k)\right\|^2 + \varepsilon \right). \qquad (41)$$

*Proof.* Alg. 4 implements the gradient estimator (9). We can use Lemma G.1 with $b_{ij} = b_i$, $w_{ij} = w_i$ and $\omega_{ij} = \omega$ for all $i \in [n]$ and $j \in [m_i]$. Using (39), we have

$$\mathbb{E}\left[\left\|g^k - \nabla f(x^k)\right\|^2\right] \leq \frac{1}{\left(\sum_{i=1}^{n}\sum_{j=1}^{m_i} w_{ij} b_{ij}\right)^2} \sum_{i=1}^{n}\sum_{j=1}^{m_i} w_{ij}^2 b_{ij}^2 \omega_{ij} \left\|\nabla f(x^k)\right\|^2 +$$

$$\frac{1}{\left(\sum_{i=1}^{n}\sum_{j=1}^{m_i} w_{ij} b_{ij}\right)^2} \sum_{i=1}^{n} \left( \sum_{j=1}^{m_i} w_{ij}^2 b_{ij} \omega_{ij} \sigma^2 + \sum_{j=1}^{m_i}\sum_{p=1}^{m_i} \min\{b_{ij}, b_{ip}\} w_{ij} w_{ip} \sigma^2 \right)$$

$$= \frac{1}{\left(\sum_{i=1}^{n} m_i w_i b_i\right)^2} \sum_{i=1}^{n} w_i^2 \left( m_i b_i^2 \omega \right) \left\|\nabla f(x^k)\right\|^2 +$$

$$\frac{1}{\left(\sum_{i=1}^{n} m_i w_i b_i\right)^2} \sum_{i=1}^{n} w_i^2 \left( m_i b_i \omega \sigma^2 + b_i m_i^2 \sigma^2 \right).$$

We add nonnegative terms to the last inequality to obtain

$$\mathbb{E}\left[\left\|g^k - \nabla f(x^k)\right\|^2\right] \leq \frac{1}{\left(\sum_{i=1}^{n} m_i w_i b_i\right)^2} \sum_{i=1}^{n} w_i^2 \left( m_i b_i^2 \omega + m_i b_i \omega \frac{\sigma^2}{\varepsilon} + b_i m_i^2 \frac{\sigma^2}{\varepsilon} \right) \left\|\nabla f(x^k)\right\|^2 +$$

$$\frac{1}{\left(\sum_{i=1}^{n} m_i w_i b_i\right)^2} \sum_{i=1}^{n} w_i^2 \left(m_i b_i^2 \omega \varepsilon + m_i b_i \omega \sigma^2 + b_i m_i^2 \sigma^2\right)$$

$$= \frac{1}{\left(\sum_{i=1}^{n} m_i w_i b_i\right)^2} \sum_{i=1}^{n} w_i^2 \left(m_i b_i^2 \omega + m_i b_i \omega \frac{\sigma^2}{\varepsilon} + b_i m_i^2 \frac{\sigma^2}{\varepsilon}\right) \left(\left\|\nabla f(x^k)\right\|^2 + \varepsilon\right).$$

Using the choice of the weights $w_i$, we get (41). $\qquad\square$

**Lemma H.2.** *Consider two quantities $\omega \geq 0$ and $\sigma^2/\varepsilon \geq 0$, and $n \in \mathbb{N}$. Also, consider a sequence of positive pairs $\{(h_i, \tau_i)\}_{i=1}^{n}$. We take $b_i = \left\lfloor \frac{t^*}{h_i} \right\rfloor$ and $m_i = \left\lfloor \frac{t^*}{\tau_i} \right\rfloor$ for all $i \in [n]$, where $t^* \equiv t^* \left(\omega, \sigma^2/\varepsilon, h_1, \tau_1, \ldots, h_n, \tau_n\right)$ is the equilibrium time from Def. 3.1. Then*

$$\left(\sum_{i \,:\, b_i \wedge m_i > 0} \frac{b_i m_i}{b_i \omega + \omega \frac{\sigma^2}{\varepsilon} + m_i \frac{\sigma^2}{\varepsilon}}\right)^{-1} \leq 1.$$

*Proof.* Assume that $j^* \in [n]$ is the smallest index that minimizes $\max\{\max\{h_{\pi_j}, \tau_{\pi_j}\}, s^*(j)\}$, then

$$t^* = \max\{\max\{h_{\pi_{j^*}}, \tau_{\pi_{j^*}}\}, s^*(j^*)\}. \tag{42}$$

Therefore, we have $t^* \geq \max\{h_{\pi_{j^*}}, \tau_{\pi_{j^*}}\} \geq \max\{h_{\pi_j}, \tau_{\pi_j}\}$ for all $j \leq j^*$ since $\max\{h_{\pi_j}, \tau_{\pi_j}\}$ are sorted. It means that $b_{\pi_j} = \left\lfloor \frac{t^*}{h_{\pi_j}} \right\rfloor \geq 1$ and $m_{\pi_j} = \left\lfloor \frac{t^*}{\tau_{\pi_j}} \right\rfloor \geq 1$ for all $j \leq j^*$. Using this, we have

$$I := \left(\sum_{i \,:\, b_i \wedge m_i > 0} \frac{b_i m_i}{b_i \omega + \omega \frac{\sigma^2}{\varepsilon} + m_i \frac{\sigma^2}{\varepsilon}}\right)^{-1} \leq \left(\sum_{i=1}^{j^*} \frac{b_{\pi_i} m_{\pi_i}}{b_{\pi_i} \omega + \omega \frac{\sigma^2}{\varepsilon} + m_{\pi_i} \frac{\sigma^2}{\varepsilon}}\right)^{-1} = \left(\sum_{i=1}^{j^*} \frac{1}{\frac{\omega}{m_{\pi_i}} + \frac{\omega \sigma^2}{b_{\pi_i} m_{\pi_i} \varepsilon} + \frac{\sigma^2}{b_{\pi_i} \varepsilon}}\right)^{-1}.$$

Since $b_{\pi_j} = \left\lfloor \frac{t^*}{h_{\pi_j}} \right\rfloor \geq 1$ and $m_{\pi_j} = \left\lfloor \frac{t^*}{\tau_{\pi_j}} \right\rfloor \geq 1$, we can also conclude that $b_{\pi_j} \geq \frac{t^*}{2h_{\pi_j}}$ and $m_{\pi_j} \geq \frac{t^*}{2\tau_{\pi_j}}$ for all $j \leq j^*$. Therefore, we obtain

$$I \leq \left(\sum_{i=1}^{j^*} \frac{1}{\frac{2\tau_{\pi_i} \omega}{t^*} + \frac{4\tau_{\pi_i} h_{\pi_i} \omega \sigma^2}{(t^*)^2 \varepsilon} + \frac{2h_{\pi_i} \sigma^2}{t^* \varepsilon}}\right)^{-1} \leq \left(\sum_{i=1}^{j^*} \frac{1}{\frac{2\tau_{\pi_i} \omega}{s^*(j^*)} + \frac{4\tau_{\pi_i} h_{\pi_i} \omega \sigma^2}{(s^*(j^*))^2 \varepsilon} + \frac{2h_{\pi_i} \sigma^2}{s^*(j^*) \varepsilon}}\right)^{-1}$$

$$= \frac{1}{s^*(j^*)} \left(\sum_{i=1}^{j^*} \frac{1}{2\tau_{\pi_i} \omega + \frac{4\tau_{\pi_i} h_{\pi_i} \omega \sigma^2}{s^*(j^*) \times \varepsilon} + \frac{2h_{\pi_i} \sigma^2}{\varepsilon}}\right)^{-1}.$$

where the last inequality follows from (42). Recall that $s^*(j^*)$ is the solution of the equation (7). Thus

$$\left(\sum_{i=1}^{j^*} \frac{1}{2\tau_{\pi_i} \omega + \frac{4\tau_{\pi_i} h_{\pi_i} \omega \sigma^2}{s^*(j^*) \times \varepsilon} + \frac{2h_{\pi_i} \sigma^2}{\varepsilon}}\right)^{-1} = s^*(j^*)$$

and

$$I \leq \frac{1}{s^*(j^*)} \times s^*(j^*) = 1.$$

$\qquad\square$

**Theorem H.3.** *Assume that Assumptions 1.1, 1.2, 1.3, 2.2 hold. Let us take $\gamma = \frac{1}{2L}$ in Alg. 4. Then for all iterations*

$$K \geq \frac{16L\Delta}{\varepsilon}, \tag{43}$$

*Alg. 4 guarantees that $\frac{1}{K} \sum_{k=0}^{K-1} \mathbb{E}\left[\left\|\nabla f(x^k)\right\|^2\right] \leq \varepsilon.$*

*Proof.* Let us fix any iteration $k \in \mathbb{N}$. Consider that $\mathcal{G}_k$ is a $\sigma$-algebra generated by $g^0, \ldots, g^{k-1}$. Then, given $\mathcal{G}_k$, $x^k$ is a deterministic vector. Using Lemma H.1, we have

$$\mathbb{E}\left[\left\|g^k - \nabla f(x^k)\right\|^2 \Big| \mathcal{G}_k\right] \leq \left(\sum_{i\,:\,b_i \wedge m_i > 0} \frac{b_i m_i}{b_i \omega + \omega \frac{\sigma^2}{\varepsilon} + m_i \frac{\sigma^2}{\varepsilon}}\right)^{-1} \left(\left\|\nabla f(x^k)\right\|^2 + \varepsilon\right).$$

Note that the choice of the parameters in Alg. 4 satisfy the conditions of Lemma H.2. Thus, we have

$$\mathbb{E}\left[\left\|g^k - \nabla f(x^k)\right\|^2 \Big| \mathcal{G}_k\right] \leq \left\|\nabla f(x^k)\right\|^2 + \varepsilon$$

for all $k \geq 0$. It is left to use the standard SGD analysis from Theorem I.1 with $B = 1$ and $C = \varepsilon$ to finish the proof. $\qquad\square$

**Theorem 4.2.** *Lett Assumptions 1.1, 1.2, 1.3, 2.2 hold. Let us take $\gamma = 1/2L$ in Shadowheart SGD (Alg. 1). Then as long as $K \geq 16L\Delta/\varepsilon$, we have the guarantee $\frac{1}{K}\sum_{k=0}^{K-1} \mathbb{E}\left[\left\|\nabla f(x^k)\right\|^2\right] \leq \varepsilon$.*

*Proof.* It immediately follows from Theorem H.3 for $h_i = h_i^k$ and $\tau_i = \tau_i^k$. $\qquad\square$

**Theorem 4.4.** *Alg. 4 converges after*

$$\sum_{k=0}^{\left\lceil \frac{16L\Delta}{\varepsilon} \right\rceil} 2t^*(\omega, \sigma^2/\varepsilon, h_1^k, \tau_1^k, \ldots, h_n^k, \tau_n^k) \tag{11}$$

*seconds, where $h_i^k > 0$ and $\tau_i^k > 0$ are computation and communication times for worker $i$ in iteration $k$.*

*Proof.* Let us fix an iteration index $k \in [n]$. In every iteration, every worker calculates $b_i$ stochastic gradients and sends $m_i$ compressed vectors. Thus, the processing time of each iteration is not greater than

$$\max_{i \in [n]} \left\{h_i^k b_i + \tau_i^k m_i\right\} = \max_{i \in [n]} \left\{h_i^k \left\lfloor \frac{t^*}{h_i^k} \right\rfloor + \tau_i^k \left\lfloor \frac{t^*}{\tau_i^k} \right\rfloor\right\}$$
$$\leq 2t^*(\omega, \sigma^2/\varepsilon, h_1^k, \tau_1^k, \ldots, h_n^k, \tau_n^k). \tag{44}$$

Using the fact that the number of iterations equals to (43), we finally get (11). $\qquad\square$

**Corollary 4.3.** Shadowheart SGD *(Alg. 1) converges after at most $T_*$ seconds, where*

$$T_* := \frac{32L\Delta}{\varepsilon} \times t^*(\omega, \sigma^2/\varepsilon, h_1, \tau_1, \ldots, h_n, \tau_n). \tag{10}$$

*Proof.* It immediately follows from Theorem 4.4 for $h_i = h_i^k$ and $\tau_i = \tau_i^k$. $\qquad\square$

# I The Classical SGD Theorem

Let us consider a slightly modified classical SGD result from (Ghadimi and Lan, 2013; Khaled and Richtárik, 2020).

**Theorem I.1.** *Assume that Assumptions 1.1 and 1.2 hold. We consider the SGD method:*

$$x^{k+1} = x^k - \gamma g(x^k),$$

*where*

$$\gamma = \min\left\{\frac{1}{L(1+B)}, \frac{\varepsilon}{2LC}\right\}$$

*For all $k \geq 0$, the vector $g(x)$ is a random vector such that $\mathbb{E}\left[g(x^k) \big| \mathcal{G}_k\right] = \nabla f(x^k)$,*

$$\mathbb{E}\left[\left\|g(x^k) - \nabla f(x^k)\right\|^2 \Big| \mathcal{G}_k\right] \leq B\left\|\nabla f(x^k)\right\|^2 + C, \tag{45}$$

where $\mathcal{G}_k$ is a $\sigma$-algebra generated by $g(x^0), \ldots, g(x^{k-1})$. The quantities $B$ and $C$ are arbitrary nonnegative constants. Then

$$\frac{1}{K} \sum_{k=0}^{K-1} \mathbb{E}\left[\left\|\nabla f(x^k)\right\|^2\right] \leq \varepsilon$$

for

$$K \geq \frac{4L\Delta(1+B)}{\varepsilon} + \frac{8L\Delta C}{\varepsilon^2}.$$

*Proof.* From Assumption 1.1, we have

$$f(x^{k+1}) \leq f(x^k) + \left\langle \nabla f(x^k), x^{k+1} - x^k \right\rangle + \frac{L}{2}\left\|x^{k+1} - x^k\right\|^2$$

$$= f(x^k) - \gamma \left\langle \nabla f(x^k), g(x^k) \right\rangle + \frac{L\gamma^2}{2}\left\|g(x^k)\right\|^2.$$

We denote $\mathcal{G}^k$ as a sigma-algebra generated by $g(x^0), \ldots, g(x^{k-1})$. Using unbiasedness and (45), we obtain

$$\mathbb{E}\left[f(x^{k+1})\big|\mathcal{G}^k\right] \leq f(x^k) - \gamma\left(1 - \frac{L\gamma}{2}\right)\left\|\nabla f(x^k)\right\|^2 + \frac{L\gamma^2}{2}\mathbb{E}\left[\left\|g(x^k) - \nabla f(x^k)\right\|^2\Big|\mathcal{G}^k\right]$$

$$\leq f(x^k) - \gamma\left(1 - \frac{L\gamma(1+B)}{2}\right)\left\|\nabla f(x^k)\right\|^2 + \frac{L\gamma^2 C}{2}.$$

Since $\gamma \leq \frac{1}{L(1+B)}$, we get

$$\mathbb{E}\left[f(x^{k+1})\big|\mathcal{G}^k\right] \leq f(x^k) - \frac{\gamma}{2}\left\|\nabla f(x^k)\right\|^2 + \frac{L\gamma^2 C}{2}.$$

We subtract $f^*$ and take the full expectation to obtain

$$\mathbb{E}\left[f(x^{k+1}) - f^*\right] \leq \mathbb{E}\left[f(x^k) - f^*\right] - \frac{\gamma}{2}\mathbb{E}\left[\left\|\nabla f(x^k)\right\|^2\right] + \frac{L\gamma^2 C}{2}.$$

Next, we sum the inequality for $k \in \{0, \ldots, K-1\}$:

$$\mathbb{E}\left[f(x^K) - f^*\right] \leq f(x^0) - f^* - \sum_{k=0}^{K-1}\frac{\gamma}{2}\mathbb{E}\left[\left\|\nabla f(x^k)\right\|^2\right] + \frac{KL\gamma^2 C}{2}$$

$$= \Delta - \sum_{k=0}^{K-1}\frac{\gamma}{2}\mathbb{E}\left[\left\|\nabla f(x^k)\right\|^2\right] + \frac{KL\gamma^2 C}{2}.$$

Finally, we rearrange the terms and use that $\mathbb{E}\left[f(x^K) - f^*\right] \geq 0$:

$$\frac{1}{K}\sum_{k=0}^{K-1}\mathbb{E}\left[\left\|\nabla f(x^k)\right\|^2\right] \leq \frac{2\Delta}{\gamma K} + L\gamma C.$$

The choice of $\gamma$ and $K$ ensures that

$$\frac{1}{K}\sum_{k=0}^{K-1}\mathbb{E}\left[\left\|\nabla f(x^k)\right\|^2\right] \leq \varepsilon.$$

$\square$

# J  Comparison with Baselines

In the following proofs, we use assumptions and definitions from Sec. 7.

*Comparison* 7.1. $T_* = \mathrm{O}(T_{\mathrm{MB}})$.

*Proof.* Without loss of generality, we assume that all workers are sorted by $\max\{h_i, \dot{\tau}_i\}$. For the Rand1 compressor, we have $\omega = d - 1$. From Corollary 4.3, we know that the time complexity of Alg. 1 is

$$T_* := \frac{L\Delta}{\varepsilon} \times t^*(\omega, \sigma^2/\varepsilon, h_1, \dot{\tau}_1, \ldots, h_n, \dot{\tau}_n)$$

up to a constant factor. Using Def. 3.1 of $t^*$, we have

$$T_* \leq \max\{\max\{h_n, \dot{\tau}_n\}, s^*(n)\} \times \frac{L\Delta}{\varepsilon}, \tag{46}$$

where $s^*(n)$ is the solution of

$$\left(\sum_{i=1}^{n} \frac{1}{2\dot{\tau}_i \omega + \frac{4\dot{\tau}_i h_i \sigma^2 \omega}{s \times \varepsilon} + \frac{2h_i \sigma^2}{\varepsilon}}\right)^{-1} = s. \tag{47}$$

Let us take $s' = 12 \max_{i \in [n]} \max\left\{\frac{h_i \sigma^2}{n\varepsilon}, \omega \dot{\tau}_i\right\}$. Using simple bounds, we have

$$\left(\sum_{i=1}^{n} \frac{1}{2\dot{\tau}_i \omega + \frac{4\dot{\tau}_i h_i \sigma^2 \omega}{s' \times \varepsilon} + \frac{2h_i \sigma^2}{\varepsilon}}\right)^{-1} \leq \left(\frac{n}{\max_{i \in [n]} \left(2\dot{\tau}_i \omega + \frac{4\dot{\tau}_i h_i \sigma^2 \omega}{s' \times \varepsilon} + \frac{2h_i \sigma^2}{\varepsilon}\right)}\right)^{-1}$$

$$= \frac{\max_{i \in [n]} \left(2\dot{\tau}_i \omega + \frac{4\dot{\tau}_i h_i \sigma^2 \omega}{s' \times \varepsilon} + \frac{2h_i \sigma^2}{\varepsilon}\right)}{n}.$$

Since $s' \geq \omega \max_{i \in [n]} \dot{\tau}_i$, we get

$$\left(\sum_{i=1}^{n} \frac{1}{2\dot{\tau}_i \omega + \frac{4\dot{\tau}_i h_i \sigma^2 \omega}{s' \times \varepsilon} + \frac{2h_i \sigma^2}{\varepsilon}}\right)^{-1} \leq \frac{\max_{i \in [n]} \left(2\dot{\tau}_i \omega + \frac{4h_i \sigma^2}{\varepsilon} + \frac{2h_i \sigma^2}{\varepsilon}\right)}{n}$$

$$\leq \frac{12 \max_{i \in [n]} \max\left\{\frac{h_i \sigma^2}{\varepsilon}, \omega \dot{\tau}_i\right\}}{n} \leq s'.$$

It means that $s^*(n) \leq s'$ since $s^*(n)$ is the solution of (47). Using the properties of $\max$ and (46), we get

$$T_* = O\left(\max\left\{\max\{h_n, \dot{\tau}_n\}, \max_{i \in [n]} \max\left\{\frac{h_i \sigma^2}{n\varepsilon}, \omega \dot{\tau}_i\right\}\right\} \times \frac{L\Delta}{\varepsilon}\right)$$

$$= O\left(\max\left\{\max_{i \in [n]} (h_i + (\omega + 1)\dot{\tau}_i), \left(\max_{i \in [n]} \frac{h_i \sigma^2}{n\varepsilon} + \max_{i \in [n]} \omega \dot{\tau}_i\right)\right\} \times \frac{L\Delta}{\varepsilon}\right)$$

$$= O\left(\max\left\{\max_{i \in [n]} (h_i + (\omega + 1)\dot{\tau}_i) \times \frac{L\Delta}{\varepsilon}, \max_{i \in [n]} h_i \times \frac{\sigma^2 L\Delta}{n\varepsilon^2}, \max_{i \in [n]} (\omega + 1)\dot{\tau}_i \times \frac{L\Delta}{\varepsilon}\right\}\right)$$

$$= O\left(\max_{i \in [n]} (h_i + (\omega + 1)\dot{\tau}_i) \left(\frac{L\Delta}{\varepsilon} + \frac{\sigma^2 L\Delta}{n\varepsilon^2}\right)\right)$$

$$= O\left(\max_{i \in [n]} (h_i + d\dot{\tau}_i) \left(\frac{L\Delta}{\varepsilon} + \frac{\sigma^2 L\Delta}{n\varepsilon^2}\right)\right),$$

where we use $\omega + 1 = d$. □

*Comparison* 7.2. $T_* = O(T_R)$.

*Proof.* From Sec. 7, we know that

$$T_* := \frac{L\Delta}{\varepsilon} \times t^*(\omega, \sigma^2/\varepsilon, h_1, \dot{\tau}_1, \ldots, h_n, \dot{\tau}_n)$$

and

$$T_R := \frac{L\Delta}{\varepsilon} \times t^*(0, \sigma^2/\varepsilon, h_1, d\dot{\tau}_1, \ldots, h_n, d\dot{\tau}_n).$$

Using Property E.7, we get $T_* = O(T_R)$ since $\omega = d - 1$ for Rand1. □

# K   Description of Alg. 5 in the Bidirectional Setting

In this section, we provide the modification of Alg. 4 with the EF21-P mechanism (Gruntkowska et al., 2023). Almost all steps are the same as in Alg. 4 except for the EF21-P mechanism (we mark the main changes with the color).

---

**Algorithm 5** Bidirectional Shadowheart SGD

1: **Input:** starting point $x^0$, stepsize $\gamma$, the ratio $\sigma^2/\varepsilon$
2: **for** $k = 0, 1, \ldots, K - 1$ **do**
3:     Find the current computation speeds $h_i^k > 0$
        and communication speeds $\tau_i^k > 0$ of the workers
4:     Find the equilibrium time $t^*$ using Def. 3.1
5:     Set $b_i = \left\lfloor \frac{t^*}{h_i^k} \right\rfloor$ and $m_i = \left\lfloor \frac{t^*}{\tau_i^k} \right\rfloor$ for all $i \in [n]$
6:     Find active workers $S_A = \{i \in [n] \ : \ b_i \wedge m_i > 0\}$
7:     Run Alg. 6 in all workers
8:     Broadcast $b_i$, and $m_i$ to all workers
9:     Init $g^k = 0$
10:    **for** $i \in S_A$ **in parallel do**
11:        $w_i \stackrel{(a)}{=} \left( b_i \omega + \omega \frac{\sigma^2}{\varepsilon} + m_i \frac{\sigma^2}{\varepsilon} \right)^{-1}$
12:        **for** $j = 1, \ldots, m_i$ **do**
13:            Receive $\mathcal{C}_{ij}\left(g_i^k\right)$ from the $i^{\text{th}}$ worker
14:            $g^k = g^k + w_i \mathcal{C}_{ij}\left(g_i^k\right)$
15:        **end for**
16:    **end for**
17:    $g^k = g^k / \left(\sum_{i=1}^n w_i m_i b_i\right)$
18:    $x^{k+1} = x^k - \gamma g^k$
19:    $p^{k+1} = \mathcal{C}_{\text{serv}}(x^{k+1} - w^k)$
20:    $w^{k+1} = w^k + p^{k+1}$
21:    Broadcast $p^{k+1}$ to all workers
22: **end for**
$(a)$ : If $\omega = 0$ and $\frac{\sigma^2}{\varepsilon} = 0$, then $w_i = 1$

---

**Algorithm 6** $i^{\text{th}}$ Worker's Strategy (init all workers with $w^0 = x^0$)

1: Receive $b_i$, and $m_i$ from the server
2: **if** $b_i \wedge m_i > 0$ **then**
3:     Init $g_i^k = 0$
4:     **for** $l = 1, \ldots, b_i$ **do**
5:         Calculate $\nabla f(w^k; \xi_{il}^k)$,     $\xi_{il}^k \sim \mathcal{D}_\xi$
6:         $g_i^k = g_i^k + \nabla f(w^k; \xi_{il}^k)$
7:     **end for**
8:     **for** $j = 1, \ldots, m_i$ **do**
9:         Send $\mathcal{C}_{ij}\left(g_i^k\right) \equiv \mathcal{C}\left(g_i^k; \nu_{ij}^k\right)$ to the server,
           $\nu_{ij}^k \sim \mathcal{D}_\nu, \mathcal{C}_{ij} \in \mathbb{U}(\omega)$
10:    **end for**
11: **end if**
12: Receive $p^{k+1}$ from the server
13: $w^{k+1} = w^k + p^{k+1}$

---

# L   Proofs for Alg. 5

**Theorem A.2.** *Let Assumptions 1.1, 1.2, 1.3, 2.2 hold. Choose $\gamma = \frac{\alpha}{16L}$. Then as long as $K \geq \frac{768L\Delta}{\alpha\varepsilon}$, Bidirectional Shadowheart SGD (Alg. 5) guarantees to find an $\varepsilon$–stationary point.*

*Proof.* In the bidirectional setting, the idea of proof is the same as in Theorem H.3. Let us fix any iteration $k \in \mathbb{N}$. The gradient estimator has the same structure as (9) but with $w^k$ instead of $x^k$ :

$$g^k = \frac{1}{\sum_{i=1}^n w_i m_i b_i} \sum_{i=1}^n w_i \sum_{j=1}^{m_i} \mathcal{C}_{ij} \left( \sum_{l=1}^{b_i} \nabla f(w^k; \xi_{il}^k) \right),$$

Consider that $\mathcal{G}_k$ is a $\sigma$-algebra generated by all random variables from the iterations $0, \ldots, k-1$. Then, given $\mathcal{G}_k$, $w^k$ is a deterministic vector. Using Lemma H.1 with $x^k \equiv w^k$ and Lemma H.2, we have

$$\mathbb{E}\left[ \left\| g^k - \nabla f(w^k) \right\|^2 \middle| \mathcal{G}_k \right] \le \left\| \nabla f(w^k) \right\|^2 + \varepsilon$$

for all $k \ge 0$. It is left to use Theorem E.3 from (Gruntkowska et al., 2023) with $B = 2$ and $C = \varepsilon$ to ensure that $\min_{0 \le k \le K-1} \mathbb{E}\left[ \left\| \nabla f(x^k) \right\|^2 \right] \le \varepsilon$ after $\frac{768 L \Delta}{\alpha \varepsilon}$ iterations. $\square$

**Corollary A.3.** *If the broadcast time of $\mathcal{C}_{\mathrm{serv}}$ is not greater than $\tau_{\mathrm{serv}}$, then* Bidirectional Shadowheart SGD *(Alg. 5) converges after at most*

$$T_{*,\mathrm{serv}} := \frac{768 L \Delta}{\alpha \varepsilon} \times \left( \tau_{\mathrm{serv}} + 2t^*(\omega, \sigma^2/\varepsilon, [h_i, \tau_i]_1^n) \right) \tag{20}$$

*seconds.*

*Proof.* Let us fix an iteration index $k \in [n]$. In every iteration, the server broadcasts one compressed vector, every worker calculates $b_i$ stochastic gradients and sends $m_i$ compressed vectors. Thus, the processing time of each iteration is not greater than

$$
\begin{aligned}
\tau_{\mathrm{serv}} + \max_{i \in [n]} \left\{ h_i^k b_i + \tau_i^k m_i \right\} &= \tau_{\mathrm{serv}} + \max_{i \in [n]} \left\{ h_i^k \left\lfloor \frac{t^*}{h_i^k} \right\rfloor + \tau_i^k \left\lfloor \frac{t^*}{\tau_i^k} \right\rfloor \right\} \\
&\le \tau_{\mathrm{serv}} + 2t^*(\omega, \sigma^2/\varepsilon, h_1^k, \tau_1^k, \ldots, h_n^k, \tau_n^k) \\
&\overset{\text{P.6.1}}{\le} \tau_{\mathrm{serv}} + 2t^*(\omega, \sigma^2/\varepsilon, h_1, \tau_1, \ldots, h_n, \tau_n).
\end{aligned}
$$

Using the converge rate from Theorem A.2, we finally get (20). $\square$

*Comparison* A.5. Assume that it takes $\dot{\tau}_{\mathrm{serv}}$ seconds to send *one coordinate* from the server to the workers. If we take $K \ge \min\left\{ d, t^*(\omega, \sigma^2/\varepsilon, [h_i, \tau_i]_1^n)/\dot{\tau}_{\mathrm{serv}} \right\}$ in TopK, then $T_{*,\mathrm{serv}} = \mathrm{O}(T_*)$.

*Proof.* From the assumption, we have $\tau_{\mathrm{serv}} = K\dot{\tau}_{\mathrm{serv}}$ and $\tau_{\mathrm{serv}}^{\mathrm{full}} = d\dot{\tau}_{\mathrm{serv}}$. For TopK, $\alpha \ge K/d$. Therefore, up to a constant factor, we obtain

$$
\begin{aligned}
T_{*,\mathrm{serv}} &= \frac{L\Delta}{\alpha \varepsilon} \times \left( \tau_{\mathrm{serv}} + t^*(\omega, \sigma^2/\varepsilon, [h_i, \tau_i]_1^n) \right) \\
&\le \frac{dL\Delta}{K\varepsilon} \times \left( K\dot{\tau}_{\mathrm{serv}} + t^*(\omega, \sigma^2/\varepsilon, [h_i, \tau_i]_1^n) \right) \\
&= \frac{dL\Delta}{\varepsilon}\dot{\tau}_{\mathrm{serv}} + \frac{dL\Delta}{K\varepsilon}t^*(\omega, \sigma^2/\varepsilon, [h_i, \tau_i]_1^n) \\
&\le \frac{dL\Delta}{\varepsilon}\dot{\tau}_{\mathrm{serv}} + \max\left\{ \frac{dL\Delta}{\varepsilon}\dot{\tau}_{\mathrm{serv}}, \frac{L\Delta}{\varepsilon}t^*(\omega, \sigma^2/\varepsilon, [h_i, \tau_i]_1^n) \right\} \\
&\le 2\left( \frac{dL\Delta}{\varepsilon}\dot{\tau}_{\mathrm{serv}} + \frac{L\Delta}{\varepsilon}t^*(\omega, \sigma^2/\varepsilon, [h_i, \tau_i]_1^n) \right).
\end{aligned}
$$

Also, up to a constant factor, we have

$$
\begin{aligned}
T_* &= \frac{L\Delta}{\varepsilon} \times (\tau_{\mathrm{serv}}^{\mathrm{full}} + t^*(\omega, \sigma^2/\varepsilon, [h_i, \tau_i]_1^n)) \\
&= \frac{L\Delta}{\varepsilon} \times (d\dot{\tau}_{\mathrm{serv}} + t^*(\omega, \sigma^2/\varepsilon, [h_i, \tau_i]_1^n)) \\
&= \frac{dL\Delta}{\varepsilon}\dot{\tau}_{\mathrm{serv}} + \frac{L\Delta}{\varepsilon}t^*(\omega, \sigma^2/\varepsilon, [h_i, \tau_i]_1^n).
\end{aligned}
$$

Therefore, $T_{*,\mathrm{serv}} = \mathrm{O}(T_*)$. $\square$

# M  Development of Adaptive Shadowheart SGD

---

**Algorithm 7** Adaptive Shadowheart SGD

---

1: **Input:** starting point $x^0$, stepsize $\gamma$, the ratio $\sigma^2/\varepsilon$
2: **for** $k = 0, 1, \ldots, K - 1$ **do**
3:      Run Alg. 8 in all workers
4:      Broadcast $x^k$ to the workers
5:      Init $l_i = 0$ for all $i \in [n]$
6:      **while** $\left( \sum_{i\,:\,l_i > 0} \left( \sum_{j=1}^{l_i} \left( \frac{\omega}{l_i^2 m_{ij}} + \frac{\omega\sigma^2}{l_i^3 m_{ij}\varepsilon} \right) + \frac{\sigma^2}{l_i \varepsilon} \right)^{-1} \right)^{-1} > \frac{1}{4}$ **do**
7:          Receive $\mathcal{C}_{i,l_i,m_{i,l_i}}(g_i)$, $l_i$ and $m_{i,l_i}$ from some worker (we indicate this worker with $i$)
8:          **if** $m_{i,l_i} = 1$ and $l_i > 1$ **then**
9:              $\bar{g}_i = \bar{g}_i + \frac{1}{m_{i,(l_i - 1)}} \hat{g}_i$ and $\hat{g}_i = 0$
10:         **end if**
11:          $\hat{g}_i = \hat{g}_i + \mathcal{C}_{i,l_i,m_{i,l_i}}(g_i)$
12:      **end while**
13:      Init $g^k = 0$
14:      **for** $i \in [n] : l_i > 0$ **do**
15:         $\bar{g}_i = \bar{g}_i + \frac{1}{m_{i,l_i}} \hat{g}_i$
16:         $w_i \overset{(a)}{=} \left( \sum_{j=1}^{l_i} \frac{\omega}{m_{ij}} + \sum_{j=1}^{l_i} \frac{\omega\sigma^2}{l_i m_{ij}\varepsilon} + \frac{l_i \sigma^2}{\varepsilon} \right)^{-1}$
17:         $g^k = g^k + w_i \bar{g}_i$
18:      **end for**
19:      $g^k = g^k / \left( \sum_{i\,:\,l_i > 0} w_i \sum_{j=1}^{l_i} j \right)$
20:      $x^{k+1} = x^k - \gamma g^k$
21: **end for**
$(a)$ : If $\omega = 0$ and $\frac{\sigma^2}{\varepsilon} = 0$, then $w_i = 1$

---

In this section, we design a new method that, unlike Alg. 1 and 4, does not require the bounds on computations times. It automatically understands when to stop the collection of compressed vectors in $g^k$.

Let us consider Alg. 7, which we call Adaptive Shadowheart SGD. It implements the following gradient estimator:

$$g^k = \frac{1}{\sum_{i=1}^{n} w_i \sum_{j=1}^{l_i} j} \sum_{i=1}^{n} w_i \sum_{j=1}^{l_i} \frac{1}{m_{ij}} \sum_{p=1}^{m_{ij}} \mathcal{C}_{ijp} \left( \sum_{r=1}^{j} \nabla f(x^k; \xi_{ir}^k) \right). \tag{48}$$

The idea is that each worker calculate and send compressed vectors *in parallel*: while the next stochastic gradients $\nabla f(x^k; \xi_{i,j+1}^k)$ are calculating, the workers are sending $\mathcal{C}_{ij\cdot} \left( \sum_{r=1}^{j} \nabla f(x^k; \xi_{ir}^k) \right)$ to server. The main difficulty is to understand when to stop. It turns out that it is sufficient to wait for the moment when the condition in Line 6 of Alg. 7 does not hold. For this method, we can prove the following guarantees.

**Theorem M.1.** *Let Assumptions 1.1, 1.2, 1.3, 2.2 hold. Let us take $\gamma = \frac{1}{2L}$ in Alg. 7. Then for all iterations $K \geq \frac{16L\Delta}{\varepsilon}$, Alg. 7 guarantees that $\frac{1}{K} \sum_{k=0}^{K-1} \mathbb{E}\left[ \left\| \nabla f(x^k) \right\|^2 \right] \leq \varepsilon$.*

**Corollary 4.6.** *If the computation and communication times are positive, the time complexity of Alg. 7 is $\frac{L\Delta}{\varepsilon} \times t^*(\omega, \sigma^2/\varepsilon, [\max\{h_i, \tau_i\}, \min\{\tau_i r_i, \max\{h_i, \tau_i\}\}]_1^n)$ up to a constant factor, where $r_i$ is defined in Def. 4.5.*

**Algorithm 8** $i^{\text{th}}$ Worker's Strategy

1: Receive $x^k$ from the server
2: Init $l_i = 1$
3: Calculate $g_i = \nabla f(x^k; \xi_{i1}^k), \quad \xi_{i1}^k \sim \mathcal{D}_\xi$
4: **while** True **do**
5:   Start calculating $\nabla f(x^k; \xi_{i,l_i+1}^k), \xi_{i,l_i+1}^k \sim \mathcal{D}_\xi$,
      and go to the next step
6:   Init $m_{i,l_i} = 0$
7:   **while** $\nabla f(x^k; \xi_{i,l_i+1}^k)$ is not calculated OR $m_{i,l_i} = 0$ **do**
8:     $m_{i,l_i} = m_{i,l_i} + 1$
9:     Send $\mathcal{C}_{i,l_i,m_{i,l_i}}(g_i), l_i$ and $m_{i,l_i}$ to the server, $\mathcal{C}_{i,l_i,m_{i,l_i}} \in \mathbb{U}(\omega)$
10:  **end while**
11:   $g_i = g_i + \nabla f(x^k; \xi_{i,l_i+1}^k)$
12:   $l_i = l_i + 1$
13: **end while**

# N   Proofs for Alg. 7

**Lemma N.1.** *Consider that Assumptions 1.3 and 2.2 hold. Then the gradient estimator* (48) *with the parameters from Alg. 7 is unbiased and*

$$
\mathbb{E}\left[\left\|g^k - \nabla f(x^k)\right\|^2\right] \leq 4 \left( \sum_{i \in [n]\,:\, l_i > 0} \left( \sum_{j=1}^{l_i} \frac{\omega}{l_i^2 m_{ij}} + \sum_{j=1}^{l_i} \frac{\omega \sigma^2}{l_i^3 m_{ij} \varepsilon} + \frac{\sigma^2}{l_i \varepsilon} \right)^{-1} \right)^{-1} \left( \left\|\nabla f(x^k)\right\|^2 + \varepsilon \right).
$$
(49)

*Proof.* Alg. 7 implements the gradient estimator (48). Note that since $\mathcal{C}_{ijp} \in \mathbb{U}(\omega)$, then $\frac{1}{m_{ij}} \sum_{p=1}^{m_{ij}} \mathcal{C}_{ijp} \in \mathbb{U}(\omega/m_{ij})$. Therefore, we can use Lemma G.1 with $\omega_{ij} = \omega/m_{ij}, b_{ij} = j, w_{ij} = w_i$, and $m_i = l_i$, and get

$$
\mathbb{E}\left[\left\|g^k - \nabla f(x^k)\right\|^2\right] \leq \frac{1}{\left( \sum_{i=1}^n w_i \sum_{j=1}^{l_i} j \right)^2} \sum_{i=1}^n w_i^2 \sum_{j=1}^{l_i} \frac{j^2 \omega}{m_{ij}} \left\|\nabla f(x^k)\right\|^2
$$

$$
+ \frac{1}{\left( \sum_{i=1}^n w_i \sum_{j=1}^{l_i} j \right)^2} \sum_{i=1}^n w_i^2 \left( \sum_{j=1}^{l_i} \frac{j \omega \sigma^2}{m_{ij}} + \sum_{j=1}^{l_i} \sum_{p=1}^{l_i} \min\{j, p\} \sigma^2 \right).
$$

Since $\sum_{j=1}^{l_i} \sum_{p=1}^{l_i} \min\{j, p\} \leq l_i^3$, we have

$$
\mathbb{E}\left[\left\|g^k - \nabla f(x^k)\right\|^2\right] \leq \frac{1}{\left( \sum_{i=1}^n w_i \sum_{j=1}^{l_i} j \right)^2} \sum_{i=1}^n w_i^2 \sum_{j=1}^{l_i} \frac{j^2 \omega}{m_{ij}} \left\|\nabla f(x^k)\right\|^2
$$

$$
+ \frac{1}{\left( \sum_{i=1}^n w_i \sum_{j=1}^{l_i} j \right)^2} \sum_{i=1}^n w_i^2 \left( \sum_{j=1}^{l_i} \frac{j \omega \sigma^2}{m_{ij}} + l_i^3 \sigma^2 \right).
$$

We add nonnegative terms to the last inequality to obtain

$$\mathbb{E}\left[\left\|g^k - \nabla f(x^k)\right\|^2\right] \leq \frac{1}{\left(\sum_{i=1}^n w_i \sum_{j=1}^{l_i} j\right)^2} \sum_{i=1}^n w_i^2 \left(\sum_{j=1}^{l_i} \frac{j^2 \omega}{m_{ij}} + \sum_{j=1}^{l_i} \frac{j\omega}{m_{ij}} \frac{\sigma^2}{\varepsilon} + l_i^3 \frac{\sigma^2}{\varepsilon}\right) \left\|\nabla f(x^k)\right\|^2$$

$$+ \frac{1}{\left(\sum_{i=1}^n w_i \sum_{j=1}^{l_i} j\right)^2} \sum_{i=1}^n w_i^2 \left(\sum_{j=1}^{l_i} \frac{j^2 \omega \varepsilon}{m_{ij}} + \sum_{j=1}^{l_i} \frac{j\omega\sigma^2}{m_{ij}} + l_i^3 \sigma^2\right)$$

$$= \frac{1}{\left(\sum_{i=1}^n w_i \sum_{j=1}^{l_i} j\right)^2} \sum_{i=1}^n w_i^2 \left(\sum_{j=1}^{l_i} \frac{j^2 \omega}{m_{ij}} + \sum_{j=1}^{l_i} \frac{j\omega}{m_{ij}} \frac{\sigma^2}{\varepsilon} + l_i^3 \frac{\sigma^2}{\varepsilon}\right) \left(\left\|\nabla f(x^k)\right\|^2 + \varepsilon\right).$$

Using $\sum_{j=1}^{l_i} j \geq \frac{l_i^2}{2}$, we obtain

$$\mathbb{E}\left[\left\|g^k - \nabla f(x^k)\right\|^2\right] \leq \frac{4}{\left(\sum_{i=1}^n w_i l_i^2\right)^2} \sum_{i=1}^n w_i^2 \left(\sum_{j=1}^{l_i} \frac{j^2 \omega}{m_{ij}} + \sum_{j=1}^{l_i} \frac{j\omega}{m_{ij}} \frac{\sigma^2}{\varepsilon} + l_i^3 \frac{\sigma^2}{\varepsilon}\right) \left(\left\|\nabla f(x^k)\right\|^2 + \varepsilon\right).$$

In the last two sums, we bound the terms $j$ with $l_i$ to get

$$\mathbb{E}\left[\left\|g^k - \nabla f(x^k)\right\|^2\right] \leq \frac{4}{\left(\sum_{i=1}^n w_i l_i^2\right)^2} \sum_{i=1}^n w_i^2 \left(\sum_{j=1}^{l_i} \frac{l_i^2 \omega}{m_{ij}} + \sum_{j=1}^{l_i} \frac{l_i \omega}{m_{ij}} \frac{\sigma^2}{\varepsilon} + l_i^3 \frac{\sigma^2}{\varepsilon}\right) \left(\left\|\nabla f(x^k)\right\|^2 + \varepsilon\right).$$

It is left to use the choice of the weights $w_i$ to obtain (49). $\qquad\square$

**Theorem M.1.** *Let Assumptions 1.1, 1.2, 1.3, 2.2 hold. Let us take $\gamma = \frac{1}{2L}$ in Alg. 7. Then for all iterations $K \geq \frac{16L\Delta}{\varepsilon}$, Alg. 7 guarantees that $\frac{1}{K} \sum_{k=0}^{K-1} \mathbb{E}\left[\left\|\nabla f(x^k)\right\|^2\right] \leq \varepsilon.$*

*Proof.* The proof of this theorem is very close to the proof of Theorem H.3. Let us fix any iteration $k \in \mathbb{N}$. Consider that $\mathcal{G}_k$ is a $\sigma$-algebra generated by $g^0, \dots, g^{k-1}$. Then, given $\mathcal{G}_k$, $x^k$ is a deterministic vector. Using Lemma N.1, we have

$$\mathbb{E}\left[\left\|g^k - \nabla f(x^k)\right\|^2 \Big| \mathcal{G}_k\right] \leq 4 \left(\sum_{i \in [n]\,:\,l_i > 0} \left(\sum_{j=1}^{l_i} \frac{\omega}{l_i^2 m_{ij}} + \sum_{j=1}^{l_i} \frac{\omega\sigma^2}{l_i^3 m_{ij}\varepsilon} + \frac{\sigma^2}{l_i\varepsilon}\right)^{-1}\right)^{-1} \left(\left\|\nabla f(x^k)\right\|^2 + \varepsilon\right).$$

The algorithm is constructed in such a way that the first bracket in the last inequality is less or equal to 1 (see Line 6 in Alg. 7). Thus

$$\mathbb{E}\left[\left\|g^k - \nabla f(x^k)\right\|^2 \Big| \mathcal{G}_k\right] \leq \left\|\nabla f(x^k)\right\|^2 + \varepsilon$$

for all $k \geq 0$. It is left to use the standard SGD analysis from Theorem I.1 with $B = 1$ and $C = \varepsilon$ to ensure that the algorithm converges after $\frac{16L\Delta}{\varepsilon}$ iterations. $\qquad\square$

**Corollary 4.6.** *If the computation and communication times are positive, the time complexity of Alg. 7 is $\frac{L\Delta}{\varepsilon} \times t^*(\omega, \sigma^2/\varepsilon, [\max\{h_i, \tau_i\}, \min\{\tau_i r_i, \max\{h_i, \tau_i\}\}]_1^n)$ up to a constant factor, where $r_i$ is defined in Def. 4.5.*

*Proof.* Let us fix an iteration and take $k \in [K]$. It is sufficient to find a time required to send enough compressed vectors such that the inequality

$$4 \left(\sum_{i \in [n]\,:\,l_i > 0} \left(\sum_{j=1}^{l_i} \frac{\omega}{l_i^2 m_{ij}} + \sum_{j=1}^{l_i} \frac{\omega\sigma^2}{l_i^3 m_{ij}\varepsilon} + \frac{\sigma^2}{l_i\varepsilon}\right)^{-1}\right)^{-1} \leq 1$$

holds. As soon as this inequality holds, the algorithm stops the loop in Line 6 from Alg. 7. Then the upper bound on the time complexity equals to the number of iterations $\times$ the upper bound on the time of each iteration. The previous inequality is equivalent to

$$\mathcal{H} := \sum_{i \in [n]\,:\,l_i > 0} \frac{1}{\sum_{j=1}^{l_i} \frac{4\omega}{l_i^2 m_{ij}} + \sum_{j=1}^{l_i} \frac{4\omega\sigma^2}{l_i^3 m_{ij}\varepsilon} + \frac{4\sigma^2}{l_i\varepsilon}} \geq 1. \tag{50}$$

Let us show that

$$t' := 128 \times t^*(\omega, \sigma^2/\varepsilon, \max\{h_1, \tau_1\}, \min\{\tau_1 r_1, \max\{h_1, \tau_1\}\}, \ldots, \max\{h_n, \tau_n\}, \min\{\tau_n r_n, \max\{h_n, \tau_n\}\})$$

is a sufficient time such that (50) holds.

By the definition of the equilibrium time $t^*$, in order to apply this mapping, we first have to find a permutation $\pi$ that sorts the input pairs $(\max\{h_i, \tau_i\}, \min\{\tau_i r_i, \max\{h_i, \tau_i\}\})$ by

$$\max\{\max\{h_i, \tau_i\}, \min\{\tau_i r_i, \max\{h_i, \tau_i\}\}\}.$$

This term equals to $\max\{h_i, \tau_i\}$. Without loss of generality, we assume that the sequence $\max\{h_i, \tau_i\}$ is sorted, thus $\pi_i = i$ for all $i \in [n]$. Therefore, we have

$$t' = 128 \min_{j \in [n]} \max\{\max\{h_j, \tau_j\}, s^*(j)\}, \tag{51}$$

where $s^*(j)$ is the solution of the equation

$$\left( \sum_{i=1}^{j} \left( 2\min\{\tau_i r_i, \max\{h_i, \tau_i\}\}\omega + \frac{4\min\{\tau_i r_i, \max\{h_i, \tau_i\}\}\max\{h_i, \tau_i\}\sigma^2\omega}{s\varepsilon} + \frac{2\max\{h_i, \tau_i\}\sigma^2}{\varepsilon} \right)^{-1} \right)^{-1} = s \tag{52}$$

for all $j \in [n]$.

Let us define $j^*$ as the smallest by index minimizer in (51). Then

$$t' = 128 \max\{\max\{h_{j^*}, \tau_{j^*}\}, s^*(j^*)\}.$$

Assume that $l_i$ is the number of iterations (the number of calculated stochastic gradients) that the $i^{\text{th}}$ worker does by the time $t'$. Since $t' \geq 4\max\{h_{j^*}, \tau_{j^*}\}$ and the workers are sorted by $\max\{h_i, \tau_i\}$, we have $t' \geq 2(h_i + \tau_i)$ for all $i \leq j^*$. Therefore, for all $i \leq j^*$, the $i^{\text{th}}$ worker will have time to calculate and send at least one compressed vector, i.e., $l_i \geq 1$ for $i \leq j^*$.

Next, the $i^{\text{th}}$ worker requires at most $\tau_i$ seconds to send a compressed vector and it waits for at least one calculated gradient. Consider that the computation time of the $j^{\text{th}}$ stochastic gradient in the $k^{\text{th}}$ iteration equals to $h_{ij}^k$. Thus $\frac{t'}{2} \leq \sum_{j=1}^{l_i} (h_{ij}^k + \tau_i)$. Indeed, if $\frac{t'}{2} > \sum_{j=1}^{l_i} (h_{ij}^k + \tau_i)$, then the $i^{\text{th}}$ worker will have time to calculate and send at least one more compressed vector because $\frac{t'}{2} \geq 2\max\{h_i, \tau_i\} \geq h_i + \tau_i$ for all $i \leq j^*$. It would contradict the definition of $l_i$. Therefore, we have

$$\frac{t'}{2} \leq \sum_{j=1}^{l_i} \left( h_{ij}^k + \tau_i \right) \leq l_i \max_{j \in [l_i]} h_{ij}^k + l_i \tau_i$$

and

$$l_i \geq \frac{t'}{2 \left( \max_{j \in [l_i]} h_{ij}^k + \tau_i \right)}. \tag{53}$$

At the same time, by the definition of $l_i$, we have

$$\frac{t'}{h_{\min}} \geq l_i,$$

because $h_{\min} > 0$ is the smallest possible calculating time. Therefore, we have

$$l_i \leq l_{\max} := \left\lceil \frac{t_{\max}}{h_{\min}} \right\rceil,$$

where $t_{\max}$ is defined in Def. 4.5 ($t_{\max} \geq t'$). Since $l_i \geq 1$ for all $i \leq j^*$, we get

$$
\mathcal{H} := \sum_{i \in [n]\,:\,l_i > 0} \left( \sum_{j=1}^{l_i} \frac{4\omega}{l_i^2 m_{ij}} + \sum_{j=1}^{l_i} \frac{4\omega\sigma^2}{l_i^3 m_{ij}\varepsilon} + \frac{4\sigma^2}{l_i\varepsilon} \right)^{-1}
$$

$$
\geq \sum_{i=1}^{j^*} \left( \underbrace{\sum_{j=1}^{l_i} \frac{4\omega}{l_i^2 m_{ij}} + \sum_{j=1}^{l_i} \frac{4\omega\sigma^2}{l_i^3 m_{ij}\varepsilon}}_{A} + \underbrace{\frac{4\sigma^2}{l_i\varepsilon}}_{B} \right)^{-1}. \tag{54}
$$

Using (53), we obtain

$$
B := \frac{4\sigma^2}{\varepsilon l_i} \leq \frac{8\sigma^2 \left( \max\limits_{j \in [l_i]} h_{ij}^k + \tau_i \right)}{\varepsilon t'} \leq \frac{8\sigma^2 \left( h_i + \tau_i \right)}{\varepsilon t'} \leq \frac{16\sigma^2 \max\{h_i, \tau_i\}}{\varepsilon t'}.
$$

For every $j^{\text{th}}$ stochastic gradient, the $i^{\text{th}}$ worker sends at least one compressed vector or $\left\lfloor \frac{h_{ij}^k}{\tau_i} \right\rfloor$ compressed vectors because it is possible that $\tau_i \leq h_{ij}^k$, then the worker will have time to send more than one compressed vector. Therefore, we have

$$
m_{ij} \geq \max \left\{ \left\lfloor \frac{h_{ij}^k}{\tau_i} \right\rfloor, 1 \right\} \geq \max \left\{ \frac{h_{ij}^k}{2\tau_i}, 1 \right\} \geq \max \left\{ \frac{\min\limits_{j \in [l_i]} h_{ij}^k}{2\tau_i}, 1 \right\}.
$$

and

$$
\frac{1}{l_i} \sum_{j=1}^{l_i} \frac{1}{m_{ij}} \leq \frac{2}{l_i} \sum_{j=1}^{l_i} \min \left\{ \frac{\tau_i}{\min\limits_{j \in [l_i]} h_{ij}^k}, 1 \right\} = 2 \min \left\{ \frac{\tau_i}{\min\limits_{j \in [l_i]} h_{ij}^k}, 1 \right\}.
$$

Using the last inequality and (53), we get

$$
\frac{1}{l_i^2} \sum_{j=1}^{l_i} \frac{1}{m_{ij}} \leq \frac{4 \left( \max\limits_{j \in [l_i]} h_{ij}^k + \tau_i \right)}{t'} \min \left\{ \frac{\tau_i}{\min\limits_{j \in [l_i]} h_{ij}^k}, 1 \right\}
$$

$$
\leq \frac{8 \max\{\max\limits_{j \in [l_i]} h_{ij}^k, \tau_i\}}{t'} \min \left\{ \frac{\tau_i}{\min\limits_{j \in [l_i]} h_{ij}^k}, 1 \right\}
$$

$$
= \underbrace{\min \left\{ \frac{8\tau_i}{t'} \times \frac{\max\{\max\limits_{j \in [l_i]} h_{ij}^k, \tau_i\}}{\min\limits_{j \in [l_i]} h_{ij}^k}, \frac{8 \max\{\max\limits_{j \in [l_i]} h_{ij}^k, \tau_i\}}{t'} \right\}}_{T}.
$$

It is clear that $T \leq \frac{8 \max\{\max\limits_{j \in [l_i]} h_{ij}^k, \tau_i\}}{t'}$. If $\tau_i < \min\limits_{j \in [l_i]} h_{ij}^k$, then $T = \frac{8\tau_i}{t'} \times \frac{\max\limits_{j \in [l_i]} h_{ij}^k}{\min\limits_{j \in [l_i]} h_{ij}^k}$. If $\tau_i \geq \min\limits_{j \in [l_i]} h_{ij}^k$

and $\tau_i < \max\limits_{j \in [l_i]} h_{ij}^k$, then $T = \frac{8 \max\limits_{j \in [l_i]} h_{ij}^k}{t'} \leq \frac{8\tau_i}{t'} \times \frac{\max\limits_{j \in [l_i]} h_{ij}^k}{\min\limits_{j \in [l_i]} h_{ij}^k}$. If $\tau_i \geq \max\limits_{j \in [l_i]} h_{ij}^k$, then $T = \frac{8\tau_i}{t'} \leq$

$\frac{8\tau_i}{t'} \times \frac{\max\limits_{j \in [l_i]} h_{ij}^k}{\min\limits_{j \in [l_i]} h_{ij}^k}$.

Thus, we have

$$
\frac{1}{l_i^2} \sum_{j=1}^{l_i} \frac{1}{m_{ij}} \leq \min \left\{ \frac{8\tau_i}{t'} \times \frac{\max\limits_{j \in [l_i]} h_{ij}^k}{\min\limits_{j \in [l_i]} h_{ij}^k}, \frac{8 \max\{\max\limits_{j \in [l_i]} h_{ij}^k, \tau_i\}}{t'} \right\}
$$

$$\leq \min \left\{ \frac{8\tau_i}{t'} \times \frac{\sup_{j\in[l_{\max}]} h_{ij}^k}{\inf_{j\in[l_{\max}]} h_{ij}^k}, \frac{8\max\{\max_{j\in[l_i]} h_{ij}^k, \tau_i\}}{t'} \right\}$$

$$\leq \min \left\{ \frac{8\tau_i r_i}{t'}, \frac{8\max\{h_i, \tau_i\}}{t'} \right\},$$

where we use the definition of $r_i$. Using the last inequality and $l_i \geq \frac{t'}{4\max\{h_i, \tau_i\}}$, we have

$$A := 4 \sum_{j=1}^{l_i} \frac{\omega}{m_{ij} l_i^2} + 4 \sum_{j=1}^{l_i} \frac{\omega\sigma^2}{m_{ij}\varepsilon l_i^3}$$

$$\leq \frac{32\omega \min\{\tau_i r_i, \max\{h_i, \tau_i\}\}}{t'} + \frac{32\omega\sigma^2 \min\{\tau_i r_i, \max\{h_i, \tau_i\}\}}{\varepsilon t' l_i}$$

$$\leq \frac{32\omega \min\{\tau_i r_i, \max\{h_i, \tau_i\}\}}{t'} + \frac{128\omega\sigma^2 \min\{\tau_i r_i, \max\{h_i, \tau_i\}\} \max\{h_i, \tau_i\}}{\varepsilon (t')^2},$$

where we use (53) and $\max_{j\in[l_i]} h_{ij}^k \leq h_i$. One can substitute the bounds on $A$ and $B$ to (54) and obtain

$$\mathcal{H} \geq \frac{1}{128} \sum_{i=1}^{j^*} \left( \frac{\omega \min\{\tau_i r_i, \max\{h_i, \tau_i\}\}}{t'} + \frac{\omega\sigma^2 \min\{\tau_i r_i, \max\{h_i, \tau_i\}\} \max\{h_i, \tau_i\}}{\varepsilon (t')^2} + \frac{\sigma^2 \max\{h_i, \tau_i\}}{\varepsilon t'} \right)^{-1}.$$

Note that $t' \geq 128 s^*(j^*)$, thus

$$\mathcal{H} \geq \sum_{i=1}^{j^*} \left( \frac{\omega \min\{\tau_i r_i, \max\{h_i, \tau_i\}\}}{s^*(j^*)} + \frac{\omega\sigma^2 \min\{\tau_i r_i, \max\{h_i, \tau_i\}\} \max\{h_i, \tau_i\}}{\varepsilon (s^*(j^*))^2} + \frac{\sigma^2 \max\{h_i, \tau_i\}}{\varepsilon s^*(j^*)} \right)^{-1}$$

$$\geq \sum_{i=1}^{j^*} \left( \frac{2\omega \min\{\tau_i r_i, \max\{h_i, \tau_i\}\}}{s^*(j^*)} + \frac{4\omega\sigma^2 \min\{\tau_i r_i, \max\{h_i, \tau_i\}\} \max\{h_i, \tau_i\}}{\varepsilon (s^*(j^*))^2} + \frac{2\sigma^2 \max\{h_i, \tau_i\}}{\varepsilon s^*(j^*)} \right)^{-1}$$

$$= s^*(j^*) \times \sum_{i=1}^{j^*} \left( 2\omega \min\{\tau_i r_i, \max\{h_i, \tau_i\}\} + \frac{4\omega\sigma^2 \min\{\tau_i r_i, \max\{h_i, \tau_i\}\} \max\{h_i, \tau_i\}}{\varepsilon s^*(j^*)} + \frac{2\sigma^2 \max\{h_i, \tau_i\}}{\varepsilon} \right)^{-1}.$$

It is left to use the definition of $s^*(j^*)$ (see (52)) to obtain that $\mathcal{H} \geq s^*(j^*) \times \frac{1}{s^*(j^*)} = 1$.

It means that after at most $t'$ seconds, we can ensure that the algorithm will finish the loop in Line 6 from Alg. 7. In the view of Theorem M.1, the time complexity is less or equal to $K \times t'$. $\qquad\square$

## O  Construction of the Lower Bound

We prove the lower bound by generalizing the *time multiple oracles protocol* from (Tyurin and Richtárik, 2023c). Note that in the classical approaches (Nemirovskij and Yudin, 1983; Carmon et al., 2020; Arjevani et al., 2022; Nesterov, 2018), the researchers bound the *number of oracle calls* required to find an $\varepsilon$–solution. Our approach is based on the idea from (Tyurin and Richtárik, 2023c), where the authors propose to bound the *time* required to find an $\varepsilon$–solution. We refer to a detailed explanation to (Tyurin and Richtárik, 2023c)[Sections 3-6].

First, we define an oracle that emulates the process of computing stochastic gradients or the process of sending a compressed vector (Tyurin and Richtárik, 2023c)[Section 4]:

$$O_\tau^{g,\mathcal{D}} : \underbrace{\mathbb{R}_{\geq 0}}_{\text{time}} \times \underbrace{\mathbb{R}^d}_{\text{point}} \times \underbrace{\{0,1\}}_{\text{control}} \times \underbrace{(\mathbb{R}_{\geq 0} \times \mathbb{R}^d \times \{0,1\})}_{\text{input state}} \to \underbrace{(\mathbb{R}_{\geq 0} \times \mathbb{R}^d \times \{0,1\})}_{\text{output state}} \times \mathbb{R}^d$$

such that
$$O_\tau^{g,\mathcal{D}}(t, x, c, (s_t, s_x, s_q)) = \begin{cases} ((t, x, 1), & 0), & c = 1, s_q = 0, \\ ((s_t, s_x, 1), & 0), & c = 1, s_q = 1, t < s_t + \tau, \\ ((0,0,0), & g(s_x; \xi)), & c = 1, s_q = 1, t \geq s_t + \tau, \\ ((0,0,0), & 0), & c = 0, \end{cases}$$

(55)

where $\xi \sim \mathcal{D}$, $g$ is an arbitrary mapping such that $g : \mathbb{R}^d \times \mathbb{S} \to \mathbb{R}^d$, and $\mathbb{S}$ is the sample space of a distribution $\mathcal{D}$. Next, we define the *time multiple oracles protocol with compression*:

---

**Protocol 9** Time Multiple Oracles Protocol with Compression

---

1: **Input:** function(s) $f \in \mathcal{F}$, computation oracles $(O_1, ..., O_n) \in \mathcal{O}(f)$, communication oracles $(\hat{\mathcal{C}}_1, ..., \hat{\mathcal{C}}_n) \in \mathcal{U}$, algorithm $A = \{(B^k, N_1^k, \ldots, N_n^k)\}_{k=0}^{\infty} \in \mathcal{A}$
2: $s_i^{\nabla f, 0} = s_i^{\mathcal{C}, 0} = 0$ for all $i \in [n]$
3: **for** $k = 0, \ldots, \infty$ **do**
4:     $(t^{k+1}, i^{k+1}, c^{\nabla f, k+1}, c^{\mathcal{C}, k+1}, x^k) = B^k(g^1, \ldots, g^k)$,                   $\triangleright t^{k+1} \geq t^k$
5:     $(s_{i^{k+1}}^{\nabla f, k+1}, g_{i^{k+1}}^{k+1}) = O_{i^{k+1}}(t^{k+1}, x^k, c^{\nabla f, k+1}, s_{i^{k+1}}^{\nabla f, k})$
       $\forall j \neq i^{k+1} : s_j^{\nabla f, k+1} = s_j^{\nabla f, k}, \quad g_j^{k+1} = 0$
6:     $g_{\text{pre}}^{k+1} = N_{i^{k+1}}^k(g^1, \ldots, g^k, g_{i^{k+1}}^1, \ldots, g_{i^{k+1}}^{k+1})$,
7:     $(s_{i^{k+1}}^{\mathcal{C}, k+1}, g^{k+1}) = \hat{\mathcal{C}}_{i^{k+1}}(t^{k+1}, g_{\text{pre}}^{k+1}, c^{\mathcal{C}, k+1}, s_{i^{k+1}}^{\mathcal{C}, k})$
8: **end for**

---

In this protocol, the server via $B^k$ returns a new point $x^k$, and broadcasts it to the $i^{k+1}$th worker. Then, the worker calls the oracle $O_{i^{k+1}}$ that calculates stochastic gradients. Next, the oracle returns the vector $g_{i^{k+1}}^{k+1}$, and the worker processes it with $N_{i^{k+1}}^k$. Finally, the worker sends $g_{\text{pre}}^{k+1}$ to the oracle $\hat{\mathcal{C}}_{i^{k+1}}$ that sends compressed vectors to the server. Using the parameters $c^{\nabla f, k+1}$ and $c^{\mathcal{C}, k+1}$, it can decide if it wants to start/stop the process of a gradient calculation and the process of communicating a compressed vector (See Sec. F in (Tyurin and Richtárik, 2023c)). As far as we know, all centralized distributed optimization methods can be described by Protocol 9, including Minibatch SGD, QSGD, Asynchronous SGD, Rennala SGD, and Shadowheart SGD.

We consider the standard function class from the optimization literature (Nesterov, 2018; Arjevani et al., 2022; Carmon et al., 2020):

**Definition O.1** (Function Class $\mathcal{F}_{\Delta, L}$). We assume that a function $f : \mathbb{R}^d \to \mathbb{R}$ is differentiable, $\|\nabla f(x) - \nabla f(y)\| \leq L \|x - y\| \quad \forall x, y \in \mathbb{R}^d$, (L-smooth) and $f(0) - \inf_{x \in \mathbb{R}^d} f(x) \leq \Delta$ ($\Delta$-bounded). The set of all functions with such properties we denote by $\mathcal{F}_{\Delta, L}$.

Next, we define the class of algorithms that we analyze.

**Definition O.2** (Algorithm Class $\mathcal{A}_{\text{zr}}$). Let us consider Protocol 9. We say that the sequence of tuples of mappings $A = \{(B^k, N_1^k, \ldots, N_n^k)\}_{k=0}^{\infty}$ is a zero-respecting algorithm, if

1. $B^k : \underbrace{\mathbb{R}^d \times \cdots \times \mathbb{R}^d}_{k \text{ times}} \to \mathbb{R}_{\geq 0} \times \mathbb{N} \times \mathbb{N} \times \mathbb{N} \times \mathbb{R}^d$ for all $k \geq 1$, and $B^0 \in \mathbb{R}_{\geq 0} \times \mathbb{N} \times \mathbb{N} \times$
$\mathbb{N} \times \mathbb{R}^d$.

2. For all $k \geq 1$ and and $g^1, \ldots, g^k \in \mathbb{R}^d$, $t^{k+1} \geq t^k$, where $t^{k+1}$ and $t^k$ are defined as $(t^{k+1}, \ldots) = B^k(g^1, \ldots, g^k)$ and $(t^k, \ldots) = B^{k-1}(g^1, \ldots, g^{k-1})$.

3. $N_i^k : \underbrace{\mathbb{R}^d \times \cdots \times \mathbb{R}^d}_{k \text{ times}} \times \underbrace{\mathbb{R}^d \times \cdots \times \mathbb{R}^d}_{k+1 \text{ times}} \to \mathbb{R}^d$ for all $k \geq 0$ and for all $i \in [n]$.

4. $\text{supp}\left(x^k\right) \subseteq \bigcup_{j=1}^k \text{supp}\left(g^j\right)$, and $\text{supp}\left(g_{\text{pre}}^{k+1}\right) \subseteq \bigcup_{j=1}^k \text{supp}\left(g^j\right) \bigcup_{j=1}^{k+1} \text{supp}\left(g_{i^{k+1}}^j\right)$, for all $k \in \mathbb{N}_0$, where $\text{supp}(x) := \{i \in [d] \mid x_i \neq 0\}$.

The set of all algorithms with this properties we define as $\mathcal{A}_{\text{zr}}$.

The properties 1 and 3 are only required to define the domains of the mappings. The property 4 ensures that these mappings are *zero-respecting* (Arjevani et al., 2022). The property 2 is explained in (Tyurin and Richtárik, 2023c)[Section 4, Definition 4.1]. It ensures that our algorithm does not "travel into the past".

The following oracle class is the same as in (Tyurin and Richtárik, 2023c). For any $f \in \mathcal{F}_{\Delta, L}$, it returns $n$ oracles that require $h_1, \ldots, h_n$ seconds to calculate a stochastic gradient. These oracles emulate the real behavior where the workers have different processing times.

**Definition O.3** (Computation Oracle Class $\mathcal{O}_{h_1,\ldots,h_n}^{\sigma^2}$). Let us consider an oracle class such that, for any $f \in \mathcal{F}_{\Delta,L}$, it returns oracles $O_i = O_{h_i}^{\nabla f, \mathcal{D}_i^{\nabla f}}$ for all $i \in [n]$, where $\nabla f(x; \xi)$ is an unbiased $\sigma^2$-variance-bounded mapping (see Assumption 1.3). The oracles $O_{h_i}^{\nabla f, \mathcal{D}_i^{\nabla f}}$ are defined in (55). We define such oracle class as $\mathcal{O}_{h_1,\ldots,h_n}^{\sigma^2}$.

The following oracle class emulates the behavior of compressors. It returns $n$ oracles that require $\tau_1, \ldots, \tau_n$ seconds to send a compressed vector to the server.

**Definition O.4** (Communication Oracle Class $\mathcal{U}_{\tau_1,\ldots,\tau_n}^{\omega}$). Let us consider an oracle class such that, it returns oracles $\hat{\mathcal{C}}_i = O_{\tau_i}^{\mathcal{C}, \mathcal{D}_i^{\mathcal{C}}}$ for all $i \in [n]$, where $\mathcal{C}$ is an unbiased compressor with a parameter $\omega$, i.e., $\mathcal{C} \in \mathbb{U}(\omega)$ (see Def. 2.1). The oracles $O_{\tau_i}^{\mathcal{C}, \mathcal{D}_i^{\mathcal{C}}}$ are defined in (55). We define such oracle class as $\mathcal{U}_{\tau_1,\ldots,\tau_n}^{\omega}$.

Finally, we present our lower bound theorem:

**Theorem O.5.** *Let us consider Protocol 9. We take any $h_i > 0$, $\tau_i > 0$ for all $i \in [n]$, $\omega \geq 0$, $L, \Delta, \varepsilon, \sigma^2 > 0$ such that $\varepsilon < c_1 L \Delta$ and $\omega + 1 \leq T$,[11] where $T = \left\lfloor \frac{\Delta L}{c_2 \varepsilon} \right\rfloor$ is the dimension of the construction. For any algorithm $A \in \mathcal{A}_{zr}$, there exists a function $f \in \mathcal{F}_{\Delta,L}$, computation oracles $(O_1, \ldots, O_n) \in \mathcal{O}_{h_1,\ldots,h_n}^{\sigma^2}(f)$, and communication oracles $(\hat{\mathcal{C}}_1, \ldots, \hat{\mathcal{C}}_n) \in \mathcal{U}_{\tau_1,\ldots,\tau_n}^{\omega},$[12] such that $\mathbb{E}\left[\inf_{k \in S_t} \left\| \nabla f(x^k) \right\|^2 \right] > \varepsilon$, where $S_t := \left\{ k \in \mathbb{N}_0 \,|\, t^k \leq t \right\}$ and*

$$
t = c_3 \times \frac{L\Delta}{\varepsilon} \times t^*(\omega, \sigma^2/\varepsilon, h_1, \tau_1, \ldots, h_n, \tau_n).
$$

*The quantities $c_1$, $c_2$, and $c_3$ are universal constants. The sequences $x^k$ and $t^k$ are defined in Protocol 9.*

# P   Proof of Theorem O.5

## P.1   The "Worst Case" Function

Let us consider the "worst case" function, which is a standard function to obtain lower bounds in the nonconvex world. We define

$$
\mathrm{prog}(x) := \max\{i \geq 0 \,|\, x_i \neq 0\} \quad (x_0 \equiv 1).
$$

In our proofs, we use the construction from (Carmon et al., 2020; Arjevani et al., 2022). For any $T \in \mathbb{N}$, the authors define

$$
F_T(x) := -\Psi(1)\Phi(x_1) + \sum_{i=2}^{T} \left[ \Psi(-x_{i-1})\Phi(-x_i) - \Psi(x_{i-1})\Phi(x_i) \right], \tag{56}
$$

where

$$
\Psi(x) = \begin{cases} 0, & x \leq 1/2, \\ \exp\left(1 - \frac{1}{(2x-1)^2}\right), & x \geq 1/2, \end{cases} \quad \text{and} \quad \Phi(x) = \sqrt{e} \int_{-\infty}^{x} e^{-\frac{1}{2}t^2} dt.
$$

The main property of the function $F_T(x)$ is that its gradients are large unless $\mathrm{prog}(x) \geq T$.

**Lemma P.1** (Carmon et al. (2020); Arjevani et al. (2022)). *The function $F_T$ satisfies:*

1. $F_T(0) - \inf_{x \in \mathbb{R}^T} F_T(x) \leq \Delta^0 T$, *where* $\Delta^0 = 12$.

---

[11]We can avoid this constraint using a slightly different construction of a compressor. However, the number of non-zero returned values by the new construction is random. See Sec. P.3.

[12]The function $f$ defined on $\mathbb{R}^T$ with $T = \Theta\left(L\Delta/\varepsilon\right)$, and the constructed compressor $\mathcal{C}$ preserves only $K = \lceil T/\omega+1 \rceil$ non-zero coordinates.

2. *The function $F_T$ is $l_1$–smooth, where $l_1 = 152$.*

3. *For all $x \in \mathbb{R}^T$, $\|\nabla F_T(x)\|_\infty \leq \gamma_\infty$, where $\gamma_\infty = 23$.*

4. *For all $x \in \mathbb{R}^T$, $\mathrm{prog}(\nabla F_T(x)) \leq \mathrm{prog}(x) + 1$.*

5. *For all $x \in \mathbb{R}^T$, if $\mathrm{prog}(x) < T$, then $\|\nabla F_T(x)\| > 1$.*

**Theorem O.5.** *Let us consider Protocol 9. We take any $h_i > 0$, $\tau_i > 0$ for all $i \in [n]$, $\omega \geq 0$, $L, \Delta, \varepsilon, \sigma^2 > 0$ such that $\varepsilon < c_1 L \Delta$ and $\omega + 1 \leq T$,[13] where $T = \left\lfloor \frac{\Delta L}{c_2 \varepsilon} \right\rfloor$ is the dimension of the construction. For any algorithm $A \in \mathcal{A}_{\mathrm{zr}}$, there exists a function $f \in \mathcal{F}_{\Delta, L}$, computation oracles $(O_1, \ldots, O_n) \in \mathcal{O}_{h_1, \ldots, h_n}^{\sigma^2}(f)$, and communication oracles $(\hat{\mathcal{C}}_1, \ldots, \hat{\mathcal{C}}_n) \in \mathcal{U}_{\tau_1, \ldots, \tau_n}^\omega$,[14] such that $\mathbb{E}\left[\inf_{k \in S_t} \left\|\nabla f(x^k)\right\|^2\right] > \varepsilon$, where $S_t := \left\{k \in \mathbb{N}_0 \,|\, t^k \leq t\right\}$ and*

$$t = c_3 \times \frac{L\Delta}{\varepsilon} \times t^*(\omega, \sigma^2/\varepsilon, h_1, \tau_1, \ldots, h_n, \tau_n).$$

*The quantities $c_1$, $c_2$, and $c_3$ are universal constants. The sequences $x^k$ and $t^k$ are defined in Protocol 9.*

*Proof.*

Without loss of generality, we assume that the workers are sorted by $\max\{h_i, \tau_i\}$ : $\max\{h_1, \tau_1\} \leq \cdots \leq \max\{h_n, \tau_n\}$.
(**Step 1**: $f \in \mathcal{F}_{\Delta, L}$)
Let us fix $\lambda > 0$ and take the function $f(x) := \frac{L\lambda^2}{l_1} F_T\left(\frac{x}{\lambda}\right)$, where the function $F_T$ is defined in Sec. P.1. Tyurin and Richtárik (2023c)[Sec. D.2, Proof of Thm. 6.4] show that the function $f$ is $L$–smooth and $f(0) - \inf_{x \in \mathbb{R}^T} f(x) \leq \Delta$ if

$$T = \left\lfloor \frac{\Delta l_1}{L\lambda^2 \Delta^0} \right\rfloor.$$

Thus, we have $f \in \mathcal{F}_{\Delta, L}$.

(**Step 2**: Oracle Class) Let us construct a stochastic gradient mapping. For our lower bound, we take

$$[\nabla f(x; \xi)]_j := \nabla_j f(x) \left(1 + \mathbb{1}[j > \mathrm{prog}(x)] \left(\frac{\xi}{p} - 1\right)\right) \quad \forall x \in \mathbb{R}^T,$$

and $\mathcal{D}_i^{\nabla f} = Bernoulli(p)$ for all $i \in [n]$, where $p \in (0, 1]$. We denote $[x]_j$ as the $j^{\mathrm{th}}$ index of a vector $x \in \mathbb{R}^T$. Let us take

$$p = \min\left\{\frac{L^2 \lambda^2 \gamma_\infty^2}{\sigma^2 l_1^2}, 1\right\}.$$

Then Tyurin and Richtárik (2023c)[Sec. D.2, Proof of Thm. 6.4] show that this mapping is unbiased and $\sigma^2$-variance-bounded.

(**Step 3**: Compression Operator) In our construction, we take the $\mathrm{Rand}K$ compressor (outputs $K$ random values of an input vector *without replacement*, scaled by $T/K$ (Def. D.1)). From Theorem D.2, we know that $\mathcal{C}$ is unbiased and $\frac{T}{K} - 1$–variance bounded, i.e.,

$$\mathbb{E}_S[\mathcal{C}(x; S)] = x, \qquad \mathbb{E}_S\left[\|\mathcal{C}(x; S) - x\|^2\right] \leq \left(\frac{T}{K} - 1\right)\|x\|^2, \qquad \forall x \in \mathbb{R}^T,$$

where

$$[\mathcal{C}(x; S)]_j := \begin{cases} \frac{T}{K} x_j, & j \in S, \\ 0, & j \notin S, \end{cases} \quad \forall j \in [T].$$

---

[13]We can avoid this constraint using a slightly different construction of a compressor. However, the number of non-zero returned values by the new construction is random. See Sec. P.3.

[14]The function $f$ defined on $\mathbb{R}^T$ with $T = \Theta(L\Delta/\varepsilon)$, and the constructed compressor $\mathcal{C}$ preserves only $K = \lceil T/\omega + 1 \rceil$ non-zero coordinates.

and $S$ is an uniformly random subset of $[T]$ without replacement. It is sufficient to take $K = \left\lceil \frac{T}{\omega+1} \right\rceil$ to ensure that $\mathcal{C} \in \mathbb{U}(\omega)$. Let us define $p_\omega := \frac{K}{T}$. We take mutually independent distributions $\mathcal{D}_i^\mathcal{C}$ that generate random subsets $S$ described above.

(**Step 4**: Analysis of Protocol)
Let us take
$$\lambda = \frac{\sqrt{2\varepsilon} l_1}{L}$$
to ensure that $\|\nabla f(x)\|^2 = \frac{L^2\lambda^2}{l_1^2} \left\| \nabla F_T(\frac{x}{\lambda}) \right\|^2 > 2\varepsilon \mathbb{1}\left[\text{prog}(x) < T\right]$ for all $x \in \mathbb{R}^T$, where we use Lemma P.1. Thus
$$T = \left\lfloor \frac{\Delta L}{2\varepsilon l_1 \Delta^0} \right\rfloor \tag{57}$$
and
$$p = \min\left\{ \frac{2\varepsilon \gamma_\infty^2}{\sigma^2}, 1 \right\}.$$

Protocol 9 generates the sequence $\{x^k\}_{k=0}^\infty$. We have
$$\inf_{k \in S_t} \left\| \nabla f(x^k) \right\|^2 > 2\varepsilon \inf_{k \in S_t} \mathbb{1}\left[\text{prog}(x^k) < T\right]. \tag{58}$$

Using Lemma P.2 with $\delta = 1/2$ and (58), we obtain
$$\mathbb{E}\left[ \inf_{k \in S_t} \left\| \nabla f(x^k) \right\|^2 \right] \geq 2\varepsilon \mathbb{P}\left( \inf_{k \in S_t} \mathbb{1}\left[\text{prog}(x^k) < T\right] \geq 1 \right) > \varepsilon$$
for
$$t \leq \frac{1}{48} t^* \left( \frac{T}{K}, \max\left\{ \frac{\sigma^2}{2\varepsilon \gamma_\infty^2}, 1 \right\}, h_1, \tau_1, \ldots, h_n, \tau_n \right) \left( \frac{\Delta L}{8\varepsilon l_1 \Delta^0} - 1 \right).$$

By the assumption of the theorem, we have $\omega + 1 \leq T$. Therefore, we get the series of inequalities:
$$\frac{T}{K} = \frac{T}{\left\lceil \frac{T}{\omega+1} \right\rceil} \geq \frac{\omega+1}{2}$$

and, using Properties 6.1 and E.1, we have
$$t^* \left( \frac{T}{K}, \max\left\{ \frac{\sigma^2}{2\varepsilon \gamma_\infty^2}, 1 \right\}, h_1, \tau_1, \ldots, h_n, \tau_n \right)$$
$$\geq t^* \left( \frac{\omega+1}{2}, \max\left\{ \frac{\sigma^2}{2\varepsilon \gamma_\infty^2}, 1 \right\}, h_1, \tau_1, \ldots, h_n, \tau_n \right)$$
$$\geq t^* \left( \frac{1}{2\gamma_\infty^2} \times \omega, \frac{1}{2\gamma_\infty^2} \times \frac{\sigma^2}{\varepsilon}, h_1, \tau_1, \ldots, h_n, \tau_n \right)$$
$$\geq \frac{1}{2\gamma_\infty^2} \times t^* \left( \omega, \sigma^2/\varepsilon, h_1, \tau_1, \ldots, h_n, \tau_n \right),$$

Thus, we can take
$$t = \frac{1}{48} \times \frac{1}{2\gamma_\infty^2} \times t^* \left( \omega, \sigma^2/\varepsilon, h_1, \tau_1, \ldots, h_n, \tau_n \right) \left( \frac{\Delta L}{8\varepsilon l_1 \Delta^0} - 1 \right).$$

$\square$

### P.1.1   Proof of Lemma P.2

**Lemma P.2.** *Let us fix $T, T' \in \mathbb{N}$ such that $T \leq T'$, consider Protocol 9 with an algorithm $A \in \mathcal{A}_{\text{zr}}$, a differentiable function $f : \mathbb{R}^{T'} \to \mathbb{R}$ such that $\text{prog}(\nabla f(x)) \leq \text{prog}(x) + 1$ for all $x \in \text{domain}(f)$.*

1. *We take stochastic oracles $O_i = O_{h_i}^{\nabla f, \mathcal{D}_i^{\nabla f}}$ with the distributions $\mathcal{D}_i^{\nabla f} = Bernoulli(p_\sigma)$, $p_\sigma \in (0, 1]$, $h_i > 0$, and the mappings*

$$[\nabla f(x; \xi)]_j = \nabla_j f(x) \left( 1 + \mathbb{1}\left[ j > \mathrm{prog}(x) \right] \left( \frac{\xi}{p_\sigma} - 1 \right) \right) \quad \forall x \in \mathbb{R}^{T'}, \forall \xi \in \{0, 1\}, \forall j \in [T].$$
(59)

2. *We take compression oracles $\hat{\mathcal{C}}_i = O_{\tau_i}^{\mathcal{C}, \mathcal{D}_i^{\mathcal{C}}}$ with the distributions $\mathcal{D}_i^{\mathcal{C}} = uniform(K, T')$ (= "uniformly random subset of $[T']$ of the size $K$ without replacement") and the mappings*

$$[\mathcal{C}(x; S)]_j := \begin{cases} \frac{T'}{K} x_j, & j \in S, \\ 0, & j \notin S, \end{cases} \quad \forall x \in \mathbb{R}^{T'}, \forall S \subseteq [n], \forall j \in [T'], \tag{60}$$

$\tau_i > 0$. *We define $p_\omega := \frac{K}{T'}$. We assume that the workers are sorted by $\max\{h_m, \tau_m\}$ : $\max\{h_1, \tau_1\} \leq \cdots \leq \max\{h_n, \tau_n\}$. With probability not less than $1 - \delta$, the following inequality holds:*

$$\inf_{k \in S_t} \mathbb{1}\left[ \mathrm{prog}(x^k) < T \right] \geq 1$$

*for*

$$t \leq \frac{1}{48} t^* (1/p_\omega, 1/p_\sigma, h_1, \tau_1, \ldots, h_n, \tau_n) \left( \frac{T}{2} + \log \delta \right),$$

*where $S_t := \left\{ k \in \mathbb{N}_0 \,|\, t^k \leq t \right\}$, the iterates $t^k$ and $x^k$ are defined in Protocol 9, and $t^*$ is the equilibrium time from Def. 3.1.*

*Proof.*
**(Part 1): The Construction of Random Variables.**
Let us fix $t \geq 0$ and define the smallest index $k(i)$ of the sequence when the progress $\mathrm{prog}(x^{k(i)})$ equals $i$ :

$$k(i) := \inf \left\{ k \in \mathbb{N}_0 \,|\, i = \mathrm{prog}(x^k) \right\} \in \mathbb{N}_0 \cup \{\infty\}.$$

If $\inf_{k \in S_t} \mathbb{1}\left[ \mathrm{prog}(x^k) < 1 \right] < 1$ holds, then exists $k \in S_t$ such that $\mathrm{prog}(x^k) = 1$, thus, by the definition of $k(1)$, $t^{k(1)} \leq t^k \leq t$, and $k(1) < \infty$. Note that $t^{k(1)}$ is the smallest time when we make progress to the $1^{\text{th}}$ (first) coordinate.

Since $x^0 = 0$ and $A$ is a zero-respecting algorithm, the algorithm can return a vector $x^k$ with a non-zero first coordinate only if some of returned by the stochastic gradients oracles and compression oracles have the first coordinate not equal to zero. The oracles $O_i$ and $\hat{\mathcal{C}}_i$ are constructed in such a way (see (59) and (60)) that they zero out a coordinate based on i.i.d. *bernoulli* and *uniform* trials. According to Protocol 9, even if a stochastic oracle returns a non-zero coordinate, it would not mean that the server will get a non-zero coordinate because a subsequent compression oracle also has to return a non-zero coordinate.

Every time when the oracle (55) evaluates $g(s_x; \xi)$, it draws i.i.d. random variables $\xi \sim \mathcal{D}$. Let us enumerate them:

1. For the stochastic/computation oracles $O_i = O_{h_i}^{\nabla f, \mathcal{D}_i^{\nabla f}}$, we consider the sequence $\{\xi^{m,j}\}_{j=1}^\infty$, where $\xi^{m,j}$ is a *bernoulli* random variable drawn in $j^{\text{th}}$ call of $g(s_x; \xi)$ in the $m^{\text{th}}$ worker in Line 5 of Protocol 9.

2. For the compression oracles $\hat{\mathcal{C}}_i = O_{\tau_i}^{\mathcal{C}, \mathcal{D}_i^{\mathcal{C}}}$, we consider the sequence $\{S^{m,j}\}_{j=1}^\infty$, where $S^{m,j}$ is a *uniform* random variable drawn in $j^{\text{th}}$ call of $g(s_x; \xi)$ in the $m^{\text{th}}$ worker in Line 7 of Protocol 9.

Let us define the following useful random variables based on previous definitions. We define

$$\eta_{m,j} := \begin{cases} \inf\{i \,|\, \xi^{m,(i+b_{m,j}^\eta - 1)} = 1 \text{ and } i \in \mathbb{N}\} \in \mathbb{N} \cup \{\infty\}, & b_{m,j}^\eta < \infty, \\ \infty, & b_{m,j}^\eta = \infty, \end{cases} \quad \forall j \in \{1, \ldots, T\},$$
(61)

$$\mu_{m,j} := \begin{cases} \inf\{i \,|\, j \in S^{m,(i+e^{\mu}_{m,j}-1)} \text{ and } i \in \mathbb{N}\} \in \mathbb{N} \cup \{\infty\}, & e^{\mu}_{m,j} < \infty, \\ \infty, & e^{\mu}_{m,j} = \infty, \end{cases} \quad \forall j \in \{1, \dots, T\},$$

(62)

where

1. For all $m \in [n]$, $j \geq 1$, $b^{\eta}_{m,j} \in \mathbb{N} \cup \{\infty\}$ is the first index of the sequence $\{\xi^{m,j}\}_{j=1}^{\infty}$ that started calculating in (55) in the stochastic oracle $O_m$ in or after the iteration $k(j-1)$.

2. For all $m \in [n]$, $j \geq 1$, $b^{\mu}_{m,j} \in \mathbb{N} \cup \{\infty\}$ is the first index of the sequence $\{S^{m,j}\}_{j=1}^{\infty}$ that started calculating in (55) in the compression oracle $\hat{\mathcal{C}}_m$ in or after the iteration $k(j-1)$.

3. For all $m \in [n]$, $j \geq 1$, if $b^{\eta}_{m,j} = \infty$, then $e^{\eta}_{m,j} = \infty$. For all $m \in [n]$, $j \geq 1$, if $b^{\eta}_{m,j} < \infty$, then $e^{\eta}_{m,j} \in \mathbb{N} \cup \{\infty\}$ is the first index of the sequence $\{\xi^{m,j}\}_{j=1}^{\infty}$ that started calculating in (55) in the stochastic oracle $O_m$ in or after the iteration $k(j-1)$ and the first moment when $\xi^{m,(i+b^{\eta}_{m,j}-1)} = 1$ for some $i \geq 1$.

4. For all $m \in [n]$, $j \geq 1$, if $b^{\eta}_{m,j} = \infty$, then $e^{\mu}_{m,j} = \infty$. For all $m \in [n]$, $j \geq 1$, if $b^{\eta}_{m,j} < \infty$, then $e^{\mu}_{m,j} \in \mathbb{N} \cup \{\infty\}$ is the first index of the sequence $\{S^{m,j}\}_{j=1}^{\infty}$ that started calculating in (55) in the compression oracle $\hat{\mathcal{C}}_m$ in or after the iteration $k(j-1)$ and the first moment when $\xi^{m,(i+b^{\eta}_{m,j}-1)} = 1$ for some $i \geq 1$.

It possible that such indexes do not exist, then we take $b^{\eta}_{m,j} = \infty$, $b^{\mu}_{m,j} = \infty$, $e^{\eta}_{m,j} = \infty$, or $e^{\mu}_{m,j} = \infty$, accordingly. By the construction, $e^{\eta}_{m,j} \geq b^{\eta}_{m,j}$ and $e^{\mu}_{m,j} \geq b^{\mu}_{m,j}$.

Let us clarify the definitions. At the beginning $x_0 = 0$, thus $k(0) = 0$. It would mean that the first index of the sequence $\{\xi^{m,j}\}_{j=1}^{\infty}$, when the worker evaluates $g(s_x; \xi)$ in (55), simply equals $b^{\eta}_{m,1} = 1$ or $b^{\eta}_{m,1} = \infty$ (by the definition, it equals to $\infty$ if the oracle was never called). Assume that $b^{\eta}_{m,1} = 1$, then $\eta_{m,1} = \inf\{i \,|\, \xi^{m,i} = 1 \text{ and } i \in \mathbb{N}\}$ is the first time when the oracle draws a "successful" random *bernoulli* trial. This random variable is distributed according to the *geometric* distribution. Then, $e^{\eta}_{m,1}$ can be equal to $\eta_{m,1} + 1$ or $\infty$. At some (random) iteration $k(1)$, the algorithm $A$ can get the first non-zero coordinate through $g^k$, then $b^{\mu}_{m,2}$ is the next index of the sequence $\{\xi^{m,j}\}_{j=1}^{\infty}$ that started calculating in (55).[15]

The server gets a non-zero coordinate if at least one worker draws a successful *bernoulli* trial, and this coordinate belongs to a set generated by the *uniform* distribution. It takes $h_i$ seconds to generate one *bernoulli* trial and $\tau_i$ seconds to generate one *uniform* trial.

Then, if $:= \inf_{k \in S_t} \mathbb{1}\left[\text{prog}(x^k) < 1\right] < 1$ holds, then

$$\hat{t}_1 := \min_{m \in [n]} \{h_m \eta_{m,1} + \tau_m \mu_{m,1}\} \leq t^{k(1)}.$$

because $h_m \eta_{m,1} + \tau_m \mu_{m,1}$ is the time required to generate $\eta_{m,1}$ *bernoulli* and $\mu_{m,1}$ *uniform* trials. In other words, the algorithm can not progress to the next coordinate before the moment when at least one worker generates "successful" *bernoulli* and *uniform* trials.

Using the same reasoning, $t^{k(j)} \geq t^{k(j-1)} + \hat{t}_j$, where

$$\hat{t}_j := \min_{m \in [n]} \{h_m \eta_{m,j} + \tau_m \mu_{m,j}\}.$$

---

[15]Let us consider an example with the $m^{\text{th}}$ worker. Assume that it starts calculations of stochastic gradients and with $\eta_{m,1} = 5$ (as an example) it gets a "successful" trial: $\xi^{m,\eta_m,\eta_{m,1}} = \xi^{m,5} = 1$ ($\xi^{m,1} = \cdots = \xi^{m,4} = 0$). And only then, starting with $e^{\mu}_{m,1}$, the compression oracle can get a vector with a non-zero coordinate. Even if there was a previous "successful" trial: $1 \in S^{m,i}$ for some $i < e^{\mu}_{m,1}$. This trial did not return a vector with a non-zero coordinate because the stochastic oracle did not return a vector with a non-zero coordinate by that time. Assume that $e^{\mu}_{m,1} = 10$, then the server waits for $\mu_{m,1} = \inf\{i \,|\, 1 \in S^{m,(i+e^{\mu}_{m,1}-1)}\}$. Assume that $\mu_{m,1} = 7$, then the time moment, when $1 \in S^{m,(\mu_{m,1}+e^{\mu}_{m,1}-1)} = 1 \in S^{m,(7+10-1)}$, is the first possible moment when the $m^{\text{th}}$ worker can send a vector with a non-zero coordinate to the server.

Combining the observations, if $\inf_{k \in S_t} \mathbb{1}\left[\text{prog}(x^k) < T\right] < 1$ holds, then $\sum_{j=1}^{T} \min_{j \in [n]} (h_i \eta_{i,j} + \tau_i \mu_{i,j}) \le t^{k(T)} \le t$. Thus

$$\mathbb{P}\left(\inf_{k \in S_t} \mathbb{1}\left[\text{prog}(x^k) < T\right] < 1\right) \le \mathbb{P}\left(\sum_{i=1}^{T} \hat{t}_i \le t\right) = \mathbb{P}\left(\sum_{i=1}^{T} \min_{j \in [n]} (h_i \eta_{i,j} + \tau_i \mu_{i,j}) \le t\right) \quad \forall t \ge 0.$$

### (Part 2): The Chernoff Method

Let us fix $s \ge 0$ and $\hat{t} \ge 0$. Using the Chernoff method, we have

$$\mathbb{P}\left(\sum_{i=1}^{T} \hat{t}_i \le \hat{t}\right) = \mathbb{P}\left(-s\left(\sum_{i=1}^{T} \hat{t}_i\right) \ge -s\hat{t}\right) = \mathbb{P}\left(\exp\left(-s\sum_{i=1}^{T} \hat{t}_i\right) \ge \exp\left(-s\hat{t}\right)\right)$$

$$\le e^{s\hat{t}} \mathbb{E}\left[\exp\left(-s\sum_{i=1}^{T} \hat{t}_i\right)\right]. \tag{63}$$

Let us bound the expected value separately. For all $j \in [T]$, let us define $\mathcal{G}_j$ as the $\sigma$–algebra generated by random variables

$$b_{1,j}^\eta, \ldots, b_{n,j}^\eta$$
$$\xi^{1,1}, \xi^{1,2}, \ldots, \xi^{1,b_{1,j}^\eta - 1},$$
$$\ldots$$
$$\xi^{n,1}, \xi^{n,2}, \ldots, \xi^{1,b_{n,j}^\eta - 1}, \tag{64}$$
$$b_{1,j}^\mu, \ldots, b_{n,j}^\mu,$$
$$S^{1,1}, S^{1,2}, \ldots, S^{1,b_{1,j}^\mu - 1},$$
$$\ldots$$
$$S^{n,1}, S^{n,2}, \ldots, S^{n,b_{n,j}^\mu - 1}.$$

The $\sigma$–algebra $\mathcal{G}_T$ contains all information about the random variables before the moment when $\text{prog}(x^k) = T - 1$. Then, we have

$$\mathbb{E}\left[\exp\left(-s\sum_{i=1}^{T} \hat{t}_i\right)\right] = \mathbb{E}\left[\mathbb{E}\left[\exp\left(-s\sum_{i=1}^{T-1} \hat{t}_i - s\hat{t}_T\right) \Big| \mathcal{G}_T\right]\right].$$

Note that if the random variables from (64) are "fixed," then $\hat{t}_i$ is deterministic for all $i \in [T - 1]$ because $\hat{t}_i$ is a deterministic function of (64), and does not depend on other subsequent random variables.

Let us show it using a contradiction proof. Without the loss of generality, assume that $\hat{t}_{T-1}$ depends on $\xi^{1,b_{1,T}^\eta} \notin$ (64). By the definition of $b_{1,T}^\eta$, it would mean that the first time when the server can get a vector $g^k$ with a non-zero coordinate in the index $T - 1$ is *after* the moment $t^{k(T-1)}$. We get a contradiction since $t^{k(T-1)}$ is the first time when the algorithm return an iterate with a non-zero coordinate in the index $T - 1$.

Thus, $\hat{t}_i$ is $\mathcal{G}_T$–measurable for all $i \in [T - 1]$ and

$$\mathbb{E}\left[\exp\left(-s\sum_{i=1}^{T} \hat{t}_i\right)\right] = \mathbb{E}\left[\exp\left(-s\sum_{i=1}^{T-1} \hat{t}_i\right) \mathbb{E}\left[\exp\left(-s\hat{t}_T\right) \big| \mathcal{G}_T\right]\right]. \tag{65}$$

Let us fix $t' \ge 0$, then, since $\hat{t}_T \ge 0$, we have

$$\mathbb{E}\left[e^{-s\hat{t}_T} \Big| \mathcal{G}_T\right] = \mathbb{E}\left[e^{-s\hat{t}_T} \Big| \hat{t}_T \le t', \mathcal{G}_T\right] \mathbb{P}\left(\hat{t}_T \le t' | \mathcal{G}_T\right) + \mathbb{E}\left[e^{-s\hat{t}_T} \Big| \hat{t}_T > t', \mathcal{G}_T\right]\left(1 - \mathbb{P}\left(\hat{t}_T \le t' | \mathcal{G}_T\right)\right)$$

$$\le \mathbb{P}\left(\hat{t}_T \le t' | \mathcal{G}_T\right) + e^{-st'}\left(1 - \mathbb{P}\left(\hat{t}_T \le t' | \mathcal{G}_T\right)\right). \tag{66}$$

We now use the result of the following lemma that we prove separately.

**Lemma P.3.** *Using the notations from the proof of Lemma P.2, we have*

$$\mathbb{P}\left(\hat{t}_j \le t' | \mathcal{G}_j\right) \le 1 - \prod_{m=1}^{n}\left(1 - \left(1 - (1-p_\omega)^{\lfloor\frac{t'}{\tau_m}\rfloor}\right)\left(1 - (1-p_\sigma)^{\lfloor\frac{t'}{h_m}\rfloor}\right)\right) \quad (67)$$

*for all $j \in [T]$.*

Let us temporarily define

$$p' := 1 - \prod_{m=1}^{n}\left(1 - \left(1 - (1-p_\omega)^{\lfloor\frac{t'}{\tau_m}\rfloor}\right)\left(1 - (1-p_\sigma)^{\lfloor\frac{t'}{h_m}\rfloor}\right)\right)$$

We substitute (67) to (66) and (65) to obtain

$$\mathbb{E}\left[\exp\left(-s\sum_{i=1}^{T}\hat{t}_i\right)\right] \le \left(p' + e^{-st'}(1-p')\right)\mathbb{E}\left[\exp\left(-s\sum_{i=1}^{T-1}\hat{t}_i\right)\right] \le \left(p' + e^{-st'}(1-p')\right)^{T}.$$

Next, using (63), we get

$$\mathbb{P}\left(\sum_{i=1}^{T}\hat{t}_i \le \hat{t}\right) \le e^{s\hat{t}}\left(p' + e^{-st'}(1-p')\right)^{T} = e^{s\hat{t}-st'T}\left(1 + \left(e^{st'}-1\right)p'\right)^{T}.$$

Let us take $s = 1/t'$, and get

$$\mathbb{P}\left(\sum_{i=1}^{T}\hat{t}_i \le \hat{t}\right) \le e^{\hat{t}/t'-T}\left(1 + (e-1)p'\right)^{T} \le e^{\hat{t}/t'-T+2p'T}. \quad (68)$$

Let us recall the definition of $p'$ :

$$p' := 1 - \prod_{m=1}^{n}\left(1 - \left(1 - (1-p_\omega)^{\lfloor\frac{t'}{\tau_m}\rfloor}\right)\left(1 - (1-p_\sigma)^{\lfloor\frac{t'}{h_m}\rfloor}\right)\right) = 1 - \prod_{m=1}^{n}(1-q_m)$$

where we define $q_m := \left(1 - (1-p_\omega)^{\lfloor\frac{t'}{\tau_m}\rfloor}\right)\left(1 - (1-p_\sigma)^{\lfloor\frac{t'}{h_m}\rfloor}\right) \in [0, 1]$. Using Lemma C.1, we have

$$p' \le \sum_{m=1}^{n} q_m.$$

Using the inequality[16] $1 - (1-p)^m \le pm$ for all $p \in [0, 1]$ and $m \in \mathbb{N}_0$, we can get the following three inequalities:

$$q_m \le 1 - (1-p_\sigma)^{\lfloor\frac{t'}{h_m}\rfloor} \le p_\sigma\left\lfloor\frac{t'}{h_m}\right\rfloor,$$

$$q_m \le 1 - (1-p_\omega)^{\lfloor\frac{t'}{\tau_m}\rfloor} \le p_\omega\left\lfloor\frac{t'}{\tau_m}\right\rfloor,$$

and

$$q_m \le \left(1 - (1-p_\omega)^{\lfloor\frac{t'}{\tau_m}\rfloor}\right)\left(1 - (1-p_\sigma)^{\lfloor\frac{t'}{h_m}\rfloor}\right) \le p_\omega\left\lfloor\frac{t'}{\tau_m}\right\rfloor p_\sigma\left\lfloor\frac{t'}{h_m}\right\rfloor.$$

Therefore, we get

$$q_m \le \min\left\{p_\sigma\left\lfloor\frac{t'}{h_m}\right\rfloor, p_\omega\left\lfloor\frac{t'}{\tau_m}\right\rfloor, p_\omega p_\sigma\left\lfloor\frac{t'}{\tau_m}\right\rfloor\left\lfloor\frac{t'}{h_m}\right\rfloor\right\}$$

---

[16]We implicitly assume that $1 - (1-p)^m = 0$ if $p = 1$ and $m = 0$. See footnote [17] for the details.

and

$$p' \leq \sum_{m=1}^{n} \min\left\{ p_\sigma \left\lfloor \frac{t'}{h_m} \right\rfloor, p_\omega \left\lfloor \frac{t'}{\tau_m} \right\rfloor, p_\omega p_\sigma \left\lfloor \frac{t'}{\tau_m} \right\rfloor \left\lfloor \frac{t'}{h_m} \right\rfloor \right\}. \tag{69}$$

Now, we have to take the right $t'$. Assume that $s^*(j)$ is the solution of

$$\left( \sum_{m=1}^{j} \frac{1}{\frac{2\tau_m}{p_\omega} + \frac{4\tau_m h_m}{p_\omega p_\sigma s} + \frac{2h_m}{p_\sigma}} \right)^{-1} = s \tag{70}$$

and

$$j^* = \inf\left\{ j \in [n] \mid s^*(j) < \max\{h_{j+1}, \tau_{j+1}\} \right\} \in [n], \qquad \max\{h_{n+1}, \tau_{n+1}\} \equiv \infty.$$

Then we take $t' = \frac{1}{24} s^*(j^*)$. If $j^* = 1$, then

$$s^*(j^*) = \left( \frac{1}{\frac{2\tau_1}{p_\omega} + \frac{4\tau_1 h_1}{p_\omega p_\sigma s^*(j^*)} + \frac{2h_1}{p_\sigma}} \right)^{-1} \geq \left( \frac{1}{\frac{2\tau_1}{p_\omega} + \frac{2h_1}{p_\sigma}} \right)^{-1} \geq \left( \frac{1}{2\tau_1 + 2h_1} \right)^{-1} \geq \frac{1}{2} \max\{h_1, \tau_1\}.$$

Otherwise, if $j^* > 1$, since $s^*(j^*) \leq s^*(j^* - 1)$, we have

$$s^*(j^*) = \left( \sum_{m=1}^{j^*} \frac{1}{\frac{2\tau_m}{p_\omega} + \frac{4\tau_m h_m}{p_\omega p_\sigma s^*(j^*)} + \frac{2h_m}{p_\sigma}} \right)^{-1}$$

$$\geq \left( \sum_{m=1}^{j^*-1} \frac{1}{\frac{2\tau_m}{p_\omega} + \frac{4\tau_m h_m}{p_\omega p_\sigma s^*(j^*-1)} + \frac{2h_m}{p_\sigma}} + \frac{1}{\frac{2\tau_{j^*}}{p_\omega} + \frac{2h_{j^*}}{p_\sigma}} \right)^{-1}$$

$$\geq \left( \sum_{m=1}^{j^*-1} \frac{1}{\frac{2\tau_m}{p_\omega} + \frac{4\tau_m h_m}{p_\omega p_\sigma s^*(j^*-1)} + \frac{2h_m}{p_\sigma}} + \frac{1}{\max\{h_{j^*}, \tau_{j^*}\}} \right)^{-1}.$$

By the definitions of $s^*(j^* - 1)$ and $j^*$, we have

$$\left( \sum_{m=1}^{j^*-1} \frac{1}{\frac{2\tau_m}{p_\omega} + \frac{4\tau_m h_m}{p_\omega p_\sigma s^*(j^*-1)} + \frac{2h_m}{p_\sigma}} \right)^{-1} = s^*(j^* - 1) \geq \max\{h_{j^*}, \tau_{j^*}\}.$$

Therefore, we get

$$s^*(j^*) \geq \left( \frac{1}{\max\{h_{j^*}, \tau_{j^*}\}} + \frac{1}{\max\{h_{j^*}, \tau_{j^*}\}} \right)^{-1} = \frac{1}{2} \max\{h_{j^*}, \tau_{j^*}\}.$$

Using $s^*(j^*) \geq \frac{1}{2} \max\{h_{j^*}, \tau_{j^*}\}$, we obtain

$$t' = \frac{1}{24} \max\left\{ \frac{1}{2} \max\{h_{j^*}, \tau_{j^*}\}, s^*(j^*) \right\} \geq \frac{1}{48} \min_{j \in [n]} \max\{\max\{h_j, \tau_j\}, s^*(j)\}. \tag{71}$$

We use the last inequality later. Let us return to the inequality (69). Using the definition of $j^*$, we obtain $\left\lfloor \frac{t'}{\tau_m} \right\rfloor = 0$ or $\left\lfloor \frac{t'}{h_m} \right\rfloor = 0$ for all $m > j^*$ and

$$p' \leq \sum_{m=1}^{n} \min\left\{ p_\sigma \left\lfloor \frac{t'}{h_m} \right\rfloor, p_\omega \left\lfloor \frac{t'}{\tau_m} \right\rfloor, p_\omega p_\sigma \left\lfloor \frac{t'}{\tau_m} \right\rfloor \left\lfloor \frac{t'}{h_m} \right\rfloor \right\}$$

$$= \sum_{m=1}^{j^*} \min\left\{ p_\sigma \left\lfloor \frac{t'}{h_m} \right\rfloor, p_\omega \left\lfloor \frac{t'}{\tau_m} \right\rfloor, p_\omega p_\sigma \left\lfloor \frac{t'}{\tau_m} \right\rfloor \left\lfloor \frac{t'}{h_m} \right\rfloor \right\}$$

$$\leq \sum_{m=1}^{j^*} \min\left\{ \frac{p_\sigma t'}{h_m}, \frac{p_\omega t'}{\tau_m}, \frac{p_\omega p_\sigma t'^2}{\tau_m h_m} \right\},$$

where we use $\lfloor x \rfloor \le x$ for all $x \ge 0$. Using $\max\{x, y, z\} \ge \frac{1}{3}(x + y + z)$ for all $x, y, z \ge 0$, we have

$$p' \le \sum_{m=1}^{j^*} \left( \max \left\{ \frac{h_m}{p_\sigma t'}, \frac{\tau_m}{p_\omega t'}, \frac{\tau_m h_m}{p_\omega p_\sigma t'^2} \right\} \right)^{-1} \le \sum_{m=1}^{j^*} \frac{3}{\frac{\tau_m}{p_\omega t'} + \frac{\tau_m h_m}{p_\omega p_\sigma t'^2} + \frac{h_m}{p_\sigma t'}}.$$

Since $t' = \frac{1}{24} s^*(j^*)$, we obtain

$$p' \le \sum_{m=1}^{j^*} \frac{3}{\frac{24\tau_m}{p_\omega s^*(j^*)} + \frac{24^2 \tau_m h_m}{p_\omega p_\sigma (s^*(j^*))^2} + \frac{24 h_m}{p_\sigma s^*(j^*)}} \le \frac{1}{4} \sum_{m=1}^{j^*} \frac{1}{\frac{2\tau_m}{p_\omega s^*(j^*)} + \frac{4\tau_m h_m}{p_\omega p_\sigma (s^*(j^*))^2} + \frac{2h_m}{p_\sigma s^*(j^*)}}.$$

Note that $s^*(j^*)$ is the solution of (70), thus, we get

$$p' \le \frac{1}{4}.$$

Substituting this inequality to (68), we obtain

$$\mathbb{P}\left( \sum_{i=1}^{T} \hat{t}_i \le \hat{t} \right) \le e^{\hat{t}/t' - \frac{T}{2}}.$$

For $\hat{t} \le t' \left( \frac{T}{2} + \log \delta \right)$, we have

$$\mathbb{P}\left( \sum_{i=1}^{T} \hat{t}_i \le \hat{t} \right) \le \delta.$$

Recall that the definition of $t'$. Using (71), we have

$$t' \ge \frac{1}{48} \min_{j \in [n]} \max\{\max\{h_j, \tau_j\}, s^*(j)\}.$$

The last term equals to the *equilibrium time* $t^*(1/p_\omega, 1/p_\sigma, h_1, \tau_1, \ldots, h_n, \tau_n)$ from Def. 3.1 since the pairs $(h_j, \tau_j)$ are sorted by $\max\{h_j, \tau_j\}$. Thus, we obtain

$$t' \ge \frac{1}{48} t^*(1/p_\omega, 1/p_\sigma, h_1, \tau_1, \ldots, h_n, \tau_n).$$

Finally, we obtain

$$\mathbb{P}\left( \inf_{k \in S_t} \mathbb{1}\left[ \mathrm{prog}(x^k) < T \right] < 1 \right) \le \mathbb{P}\left( \sum_{i=1}^{T} \hat{t}_i \le t \right) \le \delta \tag{72}$$

for

$$t \le \frac{1}{48} t^*(1/p_\omega, 1/p_\sigma, h_1, \tau_1, \ldots, h_n, \tau_n) \left( \frac{T}{2} + \log \delta \right).$$

$\square$

## P.2 Proof of Lemma P.3

**Lemma P.3.** *Using the notations from the proof of Lemma P.2, we have*

$$\mathbb{P}\left( \hat{t}_j \le t' \middle| \mathcal{G}_j \right) \le 1 - \prod_{m=1}^{n} \left( 1 - \left( 1 - (1 - p_\omega)^{\lfloor \frac{t'}{\tau_m} \rfloor} \right) \left( 1 - (1 - p_\sigma)^{\lfloor \frac{t'}{h_m} \rfloor} \right) \right) \tag{67}$$

*for all $j \in [T]$.*

*Proof.* We prove the result for $j = T$. The proofs for the cases $1 \le j < T$ are the same. We consider the conditional probability

$$\mathbb{P}\left( \hat{t}_T \le t' \middle| \mathcal{G}_T \right) = \mathbb{P}\left( \min_{m \in [n]} \{h_m \eta_{m,T} + \tau_m \mu_{m,T}\} \le t' \middle| \mathcal{G}_T \right).$$

Let us consider the $\sigma$–algebra $\mathcal{H}_T$ generated by (64) with $j = T$ and

$$
\begin{aligned}
&e^{\eta}_{1,T}, \ldots, e^{\eta}_{n,T} \\
&\xi^{1,1}, \xi^{1,2}, \ldots, \xi^{1,e^{\eta}_{1,T}-1}, \\
&\ldots \\
&\xi^{n,1}, \xi^{n,2}, \ldots, \xi^{1,e^{\eta}_{n,T}-1}, \\
&e^{\mu}_{1,T}, \ldots, e^{\mu}_{n,T}, \\
&S^{1,1}, S^{1,2}, \ldots, S^{1,e^{\mu}_{1,T}-1}, \\
&\ldots \\
&S^{n,1}, S^{n,2}, \ldots, S^{n,e^{\mu}_{n,T}-1}.
\end{aligned}
\tag{73}
$$

By the construction, $e^{\eta}_{m,T} \geq b^{\eta}_{m,T}$ and $e^{\mu}_{m,T} \geq b^{\mu}_{m,T}$. Since $\mathcal{G}_T \subseteq \mathcal{H}_T$, we have

$$
\mathbb{P}\left(\hat{t}_T \leq t' | \mathcal{G}_T\right) = \mathbb{E}\left[\mathbb{P}\left(\min_{m \in [n]} \{h_m \eta_{m,T} + \tau_m \mu_{m,T}\} \leq t' \Big| \mathcal{H}_T\right)\Big| \mathcal{G}_T\right].
$$

Since $\mathcal{G}_T \subseteq \mathcal{H}_T$, then $b^{\mu}_{m,T}$ is $\mathcal{H}_T$–measurable. By the definition of $e^{\eta}_{m,T}$, $\eta_{m,T}$ are $\mathcal{H}_T$-measurable. Let us show it using a contradiction proof. Without the loss of generality, assume that $\eta_{m,T}$ depends on $\xi^{1,e^{\eta}_{m,T}} \notin$ (73). It would mean that the first time, when $\xi^{m,i} = 1$ after the iteration $k(T-1)$, happens with $i \geq e^{\eta}_{m,T}$. At the same time, by the definition of $e^{\eta}_{m,T}$, there exists $i < e^{\eta}_{m,T}$ such that $\xi^{m,i} = 1$ calculated after the iteration $k(T-1)$. We get a contradiction. Given $\mathcal{H}_T$, $\mu_{m,T}$ are mutually independent since $\{S^{m,j}\}_{j=1}^{\infty}$ are mutually independent and $e^{\mu}_{m,T}$ are $\mathcal{H}_T$-measurable. Therefore, we have

$$
\begin{aligned}
\mathbb{P}\left(\hat{t}_T \leq t' | \mathcal{G}_T\right) &= \mathbb{E}\left[\mathbb{P}\left(\min_{m \in [n]} \{h_m \eta_{m,T} + \tau_m \mu_{m,T}\} \leq t' \Big| \mathcal{H}_T\right)\Big| \mathcal{G}_T\right] \\
&= 1 - \mathbb{E}\left[\mathbb{P}\left(\bigcap_{m=1}^{n} \{h_m \eta_{m,T} + \tau_m \mu_{m,T} > t'\} \Big| \mathcal{H}_T\right)\Big| \mathcal{G}_T\right] \\
&= 1 - \mathbb{E}\left[\prod_{m=1}^{n} \mathbb{P}\left(h_m \eta_{m,T} + \tau_m \mu_{m,T} > t' | \mathcal{H}_T\right)\Big| \mathcal{G}_T\right].
\end{aligned}
\tag{74}
$$

Let us consider the probability $\mathbb{P}\left(h_m \eta_{m,T} + \tau_m \mu_{m,T} > t' | \mathcal{H}_T\right)$:

$$
\begin{aligned}
\mathbb{P}\left(h_m \eta_{m,T} + \tau_m \mu_{m,T} > t' | \mathcal{H}_T\right) &= 1 - \mathbb{P}\left(h_m \eta_{m,T} + \tau_m \mu_{m,T} \leq t' | \mathcal{H}_T\right) \\
&\geq 1 - \mathbb{P}\left(h_m \eta_{m,T} \leq t', \tau_m \mu_{m,T} \leq t' | \mathcal{H}_T\right) \\
&= 1 - \mathbb{E}\left[\mathbb{1}\left[h_m \eta_{m,T} \leq t'\right] \mathbb{1}\left[\tau_m \mu_{m,T} \leq t'\right] | \mathcal{H}_T\right]
\end{aligned}
$$

because the event $\{h_m \eta_{m,T} \leq t'\} \bigcap \{\tau_m \mu_{m,T} \leq t'\}$ follows from $\{h_m \eta_{m,T} + \tau_m \mu_{m,T} \leq t'\}$. Since $\eta_{m,T}$ is $\mathcal{H}_T$–measurable, we have

$$
\begin{aligned}
\mathbb{P}\left(h_m \eta_{m,T} + \tau_m \mu_{m,T} > t' | \mathcal{H}_T\right) &\geq 1 - \mathbb{E}\left[\mathbb{1}\left[\tau_m \mu_{m,T} \leq t'\right] | \mathcal{H}_T\right] \mathbb{1}\left[h_m \eta_{m,T} \leq t'\right] \\
&= 1 - \mathbb{P}\left(\tau_m \mu_{m,T} \leq t' | \mathcal{H}_T\right) \mathbb{1}\left[h_m \eta_{m,T} \leq t'\right].
\end{aligned}
\tag{75}
$$

Let us consider the probability $\mathbb{P}\left(\tau_m \mu_{m,T} \leq t' | \mathcal{H}_T\right)$. Given $\mathcal{H}_T$, if $e^{\mu}_{m,T} = \infty$, then $\mu_{m,T} = \infty$ and $\mathbb{P}\left(\tau_m \mu_{m,T} \leq t' | \mathcal{H}_T\right) = 0$. Otherwise, if $e^{\mu}_{m,T} = e < \infty$, then

$$
\mu_{m,T} = \inf\{i \,|\, j \in S^{m,(i+e-1)} \text{ and } i \in \mathbb{N}\}.
$$

and it is distributed with the *geometric* distribution with $p_\omega$. Thus, we have[17]

$$
\mathbb{P}\left(\tau_m \mu_{m,T} \leq t' | \mathcal{H}_T\right) = \begin{cases} 1 - (1 - p_\omega)^{\left\lfloor \frac{t'}{\tau_m} \right\rfloor}, & e^{\mu}_{m,T} < \infty \\ 0, & e^{\mu}_{m,T} = \infty \end{cases} \leq 1 - (1 - p_\omega)^{\left\lfloor \frac{t'}{\tau_m} \right\rfloor},
$$

---

[17]We implicitly assume that $1 - (1 - p_\omega)^{\left\lfloor \frac{t'}{\tau_m} \right\rfloor} = 0$ if $p_\omega = 1$ and $\left\lfloor \frac{t'}{\tau_m} \right\rfloor = 0$, because if $t' < \tau_m$, then $\mathbb{P}\left(\tau_m \mu_{m,T} \leq t'\right) = 0$ for the r.v. $\mu_{m,T}$ from the *geometric* distribution for all $p_\omega \in (0, 1]$.

because the probability that $j^{\text{th}}$ coordinate belongs to $S^{m,(i+e-1)}$ equals to $p_\omega$. We substitute this inequality to (75) and get

$$\mathbb{P}\left(h_m\eta_{m,T} + \tau_m\mu_{m,T} > t'|\mathcal{H}_T\right) \geq 1 - \left(1 - (1-p_\omega)^{\left\lfloor \frac{t'}{\tau_m} \right\rfloor}\right)\mathbb{1}\left[h_m\eta_{m,T} \leq t'\right].$$

Next, we substitute this inequality to (74) and obtain

$$\mathbb{P}\left(\hat{t}_T \leq t'|\mathcal{G}_T\right) \leq 1 - \mathbb{E}\left[\left.\prod_{m=1}^{n}\left(1 - \left(1 - (1-p_\omega)^{\left\lfloor \frac{t'}{\tau_m} \right\rfloor}\right)\mathbb{1}\left[h_m\eta_{m,T} \leq t'\right]\right)\right|\mathcal{G}_T\right].$$

Given $\mathcal{G}_T$, $\eta_{m,T}$ are independent because $b_{m,T}^\eta$ are $\mathcal{G}_T$–measurable. Thus

$$\mathbb{P}\left(\hat{t}_T \leq t'|\mathcal{G}_T\right) \leq 1 - \prod_{m=1}^{n}\left(1 - \left(1 - (1-p_\omega)^{\left\lfloor \frac{t'}{\tau_m} \right\rfloor}\right)\mathbb{E}\left[\mathbb{1}\left[h_m\eta_{m,T} \leq t'\right]|\mathcal{G}_T\right]\right)$$

$$= 1 - \prod_{m=1}^{n}\left(1 - \left(1 - (1-p_\omega)^{\left\lfloor \frac{t'}{\tau_m} \right\rfloor}\right)\mathbb{P}\left(h_m\eta_{m,T} \leq t'|\mathcal{G}_T\right)\right).$$

Using the same reasoning as with $\mu_{m,T}$, we get

$$\mathbb{P}\left(h_m\eta_{m,T} \leq t'|\mathcal{G}_T\right) = \begin{cases} 1 - (1-p_\sigma)^{\left\lfloor \frac{t'}{h_m} \right\rfloor}, & b_{m,T}^\eta < \infty \\ 0, & b_{m,T}^\eta = \infty \end{cases} \leq 1 - (1-p_\sigma)^{\left\lfloor \frac{t'}{h_m} \right\rfloor}.$$

because, given $\mathcal{G}_T$, $\eta_{m,T}$ equals $\infty$ or a random variable distributed according to the *geometric* distribution with $p_\sigma$. Therefore, we obtain

$$\mathbb{P}\left(\hat{t}_T \leq t'|\mathcal{G}_T\right) \leq 1 - \prod_{m=1}^{n}\left(1 - \left(1 - (1-p_\omega)^{\left\lfloor \frac{t'}{\tau_m} \right\rfloor}\right)\left(1 - (1-p_\sigma)^{\left\lfloor \frac{t'}{h_m} \right\rfloor}\right)\right).$$

$\square$

## P.3 Another Construction

In the proof of Theorem O.5, we could use the following construction:

(**Step 3**: Compression Operator) Let us define $p_\omega := \frac{1}{\omega+1}$. In our construction, we take a compressor that outputs random coordinates of an input vector, scaled by $1/p_\omega$, where each coordinate is taken with the probability $p_\omega$. Each worker has access to the independent compressed realizations of a such compressor. More formally, we assume that

$$[\mathcal{C}(x;S)]_j := \begin{cases} \frac{1}{p_\omega}x_j, & j \in S, \\ 0, & j \notin S, \end{cases} \quad \forall j \in [T],$$

where $S$ is a random subset of $[T]$, where each element from $[T]$ appears with the probability $p_\omega$ *independently*. Then

$$\mathbb{E}_S\left[[\mathcal{C}(x;S)]_j\right] = x_j$$

and

$$\mathbb{E}_S\left[\|\mathcal{C}(x;S)\|^2\right] = \mathbb{E}_S\left[\sum_{j=1}^{T}\mathbb{1}\left[j \in S\right]\frac{1}{p_\omega^2}x_j^2\right] = \sum_{j=1}^{T}\mathbb{P}\left(j \in S\right)\frac{1}{p_\omega^2}x_j^2 = \sum_{j=1}^{T}\frac{1}{p_\omega}x_j^2 = (\omega+1)\|x\|^2.$$

Thus, we have $\mathcal{C} \in \mathbb{U}(\omega)$. We take mutually independent distributions $\mathcal{D}_i^\mathcal{C}$ that generate random subsets $S$ described above.

This construction is also valid and does not require the assumption $\omega + 1 \lesssim {}^{L\Delta}/_\varepsilon$. However, unlike the construction from Theorem O.5, this construction can return a random number of non-zero coordinates.

## Q Experiments

The experiments were prepared in Python. The distributed environment was emulated on machines with Intel(R) Xeon(R) Gold 6226R CPU @ 2.90GHz and 64 cores.

### Q.1 Experiments with Logistic Regression

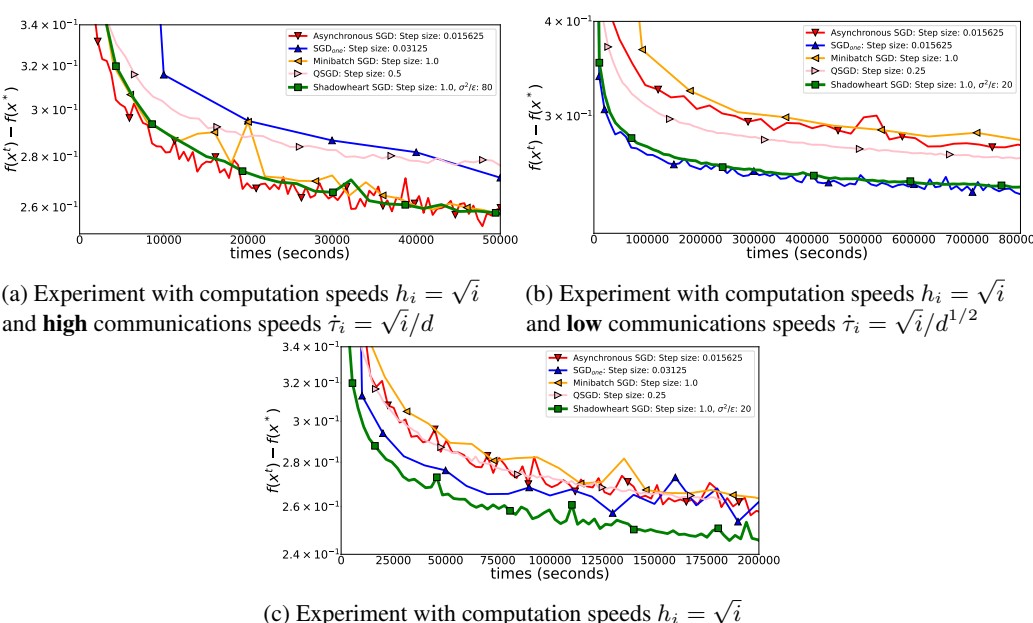

(a) Experiment with computation speeds $h_i = \sqrt{i}$ and **high** communications speeds $\dot{\tau}_i = \sqrt{i}/d$

(b) Experiment with computation speeds $h_i = \sqrt{i}$ and **low** communications speeds $\dot{\tau}_i = \sqrt{i}/d^{1/2}$

(c) Experiment with computation speeds $h_i = \sqrt{i}$ and **medium** communications speeds $\dot{\tau}_i = \sqrt{i}/d^{3/4}$

We start our experiments with a practical setup: a logistic regression problem with the *MNIST* dataset (LeCun et al., 2010). The optimization steps of algorithms are emulated in Python, where we fix the number of workers to $n = 100$, each worker has access to the *MNIST* dataset and sample 4 samples when calculating a stochastic gradient. We compare Shadowheart SGD with QSGD, Asynchronous SGD (we implement the version from (Koloskova et al., 2022)), Minibatch SGD, and SGD_one. SGD_one is the method described in Sec. 7, where SGD is run on the fastest worker locally. In Shadowheart SGD, we fine-tune the parameter $\sigma^2/\varepsilon \in \{1, 5, 10, 20, 30, 40, 80, 120, 150, 200\}$. In all the methods, we also finetune the step sizes. The dimension of the problem in the logistic regression problem is $d = 7850$. In Shadowheart SGD and QSGD, we take RandK with $K = 700$.

We assume that the computations time $h_i$ of a stochastic gradient equals to $\sqrt{i}$ seconds in the $i^{\text{th}}$ worker. We consider three communication time setups, where it takes $\dot{\tau}_i$ seconds to send *one coordinate* from the $i^{\text{th}}$ worker to the server and

1. $\dot{\tau}_i = \sqrt{i}/d$  (High-speed communications),

2. $\dot{\tau}_i = \sqrt{i}/d^{3/4}$  (Medium-speed communications),

3. $\dot{\tau}_i = \sqrt{i}/d^{1/2}$  (Low-speed communications).

In the high-speed regime, the communication between the server and the worker is relatively fast. At the same time, in low-speed regimes, communication is expensive. We are ready to present the results of our experiments in Fig. 1a, 1c, and 1b.

In Figure 1a, one can see that Shadowheart SGD, Asynchronous SGD, and Minibatch SGD are the fastest because it is not expensive to send a non-compressed vector in the "high-speed communications" regime. SGD_one is the slowest since it utilizes only one worker.

Next, we analyze Figure 1b, where Shadowheart SGD and SGD$_{\text{one}}$ have the best performance. SGD$_{\text{one}}$ improves the convergence relative to other methods because the communication speed is much slower than in Figure 1a, and it is expensive to send a non-compressed vector.

One can see that Shadowheart SGD is very robust to all regimes and has one of the best convergence rates in all experiments. Notably, in the "medium-speed communications" regime, where it is still expensive to send a non-compressed vector, our new method converges faster than other baseline methods.

## Q.2 Experiments with quadratic optimization tasks and multiplicative noise

In real machine learning tasks, it is not easy to control noise. Thus, we generated synthetic quadratic optimization tasks where we can control the noise of stochastic gradients. In particular, we consider

$$f(x) = \frac{1}{2}x^\top \mathbf{A}x - b^\top x$$

for all $x \in \mathbb{R}^d$ and take $d = 1000$,

$$\mathbf{A} = \frac{1}{4}\begin{pmatrix} 2 & -1 & & 0 \\ -1 & \ddots & \ddots & \\ & \ddots & \ddots & -1 \\ 0 & & -1 & 2 \end{pmatrix} \in \mathbb{R}^{d \times d} \quad \text{and} \quad b = \frac{1}{4}\begin{bmatrix} -1 \\ 0 \\ \vdots \\ 0 \end{bmatrix} \in \mathbb{R}^d.$$

We consider the following stochastic gradients:

$$[\nabla f(x;\xi)]_j := \nabla_j f(x)\left(1 + \mathbb{1}\left[j > \text{prog}(x)\right]\left(\frac{\xi}{p} - 1\right)\right) \quad \forall x \in \mathbb{R}^d, \tag{76}$$

where $\xi \sim \text{Bernoulli}(p)$ for all $i \in [n]$, and $p \in (0, 1]$. We denote $[x]_j$ as the $j^{\text{th}}$ index of a vector $x \in \mathbb{R}^d$. In our experiments, we take the starting point $x^0 = [\sqrt{d}, 0, \ldots, 0]^\top$ and $p \in \{10^{-3}, 10^{-4}\}$; the smaller $p$ the larger the noise of stochastic gradients.

### Q.2.1 Discussion of the experiments from Sec. Q.2.2

Using this setup, in Figures 2, 3, 4, we fix all parameters except one that we vary to understand the dependencies. In all experiments, we observe that Shadowheart SGD is the most robust to input changes among other *centralized* methods (QSGD, Asynchronous SGD, Minibatch SGD) and can converge significantly faster. At the same time, we observe that SGD$_{\text{one}}$ can be faster than our method in some setups. It happens in the regimes when communication is expensive (see Figure 2a), which is expected and discussed in Sec. 7. Even if communication is expensive, SGD$_{\text{one}}$ starts to slow down relative to other methods when we increase the noise (compare Figures 2b and 2a). The following experiments agree with our theoretical discussion in Sec. 7.

### Q.2.2 Plots

In these experiments, we take $n = 10000$, $p = 10^{-3}$, $h_i = \sqrt{i}$, $\dot{\tau}_i = \sqrt{i}/d^{3/4}$ as base parameters; in each plot, we vary one parameter.

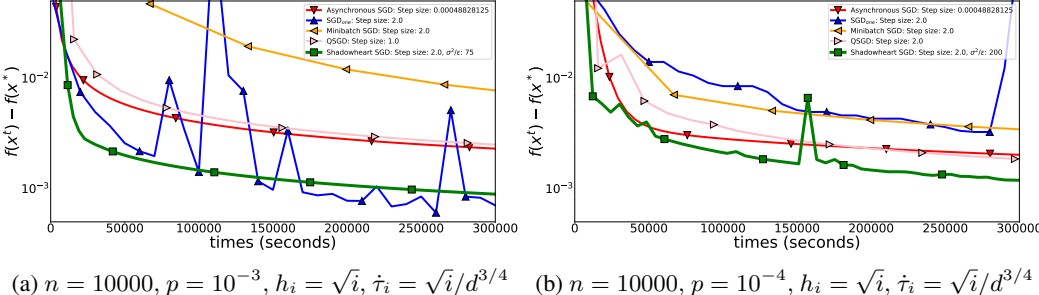

(a) $n = 10000$, $\underline{p = 10^{-3}}$, $h_i = \sqrt{i}$, $\dot{\tau}_i = \sqrt{i}/d^{3/4}$    (b) $n = 10000$, $\underline{p = 10^{-4}}$, $h_i = \sqrt{i}$, $\dot{\tau}_i = \sqrt{i}/d^{3/4}$

Figure 2: SGD$_{\text{one}}$ starts to slow down relative to Shadowheart SGD and other methods when we increase the noise.

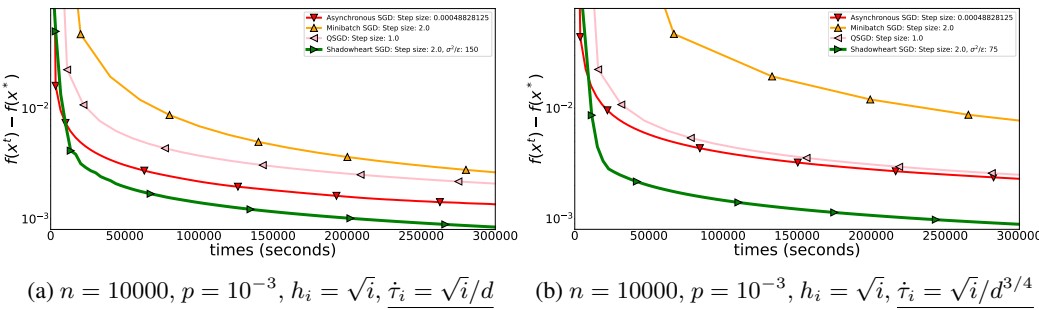

(a) $n = 10000$, $p = 10^{-3}$, $h_i = \sqrt{i}$, $\underline{\dot{\tau}_i = \sqrt{i}/d}$      (b) $n = 10000$, $p = 10^{-3}$, $h_i = \sqrt{i}$, $\underline{\dot{\tau}_i = \sqrt{i}/d^{3/4}}$

Figure 3: The non-compressed methods Asynchronous SGD and Minibatch SGD slow down relative to Shadowheart SGD when we increase the communication times.

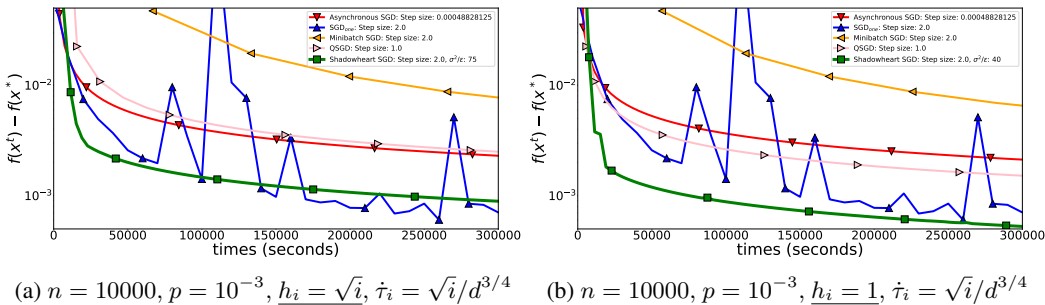

(a) $n = 10000$, $p = 10^{-3}$, $\underline{h_i = \sqrt{i}}$, $\dot{\tau}_i = \sqrt{i}/d^{3/4}$      (b) $n = 10000$, $p = 10^{-3}$, $\underline{h_i = 1}$, $\dot{\tau}_i = \sqrt{i}/d^{3/4}$

Figure 4: Shadowheart SGD improves when we decrease the computation times from $\sqrt{i}$ to $1$.

### Q.3 Experiments with quadratic optimization tasks and additive noise

In this section, we consider the same problem as in Sec. Q.2. However, unlike the *multiplicative* noise, we consider the following *additive* noise:

$$[\nabla f(x; \xi)]_j := \nabla_j f(x) + \zeta \quad \forall x \in \mathbb{R}^d,$$

where $\zeta \sim \mathcal{N}(0, \sigma^2)$ is a sample from the normal distribution. Be default, we take $n = 100$ workers, the dimension $d = 100$, and use the Rand1 compressor ($\omega = d/1 - 1 = 99$), $x_0 = [1, \cdots, 1]^\top$, $\sigma = 10^{-1}$, $\varepsilon = 10^{-4}$; thus, the ratio $\sigma^2/\varepsilon = 10^2$. For all methods, we choose the step sizes in such a way that they converge to the same neighborhood of the stationary point.

In Figure 5, we sample $h_i^k$ and $\dot{\tau}_i^k$ from the uniform distribution $U(0.1, 1)$, hence the communication and computation time vary on each iteration for each client. If we increase the number of clients $n$, Shadowheart SGD improves (Fig. 5) compared to other methods, confirming our theory.

In Figure 6, we can see the similar results with different ratios $\sigma^2/\varepsilon$: Shadowheart SGD is much better when the ratio is large (Fig. 6). On the other hand, when $\sigma^2/\varepsilon$ is small (Fig. 6a) SGD$_{\text{one}}$ can be better because, intuitively, we only need a few workers to find the minimum with a small noise (see also Sec. 7).

Next, we perform a series of experiments with different computation time and communication times ratios. We take $\dot{\tau}_i^k/h_i^k = c$ for all $i \in [n]$, where $c > 0$.

In Figure 7, we take $h_i^k \sim U(0.1, 1)$ and $\dot{\tau}_i^k \sim c \cdot U(0.1, 1)$. Shadowheart SGD is better in the high and medium communication speed regimes (Fig. 7b), when the communication times are not too large. On the other hand, with large $c = 10^2$, Shadowheart SGD spends much time on sending gradients to the server, whereas SGD$_{\text{one}}$ does not spend time on communication and does not compress (Fig. 7c). Similar to Figure 7, we obtain the results with $h_i^k = \sqrt{i}$ and $\dot{\tau}_i^k = c \cdot \sqrt{i}$ in Figure 8.

### Q.3.1 Plots

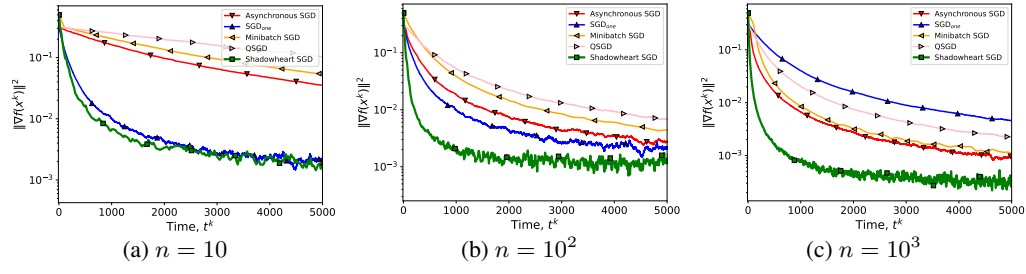

(a) $n = 10$      (b) $n = 10^2$      (c) $n = 10^3$

Figure 5: $h_i^k, \dot\tau_i^k \sim U(0.1, 1)$

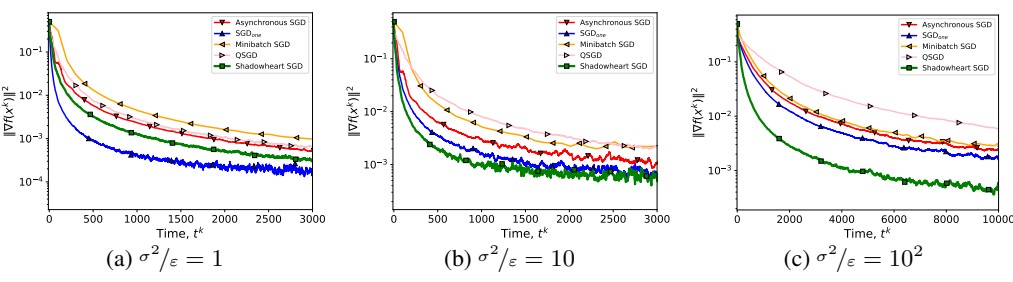

(a) $\sigma^2/\varepsilon = 1$      (b) $\sigma^2/\varepsilon = 10$      (c) $\sigma^2/\varepsilon = 10^2$

Figure 6: $h_i^k, \dot\tau_i^k \sim U(0.1, 1)$

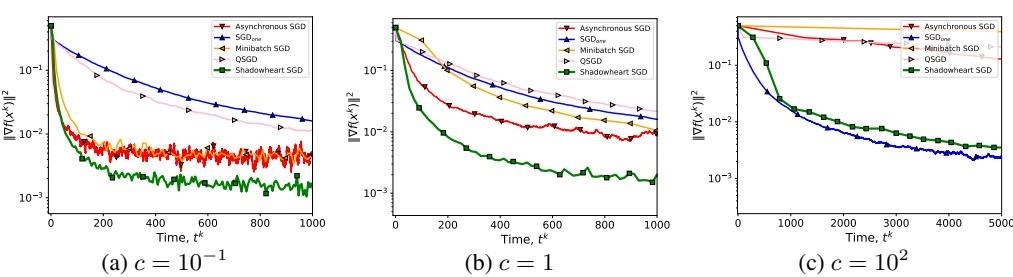

(a) $c = 10^{-1}$      (b) $c = 1$      (c) $c = 10^2$

Figure 7: $h_i^k \sim U(0.1, 1), \dot\tau_i^k \sim c \cdot U(0.1, 1)$

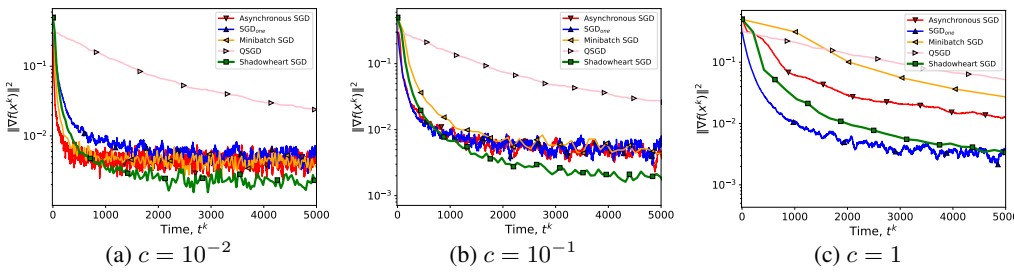

(a) $c = 10^{-2}$      (b) $c = 10^{-1}$      (c) $c = 1$

Figure 8: $h_i^k = \sqrt{i}, \dot\tau_i^k = c \cdot \sqrt{i}$

