# OpenReview forum: "Shadowheart SGD: Distributed Asynchronous SGD with Optimal Time Complexity Under Arbitrary Computation and Communication Heterogeneity"
_NeurIPS.cc/2024/Conference — NeurIPS 2024 poster_

### Official Review · Reviewer_U2d9 · 2024-07-14

**Soundness:** 3
**Presentation:** 2
**Contribution:** 3
**Rating:** 5
**Confidence:** 4

**Summary:**

The paper studied distributed asynchronous SGD with heterogeneous communication and computation times for non-convex stochastic optimization problems. In particular, the paper proposed a new algorithm called Shadowheart SGD and analyzed his time complexity. The time complexity is proven to be optimal in the famly of centralized methods with compressed communication by providing a lower bound. This time complexity is also shown to be better than those in the literature.

**Strengths:**

1. The paper considered a general scenario of distributed/federated learning by considering asynchronous SGD, arbitrary computation and communication time, and unbiased compression.

2. The paper proposed a new and non-trivial algorithm called Shadowheart SGD, which is optimal in terms of time complexity.

3. The paper introduced equilibrium time, which is a key parameter to characterize the number of computations and communications in Shadowheart SGD.

4. The the paper also proposed Adaptive Showheart SGD, which does not require the knowledge of equilibrium time, the communication and the computation times.

**Weaknesses:**

1. The paper was not well written and is very hard to read. It reads like a summary of the results that are in the appendix. In addition, the intuitive explanations are very few and many notations were not well explained.

2. The paper considered only IID data distribution. This almost rules out the application of the algorithm in federated learning. The results are nice contributions to the existing results but it seems minor.

3. It lacks experimental results. All the experiments use either logistic regression with MNIST dataset or quadratic optimization and multiplicative noise. Since the paper proposed a new algorithm, i believe the experiments need to be much more extensive.

**Questions:**

1. In page 3, why $D_v$ is a distribution on $\mathbb{S}_{\xi}$?

**Limitations:**

The paper only focuses on the IID dataset.

---

> ### Author Rebuttal · Authors · 2024-08-05
>
> Thank you for the comments! Let us respond to the weaknesses:
>
> > The paper was not well written and is very hard to read. It reads like a summary of the results that are in the appendix. In addition, the intuitive explanations are very few and many notations were not well explained.
>
> We are sorry if some parts of the paper are not clear. We will be happy to clarify the results. Note that we have an extensive list of notations in Section B. The intuitive explanations of our algorithm, the lower bound, and the equilibrium time are presented in Sections 4, 5, and 6.
>
> > The paper considered only IID data distribution. This almost rules out the application of the algorithm in federated learning. The results are nice contributions to the existing results but it seems minor.
>
> This is true, we do not hide this (Lines 33-34). However, in this paper, we chose to focus on this homogeneous setting due to the significant and unexpected results it yields, such as the equilibrium time. We believe that we developed enough new results for one paper. Of course, the non-IID setup is equally important, and we are currently considering it, but it will require a separate project.
>
> > It lacks experimental results. All the experiments use either logistic regression with MNIST dataset or quadratic optimization and multiplicative noise. Since the paper proposed a new algorithm, i believe the experiments need to be much more extensive.
>
> Notice that our experiments are deliberately and scrupulously designed to capture the dependencies from the theory. We have many plots with different input parameters to understand the behaviors of algorithms. But in order to run controlled experiments, we have to exclude all external noise and parameters that more complex losses and models introduce. That is why we choose standard optimization problems that have fewer hyperparameters and uncertainties.
>
> > In page 3, why
>
> There is a typo. Thank you, we will fix it!

---

### Official Review · Reviewer_VesE · 2024-07-16

**Soundness:** 3
**Presentation:** 4
**Contribution:** 3
**Rating:** 7
**Confidence:** 3

**Summary:**

The paper proposes a method (Shadowheart SGD) for centralized asynchronous optimization with compression. The lower complexity bounds are proposed, as well. It is shown that Shadowheart SGD achieves the lower bounds, meaning that the method is optimal.

**Strengths:**

The main strength is the development of an optimal method.

The paper contribution is outlined and clearly presented.

The proposed method is compared to others in several regimes of communication delays, showing how Shadowheart SGD can outperform its counterparts.

**Weaknesses:**

Personally, I did not understand the discussion of equilibrium time well. I understand that it comes from the analysis; also the examples are illustrative. But still, can equilibrium time have some interpretation like mixing time or first hit time in Markov chains? Or maybe it is possible to interpret equilibrium time using some random process, like the authors do in [1].

Also citation of MARINA on line 76, page 3 is missing.

[1] Vogels, T., Hendrikx, H., & Jaggi, M. (2022). Beyond spectral gap: The role of the topology in decentralized learning. Advances in Neural Information Processing Systems, 35, 15039-15050.

**Questions:**

See question on equilibrium time above.

**Limitations:**

The paper is mostly theoretical and does not have negative societal impact.

---

> ### Author Rebuttal · Authors · 2024-08-05
>
> Thank you very much for the review! Let us clarify the weaknesses and questions:
>
> > Personally, I did not understand the discussion of equilibrium time well. I understand that it comes from the analysis; also the examples are illustrative.
>
> Indeed, the equilibrium time, in some sense, is a "mixing time," a time when the server can aggregate enough information from stochastic vectors to confidently do the optimization step since randomness is stabilized by that time. We believe one of the best possible intuitive explanations of the equilibrium time is presented in Lines 130-138. The larger time $t,$ the larger $b_i$ and $m_i.$ Due to the law of large numbers, $g^k$ tends to exact gradient $\nabla f(x^k)$ when $t \to \infty.$ The equilibrium time $t^*$, in some sense, says when $g^k$ is close enough to $\nabla f(x^k)$, and the server can confidently do the next optimization step.
>
> > Also citation of MARINA on line 76, page 3 is missing.
>
> We will add the relevant citation. Thank you.

---

> > ### Comment · Reviewer_VesE · 2024-08-09
> > **Answer to Authors**
> >
> > Dear Authors,
> >
> > Thank you for the reply. I decided to maintain my score.

---

### Official Review · Reviewer_Ptpi · 2024-07-18

**Soundness:** 3
**Presentation:** 3
**Contribution:** 3
**Rating:** 6
**Confidence:** 4

**Summary:**

This paper considers distributed and centralized smooth non-convex optimization when workers have different computation and communication speeds. These different speeds on the workers (and even possibly the server) characterize the problem's device heterogeneity. The authors provide a new algorithm that uses unbiased compression and, more importantly, assigns a time budget for each communication round to optimize for total time complexity. This time budget is calculated based on the speeds of different workers and is the main contribution of this paper. Adapting classic lower bounds for distributed zero-respecting algorithms, and using RandK compression algorithm, the authors show their algorithm is optimal. The authors also offer extensions, such as an adaptive variant of their algorithm, and consider the setting when server communication requires some time.

**Strengths:**

The paper is well written, and a reader not familiar with the literature gets to a very good understanding of the current literature in the area after reading the first few sections of the paper. The results are also presented in a transparent, easy to understand, and mathematically sound manner. The studied problem is a fundamental problem in distributed optimization, and underlies several more complicated seeming problems in Federated learning. The authors attempt at showing optimality of their procedure at least for RandK also provides context on any scope for improvement in the current result. Finally, the authors consider several important extreme limits as well as baselines, making clear comparisons with them. Overall, it is a good paper with only some flaws that can be corrected using remarks and small additional comments, that can be accommodated in the camera ready version.

I support accepting the paper, and am happy to further increase my score based on author rebuttal and discussion.

**Weaknesses:**

I have the following comments in a decreasing order of importance:

1. I am not sure if the lower bound actually holds for all unbiased compression schemes. For instance, if the unbiased compression scheme is to add random noise to the gradient vector before arbitrarily picking a direction to communicate, then it woun't be distributed zero-respecting. Ofcourse, with randK this does not happen because if it picks a co-ordinate it wouldn't make it non-zero if it isn't already. This is an important subtlety and should be incorporated in the discussion, but either removing the above compression scheme I mentioned through an additional constraint in the definition of distributed zero-respecting algorithms (with compression), or state that the lower bound is only for randK compression operator. Otherwise, it is not accurate to say the proposed method is optimal.

2. In the introduction while specifying the scope of the problem, it would also be useful to discuss the issue of bit communication complexity somewhere. It is true that in some settings, the communication time complexity only depends on the total number of communications, but in more optimized channel communication the bit complixty is an important factor as well, and for instance in the general setup $\tau_i$ should be a function of $K$ for say randK compressor. I ofcourse don't expect the authors to solve all problems in a single problem, but discussing this would make the survey more exhaustive in my opinion.

3. The authors should comment on when (12) is a reasonable approximation, even if the calculations are not exact.

4. Small typos: line 112 should be "is not negligible", line 134 should have $t/2$ instead of $t$, unless the authors are using some interlacing idea to compute and communicate simultaneously, in line 217 it should be "it is left",

**Questions:**

1. Do the authors have thoughts on statistical heterogeneity can be considered in this setting? I am concerned that the current algorithm might heavily bias the final output towards the databases of agents that are pretty quick.

2. Do the authors have thoughts on the second weakness I raised above? In particular, it is an open question to obtain algorithms with optimal round and bit communication complexity, and understanding the regime with non-trivial bit complexity can help make progress towards that setting.

**Limitations:**

Nothing big, but please see my comments above in the weaknesses.

---

> ### Author Rebuttal · Authors · 2024-08-05
>
> Thank you for your very positive comments! We now respond to weaknesses:
>
> > 1. I am not sure if the lower bound actually holds for all unbiased compression schemes.
>
> In the lower bound (Theorem O.5), we chose the particular compressor, Rand$K.$ Thus, indeed, our lower bound works only with one compressor. We do not hide this since we state in Theorem O.5, saying, "there exists ... communication oracles," which indicates that the theorem works with fixed compressors. We also admit this in Line 205 of the main part. We will add this detail more explicitly.
>
> We want to note that choosing one particular bad compressor (RandK compressor in this case) is the same as choosing one particular bad function, which is the standard approach in the literature. For instance, Yurii Nesterov, in his celebrated book [1] (like many other works), chooses one particular bad function (special quadratic function) to show the lower bounds. As far as we know, nobody showed that the lower bounds from [1] work for *any* function (because it is a very difficult task, and possibly it is not true). Proving the lower bound for any compressor can be also very difficult and even impossible.
>
> > 2. In the introduction while specifying the scope of the problem, it would also be useful to discuss the issue of bit communication complexity somewhere.
>
> > 2. Do the authors have thoughts on the second weakness I raised above? In particular, it is an open question to obtain algorithms with optimal round and bit communication complexity, and understanding the regime with non-trivial bit complexity can help make progress towards that setting.
>
> Let us clarify this weakness and question. The parameter $\tau_i$ is # of seconds required to send a compressed vector $C(x)$ to the server (e.g., Line 20). For simplicity, let us consider the Rand$K$ compressor. Then $\tau_i$ is # of seconds required to send $K$ non-zero coordinates of the vector $x.$ Now assume that the communication is proportional to the number of coordinates/bits that a worker sends, i.e., it takes
> $\dot{\tau}_i$ seconds is to send a **one coordinate/bit** (see Section 7). Then, clearly, we have
> $$\tau_i = K \times \dot{\tau}_i,$$ and, indeed, $\tau_i$ is a function of $K!$ Hence, $\tau_i$ is proportional to the number of coordinates/bits that a worker sends in one compressed message, which we believe is in line with the expectations of the reviewer. We discuss this in Section 7 for Rand$1$ in detail when comparing the methods. In general, we do not specify the exact dependencies between communication times $\tau_i$ and the number of bits/information that a compressor sends. While it is clear for RandK, for all possible compressors the dependence can be very nontrivial, so we decided to abstract from this away. The optimal time complexity ((10) from the paper) captures bits communication using the times $\tau_i.$ We agree that this should be noted and clarified in the main part, thank you!
>
> P.S. We also asked ourselves what would be the optimal choice of $K$ in Rand$K$ to get the best possible time complexity. It turns out an optimal choice is $K = 1$ (up to a constant factor). See Property 6.2.
>
> > 3. The authors should comment on when...
>
> This $\approx$ is tight up to a constant because
> $$\frac{1}{2 x} \leq \frac{1}{x + y} \leq \frac{1}{x}$$ if $x \geq y,$
> and
> $$\frac{1}{2 y} \leq \frac{1}{x + y} \leq \frac{1}{y}$$ if $y \geq x.$
>
> > 4. Small typos
>
> Thank you for your attention. We will fix them.
>
> > 1. Do the authors have thoughts on statistical heterogeneity can be considered in this setting?
>
> This is a good future research direction that we indeed consider right now. But in this paper, we decided to focus on this setting due to the current non-trivial and non-obvious results (e.g. the equilibrium time) that we believe deserve a separate paper.
>
> [1]: Nesterov, Yurii, Lectures on convex optimization, 2018

---

> > ### Comment · Reviewer_Ptpi · 2024-08-14
> >
> > Thanks for the response; I will keep my positive score and recommend accepting the paper with the aforementioned clarifications.

---

### Official Review · Reviewer_zjmf · 2024-07-22

**Soundness:** 3
**Presentation:** 3
**Contribution:** 3
**Rating:** 7
**Confidence:** 2

**Summary:**

The paper presents a novel method for non-convex stochastic optimization in an asynchronous centralized distributed setup, focusing on improving time complexity in heterogeneous environments. Additionally, the authors demonstrate that the proposed method achieves theoretically optimal time complexity under compression.

**Strengths:**

1. **Novel Method**: The introduction of Shadowheart SGD guarantees finding an $\varepsilon$-stationary point with optimal time complexity in heterogeneous environments.
2. **Theoretical Contributions**: The paper provides rigorous proofs showing that Shadowheart SGD has better or equal time complexity compared to existing centralized methods.
3. **Optimal Time Complexity**: The time complexity analysis shows that Shadowheart SGD can be significantly better than previous methods in many regimes.

**Weaknesses:**

1. The definition and calculation of equilibrium time are complex and not very intuitive. The implicit nature of this definition might make practical implementation and understanding challenging.
2. While the proposed method improves time complexity, the paper does not fully discuss the impact on final convergence and loss in real-world scenarios with limited training datasets. Additionally, the experiments only compare the loss of different SGD methods under the same wall time, but it would be insightful to know if Shadowheart SGD maintains lower loss when comparing the number of training samples.

**Questions:**

1. Can the process of calculating equilibrium time be simplified or made more intuitive? Are there any practical algorithms or tools recommended for this calculation?
2. Similar to Weakness 2. Regarding the impact on convergence and loss in scenarios with limited training datasets, how does Shadowheart SGD perform when comparing the loss against the number of training samples?

**Limitations:**

Yes

---

> ### Author Rebuttal · Authors · 2024-08-05
>
> Thank you for the positive review! Let us address the weaknesses and questions:
>
>  > The definition and calculation of equilibrium time are complex and not very intuitive. The implicit nature of this definition might make practical implementation and understanding challenging.
>
> Unfortunately, with the equilibrium time, we have a situation like the math community has, for instance, with the Gamma function, which also does not have an explicit formula, but it does not make the Gamma function less useful. We devoted the whole Section 6 explaining the idea behind the equilibrium time. Note that in Section 6.1, we describe an algorithm for finding the equilibrium time numerically.
>
> > Can the process of calculating equilibrium time be simplified or made more intuitive? Are there any practical algorithms or tools recommended for this calculation?
>
> Yes, we describe an algorithm in Section 6.1.
>
> > While the proposed method improves time complexity, the paper does not fully discuss the impact on final convergence and loss in real-world scenarios with limited training datasets. Additionally, the experiments only compare the loss of different SGD methods under the same wall time, but it would be insightful to know if Shadowheart SGD maintains lower loss when comparing the number of training samples.
>
> > Similar to Weakness 2. Regarding the impact on convergence and loss in scenarios with limited training datasets, how does Shadowheart SGD perform when comparing the loss against the number of training samples?
>
> It is easy to show that Shadowheart SGD requires the number of training samples less than Rennala SGD and Asynchronous SGD. All these three algorithms utilize the computation resources almost all the time during optimization processes, meaning that they use
> $$\approx \sum_{i=1}^n \lfloor\frac{t}{h_i}\rfloor$$
> training samples by a time $t.$ Since the time complexity $t_{\textnormal{shadowheart}}$ of Shadowheart SGD is smaller than in other methods, we can conclude that
> $$\sum_{i=1}^n \lfloor\frac{t_{\textnormal{shadowheart}}}{h_i}\rfloor < \sum_{i=1}^n \lfloor\frac{t_{\textnormal{rennala}}}{h_i}\rfloor$$
> and
> $$\sum_{i=1}^n \lfloor\frac{t_{\textnormal{shadowheart}}}{h_i}\rfloor < \sum_{i=1}^n \lfloor\frac{t_{\textnormal{asynchronous}}}{h_i}\rfloor.$$ Therefore, Shadowheart SGD uses less training samples to find an $\varepsilon$--stationary point.
>
> Comparing to Minibatch SGD, which requires $\frac{n L \Delta}{\varepsilon} + \frac{\sigma^2 L \Delta}{\varepsilon^2}$ training data to find an $\varepsilon$--stationary point, our method indeed can *calculate* more stochastic gradients because $b_i = \lfloor \frac{t^*}{h_i} \rfloor$ can be large if $h_i$ is small. However, Minibatch SGD provably requires more time to solve the problem since some workers are idle during the optimization processes (see Lines 55-56 and Table 1). Notice that we work with i.i.d. stochastic gradients, and nothing stops us from reusing previously used training samples. For instance, we can organize a stochastic gradient calculation in a way that workers calculate $\nabla f(x;\xi),$ where $\xi$ is a uniformly random sample *with replacement*.
>
> What if we can not reuse samples? When we were writing the paper, we tried to use all workers' time and utilize them as much as we could to improve the performance. That is why we take $b_i = \lfloor \frac{t^*}{h_i} \rfloor.$ But if we want to limit the number of samples that Shadowheart SGD uses, then we can slightly modify this algorithm and take $b_i = \min \left[\lfloor \frac{t^*}{h_i} \rfloor, b_{\max}\right],$ where $b_{\max}$ is some parameter that bounds the number of used samples.

---

### Decision · Program_Chairs · 2024-09-25

**Decision:**

Accept (poster)

**Comment:**

This paper studies a centralized distributed stochastic optimization algorithm focusing on the time complexities thru optimizing the allocation of computation time of the workers under knowledge of the computation time & delay time of these workers. While the paper has received generally positive reviews and I thus recommend acceptance. However, it should be noted that some of the important concepts such as the notion of equilibrium time is not well explained, and the paper has deferred much of these important discussions to the appendix. I recommend the authors to consider these comments and revise their paper thoroughly before publication.